# Hardwiring tissue-specific AAV transduction in mice through engineered receptor expression

James Zengel[1,11], Yu Xin Wang[2,10,11], Jai Woong Seo [3,11], Ke Ning[4,11], James N. Hamilton[2], Bo Wu[3], Marina Raie [3], Colin Holbrook[2], Shiqi Su[2], Derek R. Clements[1,5], Sirika Pillay[1], Andreas S. Puschnik [1], Monte M. Winslow [6,7], Juliana Idoyaga [1,5], Claude M. Nagamine[8], Yang Sun [4,9], Vinit B. Mahajan[4,9], Katherine W. Ferrara [3], Helen M. Blau [2] & Jan E. Carette [1]✉

The development of transgenic mouse models that express genes of interest in specific cell types has transformed our understanding of basic biology and disease. However, generating these models is time- and resource-intensive. Here we describe a model system, SELective Expression and Controlled Transduction In Vivo (SELECTIV), that enables efficient and specific expression of transgenes by coupling adeno-associated virus (AAV) vectors with Cre-inducible overexpression of the multi-serotype AAV receptor, AAVR. We demonstrate that transgenic AAVR overexpression greatly increases the efficiency of transduction of many diverse cell types, including muscle stem cells, which are normally refractory to AAV transduction. Superior specificity is achieved by combining Cre-mediated AAVR overexpression with whole-body knockout of endogenous *Aavr*, which is demonstrated in heart cardiomyocytes, liver hepatocytes and cholinergic neurons. The enhanced efficacy and exquisite specificity of SELECTIV has broad utility in development of new mouse model systems and expands the use of AAV for gene delivery in vivo.

Genetically engineered in vivo mouse models have had an enormous impact for biomedical research across many fields, including oncology, neurobiology, hereditary genetic disorders and infectious disease[1–8]. This research often requires the tissue-specific expression of transgenes to better mirror human diseases, to visualize pathways being studied or to modulate the expression of a disease-related gene. Tissue-specificity can be achieved through the generation of transgenic mouse models with cell-type-specific promoter-driven expression of the gene of interest or integration of cell-type-specific Cre recombinase[9–11]. However, the generation of transgenic mice is time consuming

[1]Department of Microbiology and Immunology, Stanford University School of Medicine, Stanford, CA, USA. [2]Baxter Laboratory for Stem Cell Biology, Department of Microbiology and Immunology, Stanford University School of Medicine, Stanford, CA, USA. [3]Department of Radiology, Stanford University School of Medicine, Stanford, CA, USA. [4]Department of Ophthalmology, Stanford University School of Medicine, Stanford, CA, USA. [5]Immunology Program, Stanford University School of Medicine, Stanford, CA, USA. [6]Department of Genetics, Stanford University School of Medicine, Stanford, CA, USA. [7]Department of Pathology, Stanford University School of Medicine, Stanford, CA, USA. [8]Department of Comparative Medicine, Stanford University School of Medicine, Stanford, CA, USA. [9]Palo Alto Veterans Administration, Palo Alto, CA, USA. [10]Present address: Center for Genetic Disorders and Aging, Sanford Burnham Prebys Medical Discovery Institute, La Jolla, CA, USA. [11]These authors contributed equally: James Zengel, Yu Xin Wang, Jai Woong Seo, Ke Ning. ✉e-mail: carette@stanford.edu

**Fig. 1 | Production and characterization of a platform for in vivo over-expression of AAVR in mice. a**, Construction of SELECTIV mice that allow Cre-mediated overexpression of the mouse AAV receptor (*Aavr*). The construct contains the CAG promoter followed by the loxP-STOP-loxP sequence, a strong transcription stop sequence that can be removed by Cre recombination. It encodes the mouse *Aavr* fused to mCherry and spCas9 after an F2A self-cleaving peptide sequence. This cassette was inserted into the C57BL/6 mouse genome using targeted integration in the H11 locus. SELECTIV mice can be crossed with mice expressing Cre recombinase under specific promoters to produce mice with cell-type- or tissue-specific overexpression of *Aavr*. **b**, Transduction of MEFs overexpressing AAVR. MEFs were generated from embryos from a SELECTIV-WB and WT mouse breeding (individual derived MEFS, control $n = 4$; SELECTIV-WB $n = 6$ for AAV2, AAV8 and AAV9, and $n = 5$ for AAV4). MEFs were transduced with AAV2-, AAV4-, AAV8- and AAV9-luciferase. Transduction was assessed by luciferase activity 48 h later. Mean value and s.e.m. are shown. Fold-changes are indicated, and the *P* value was calculated using a two-way analysis of variance (ANOVA) with Holm–Šídák's multiple comparisons test, where ****$P < 0.0001$; NS, not

significant (AAV2 $P = 1.6 \times 10^{-12}$, AAV4 $P = 0.65$, AAV8 $P < 1 \times 10^{-15}$, AAV9 $P < 1 \times 10^{-15}$). **c**, CRISPR genome editing using AAV-encoded sgRNA and endogenous spCas9 in SELECTIV-WB MEFs. MEFs were transduced at an MOI = 50,000 vg per cell with AAV9 encoding an sgRNA targeting PCSK9 under a U6 promoter and a GFP reporter. GFP-positive cells were sorted and isolated and indels were detected by deep sequencing. **d**, Schematic of study to test in vivo transduction in control and SELECTIV-WB mice after intramuscular (i.m.) injection of AAV2-luc ($10^{11}$ vg) into the tibialis anterior (TA) muscle. **e**, In vivo transduction was measured over time (control $n = 4$ mice days 4–60 and $n = 3$ mice day 120, SELECTIV-WB $n = 4$ mice days 4–28 and $n = 3$ mice days 33–120) in mice injected (i.m.) with AAV2-luciferase by in vivo imaging and quantification. Mean value and s.e.m. are shown. The *P* value was calculated using a two-way ANOVA by fitting a mixed model with two-tailed Holm–Šídák's multiple comparisons test, where *$P < 0.05$ and **$P < 0.01$ (day 4 $P = 0.034$, day 7 $P = 0.0090$, day 14 $P = 0.057$, day 21 $P = 0.11$, day 28 $P = 0.11$, day 33 $P = 0.014$, day 35 $P = 0.017$, day 39 $P = 0.11$, day 46 $P = 0.0035$, day 53 $P = 0.11$, day 60 $P = 0.017$, day 120 $P = 0.0035$). NGS, next-generation sequencing; RLU, relative luciferase units.

and expensive and new models are required for each gene of interest, which has severely limited the breadth and scale of these studies[12–14]. Transgene expression using viral vectors provides an alternative, which allows for temporal expression of genes in mice with reduced cost and effort. Adeno-associated virus (AAV) has emerged as a viral vector of choice for in vivo transduction and is used as an important research tool to assess gene function by allowing overexpression or knockout (KO) in diverse research areas including neurobiology[15,16]. Moreover, AAV has become a core platform for the treatment of genetic diseases with multiple approved drug products[17–20] and many others in clinical development[21] and, therefore, is often used to test novel gene replacement or editing therapies in animal models of disease[16]. Limitations in the usage of AAV in murine models include cell types that are refractory to transduction and difficulties achieving tissue-specificity.

Here we generate a robust system for in vivo transgene delivery by combining the versatility of AAV-mediated transgene delivery with the specificity of cell-type-specific Cre-mediated recombination. Our approach exploits the absolute dependence of AAV transduction on the expression of its receptor[22], AAVR (also named KIAA0319L). Because almost all natural and engineered capsid variants of AAV require AAVR[22–26], the approach is compatible with a wide variety of existing AAV vectors. This allows flexibility in the choice of AAV variant tailored to the research question. The system allows for the generation of mice with specific overexpression of AAVR through breeding with available Cre mouse lines, as well as the option to detarget AAV from other tissue when paired with *Aavr*-KO mice. By directing AAV transduction through the regulated expression of its receptor, the approach provides precise control over which cells express the AAV-vectored transgene.

## Results

### Generation of mice with controllable AAVR overexpression

We hypothesized that AAV transgene expression could be targeted to specific tissues or cell types by selectively overexpressing AAVR. To control AAVR expression, we designed a Cre-mediated AAVR overexpression cassette (Fig. 1a). Cre-mediated removal of the stop sequence results in high levels of AAVR fused to the red fluorescent protein mCherry. We further cloned *spCas9* behind an F2A self-cleaving peptide in frame with AAVR-mCherry. Transgenic mice that contain a single copy of this cassette at the *Hipp11* (*H11*) locus were generated using site-specific integration[27]. Mice containing this cassette were named SELECTIV.

To study the impact of AAVR overexpression in vivo, we generated a mouse line with full body overexpression of AAVR (SELECTIV-WB) through a cross with E2A-Cre mice[28] (Extended Data Fig. 1). These mice expressed high levels of AAVR in all tissues analyzed (Extended Data Fig. 2a,b). The mice were viable and fertile and had no apparent differences in immune cell profiles compared with wild-type (WT) mice (Extended Data Fig. 2c,d).

We assessed AAV transduction in mouse embryonic fibroblasts (MEFs) derived from control (*Aavr*+/+) or SELECTIV-WB embryos. To test functionality of AAVR-mCh overexpression, control and SELECTIV-WB MEFs were transduced with three AAVR-dependent serotypes (AAV2, AAV8 and AAV9)[22] and one AAVR-independent serotype (AAV4)[23] encoding luciferase (Fig. 1b). As expected, AAV4 transduction was not affected by AAVR overexpression. AAVR overexpression greatly increased transduction by all three AAVR-dependent serotypes. The fold increase was especially apparent for AAV8 and AAV9, which are known to have low transduction efficiency in vitro[29,30].

We incorporated spCas9 in the SELECTIV construct to allow for gene KO experiments by AAV delivery of a single guide RNA (sgRNA) targeted against a gene of interest. To test the functionality of this approach, we transduced SELECTIV-WB MEFs with AAV9 encoding sgRNA targeting PCSK9. This resulted in efficient genome editing (>50% indel formation) as we demonstrated using deep sequencing of an amplicon corresponding to the genomic locus targeted by the gRNA

(Fig. 1c). These results show that the SELECTIV system functionally expresses spCas9 for use in CRISPR-based experiments.

### AAVR overexpression allows for increased transduction in vivo

The usage of AAV in animal models often requires high viral titers to achieve sufficient transgene expression in vivo. Moreover, certain cell types are inherently more refractory to AAV transduction. Given its essential role in AAV transduction of multiple serotypes, we reasoned that AAVR overexpression would enhance in vivo transduction.

We first focused on muscle transgene delivery, due to the extensive use of AAV-based vectors to study and treat muscle diseases[31–33]. Mice with whole-body AAVR overexpression (SELECTIV-WB) and control mice (*Aavr*+/+) were intramuscularly injected with AAV2-luciferase (Fig. 1d). To test the durability of transgene expression, muscle was injured with a BaCl$_2$ injection into the transduced muscle. Luciferase activity was tracked over time using in vivo imaging (Fig. 1e and Extended Data Fig. 3). At early timepoints (4 and 7 d post-injection) AAVR overexpression resulted in a significant increase in transduction compared with control mice. Enhanced transgene expression remained after injury and was maintained for up to 120 d post-injection. We found sustained transgene expression over time and after injury, potentially due to transduction of a persistent cell population.

### AAVR overexpression enhances muscle stem cell (MuSC) transduction

In SELECTIV-WB mice, there was a durable increase in AAV transduction. We hypothesized that this was due to increased transduction of MuSCs (also known as satellite cells), which are required to regenerate injured muscle[34,35]. Many AAV serotypes have poor MuSC transduction, while muscle fibers are efficiently transduced[36,37]. It has been proposed that the quiescent state or unique niche of MuSCs is responsible for poor transduction by AAV[36]. Published gene expression data indicate that MuSCs are amongst the cell types with the lowest endogenous *Aavr* expression in a large survey of mouse tissues (Extended Data Fig. 4)[38]. We generated SELECTIV;Pax7-CreER^T2 (SELECTIV-Pax7^CE) mice to enable tamoxifen-inducible AAVR overexpression in MuSCs[39] (Extended Data Fig. 5). This system was first tested ex vivo with MuSC-derived primary myoblasts from SELECTIV-Pax7^CE mice, which were tamoxifen- or mock-treated and transduced with AAV2- or AAV8-GFP (Fig. 2a). Fluorescence microscopy showed that AAVR overexpression increased transduction efficiency (Fig. 2b). Transduction was quantified with AAV2 or AAV8 encoding luciferase. AAVR overexpression increased transduction by AAV2 (8.5-fold) and AAV8 (403-fold) (Fig. 2c). These results highlight a critical role of AAVR expression for efficient ex vivo MuSC transduction.

We next assessed MuSC transduction in vivo, through intramuscular delivery of AAV2-GFP into tamoxifen-treated and control untreated SELECTIV-Pax7^CE mice (Fig. 2d). After 21 d, GFP expression was measured in MuSCs (CD11b−/CD45−/Sca1−/CD31−/α7-integrin+/CD34+) (Fig. 2e,f). There was a modest transduction of MuSCs in control mice (3–5.5% GFP+), while AAVR overexpression increased MuSC transduction by >4-fold (14–26% GFP+). These results show that AAVR expression is a limiting factor for AAV transduction in MuSCs, and suggest that increased AAVR expression in cells refractory to AAV transduction greatly improves transduction.

We next tested transduction of MuSCs after systemic delivery of AAV. SELECTIV-Pax7^CE mice with or without tamoxifen treatment were intravenously injected with AAV9-GFP and MuSC transduction was assessed (Fig. 2g). In the tibialis anterior muscle, an average of 0.13% of MuSCs were transduced in the control group, while in the tamoxifen-treated group an average of 4.7% of MuSCs were transduced (36-fold increase; Fig. 2h). In the diaphragm, there was an average 0.18% MuSC transduction in the control group and 5.3% in the treated group (30-fold; Fig. 2h). These results demonstrate a clear role of AAVR in MuSC transduction in vivo, suggesting that the effects of their

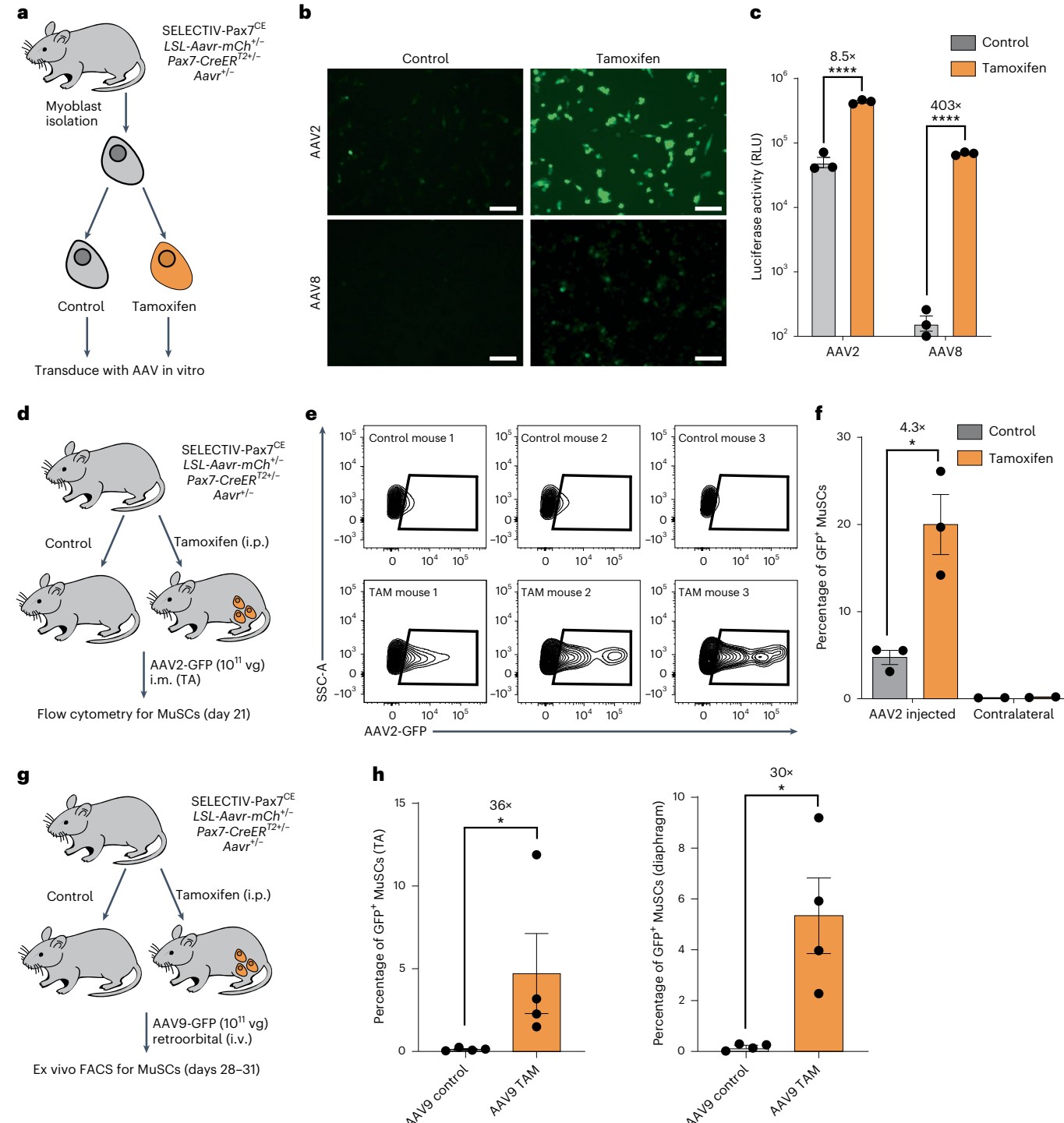

**Fig. 2 | Targeting of MuSCs in vitro and in vivo with inducible AAVR overexpression increased AAV transduction. a**, SELECTIV-Pax7^CE mice were generated, which allow for overexpression of AAVR in Pax7⁺ cells (MuSCs and myoblasts) after tamoxifen treatment. These cells were transduced with AAV in vitro to assess the role of AAVR overexpression in these cells. **b**, SELECTIV-Pax7^CE myoblasts with or without tamoxifen treatment were transduced with AAV2- or AAV8-GFP and imaged at 48 h post-transduction. Scale bars, 100 μm. **c**, Similarly treated cells were transduced with AAV2- or AAV8-luciferase (*n* = 3 wells) and luciferase activity was assessed at 48 h post-transduction. Mean value and s.e.m. are shown. The *P* value was calculated using a two-way ANOVA with Holm–Šídák's multiple comparisons test, with ****P < 0.0001 (AAV2 P = 1.6 × 10⁻⁵, AAV8 P = 1.1 × 10⁻⁸). **d**, In vivo transduction of MuSCs by AAV2-GFP was assessed in SELECTIV-Pax7^CE mice with or without tamoxifen treatment. **e**, Cells were isolated from the full TA muscle of the injected leg or the uninjected contralateral leg

and the percentage of the transduced MuSCs was determined by FACS for GFP-positive cells in the CD11b⁻/CD45⁻/Sca1⁻/CD31⁻/a7-integrin⁺/CD34⁺ population, and the percentage of GFP-positive MuSCs was quantified for each mouse. **f**, The percentage of transduced cells in the MuSC population was calculated for each group for the injected leg and the contralateral leg. Mean value and s.e.m. are shown (*n* = 3 mice, *n* = 2 contralateral). The *P* value was calculated using an unpaired *t*-test (two-tailed), with *P < 0.05 (AAV2 injected P = 0.013). **g**, In vivo transduction of MuSCs by AAV9-GFP after systemic (intravenous (i.v.)) delivery was assessed in SELECTIV-Pax7^CE mice with or without tamoxifen treatment. **h**, MuSCs were isolated from the TA muscle or the diaphragm, and the numbers of GFP⁺ cells were assessed by flow cytometry. Mean value and s.e.m. are shown (*n* = 4 mice). The *P* value was calculated using a two-tailed Mann–Whitney test, with *P < 0.05. The comparisons gave P = 0.0286 for the TA and P = 0.0286 for the diaphragm. i.p., intraperitoneal.

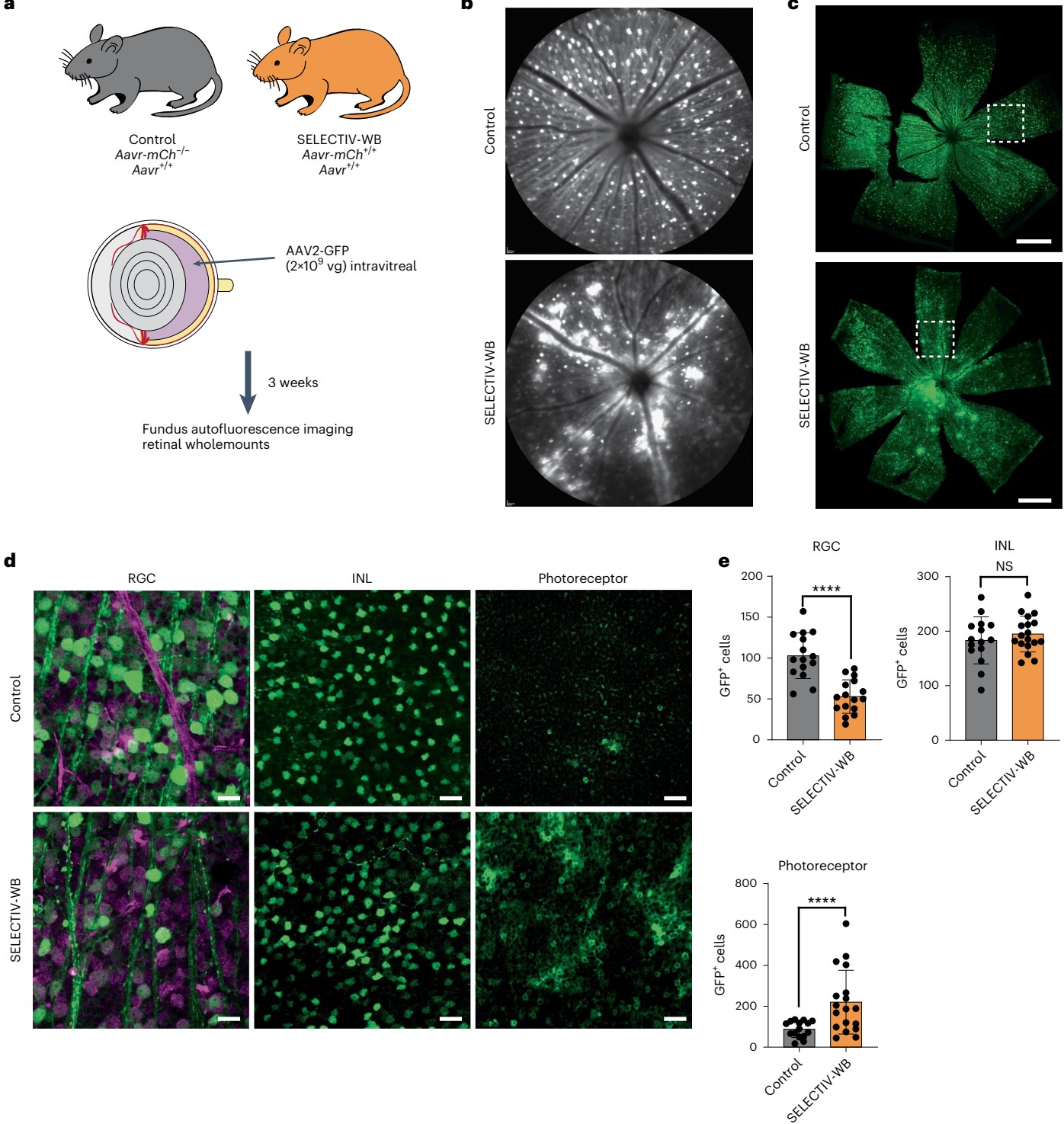

**Fig. 3 | AAVR overexpression changed the transduction profile in the eye towards the photoreceptor layer upon i.v. injection. a**, Control and SELECTIV-WB mice were injected intravitreally with AAV2-GFP (*n* = 5 mice per group, male and female, 6–7 months of age). Retinal transduction was tracked over time using fundus autofluorescence imaging and ex vivo by fluorescent microscopy of retinal wholemounts. **b**, Representative fundus photographs show GFP expression in retinas of live mice 3 weeks after AAV2-GFP intravitreal injection. **c**, Representative confocal images of retinal wholemounts (RGC layer facing upwards) of SELECTIV-WB and control mice. **d**, High-magnification view of wholemounts taken by confocal Z-stacks; 500 μm from optic nerve head; white dotted squares in **c**; RGCs are labeled with RBPMS in purple. **e**, Quantification of GFP-positive cells in each layer from the control and SELECTIV-WB groups. Scale bars, 500 μm (**c**) and 50 μm (**d**). For quantification, *n* = 5, 6 retinas (3 images per retina; 2 samples in RGC SELECTIV-WB group had 2 images per retina). Mean value and s.e.m. are shown. The *P* value was calculated using an unpaired *t*-test (two-tailed), with ****$P < 0.0001$, **$P < 0.01$ (RGC $P = 3.7 \times 10^{-6}$, INL $P = 0.38$, photoreceptor $P = 0.0032$).

quiescent state or niche either are not important for transduction or that these factors can be overcome by overexpression of AAVR.

## Transduction of deeper retinal layers in SELECTIV mice

To further evaluate the utility of the SELECTIV system for providing a rapid and flexible expression system in mice, we applied it to retinal transduction. AAV is used in gene replacement therapies treating blindness[17]. Mouse models for testing AAV vectors or studying disease mechanisms in this organ are an active area of research[21]. Transduction differences based on AAVR expression were assessed in SELECTIV-WB and control mice ($Aavr^{+/+}$) after intravitreal injection of AAV2-GFP (Fig. 3a). The pattern of GFP expression was determined in the retinas by fundus autofluorescence imaging. Control mice had uniform but moderate GFP, while SELECTIV-WB mice had large areas of robust transduction in regions along the blood vessels (Fig. 3b). Histological analysis shows transduction present throughout the inner retinal layer of the retinal ganglion cells (RGCs) in both groups (Fig. 3c). We then evaluated the AAV-based transduction in RGCs, amacrine cells (inner nuclear layer (INL)) and photoreceptors by co-labeling with RBPMS, a reliable marker to identify and quantify RGCs. We found strong GFP expression in RGCs and expression was partially colocalized with RBPMS labeling (Fig. 3d). Cell quantification (Fig. 3e) indicated that photoreceptor layer transduction in the SELECTIV-WB group was significantly higher than in the control group ($220.11 \pm 155.97$ versus $87.80 \pm 39.93$); RGC layer transduction was lower in the SELECTIV-WB group compared with the control group ($52.94 \pm 20.46$ versus $102.80 \pm 27.96$); INL transduction was similar in retinas transduced in the SELECTIV-WB and control groups ($194.94 \pm 32.90$ versus $183.20 \pm 43.07$). These results demonstrated that AAVR expression affected cell-type specificity and the depth of transduction after intravitreal injection with AAV2. The overexpression of AAVR allows for transduction of both the inner and outer mouse retina, with deeper transduction into the photoreceptor cells. The SELECTIV system has demonstrated that AAVR overexpression enhances the efficiency of retina transduction without more complicated and invasive procedures such as subretinal injection.

## SELECTIV-KO mice allow for precise in vivo transduction

The preceding results indicate that AAVR overexpression can significantly enhance in vivo transduction and allow targeting using targeted Cre. Given the essentiality of AAVR for in vivo transduction[22], we reasoned that generating SELECTIV mice with homozygous inactivation of endogenous *Aavr* (SELECTIV-KO mice) would effectively prevent transduction of all nontarget cell types. Integrating mouse lines that express Cre from tissue-specific promoters[9] with SELECTIV-KO mice would generate mice hardwired for efficient and cell-type-selective AAV transduction. These mice would allow for expression of desired transgenes in specific cells, providing a flexible and cost-efficient alternative to classical approaches of generating new transgenic mouse lines.

We generated SELECTIV-KO-Myh6 mice that express Cre in heart cardiomyocytes and modestly in pulmonary vascular smooth muscle cells[40], while endogenous *Aavr* is not expressed (Fig. 4a and Extended Data Fig. 6). As littermate controls, we use mice that do not express Cre and are heterozygous for *Aavr* (named 'control'). We weighed the distinct advantage of using littermate controls[41] against the potential disadvantage of a heterozygous *Aavr*-KO allele which may show haploinsufficiency. Although we cannot exclude haploinsufficiency in all tissues, we have previously found that heterozygous deletion of endogenous *Aavr* does not significantly reduce AAV9-luciferase transduction upon systemic administration[41]. To assess transduction in these mice we delivered AAV9-luc systemically through intravenous injection. In vivo imaging revealed a gradual increase of luciferase expression in SELECTIV-KO-Myh6 and control mice, with spatially restricted expression in SELECTIV-KO-MyH6 mice (Extended Data Fig. 7a). At day 21, luciferase activity was determined in lysates of several major organs (Fig. 4b). As expected for AAV9 transduction in mice expressing endogenous *Aavr*[42], luciferase activity in control mice was high in liver, heart and muscle, and lower in lung. In striking contrast, the SELECTIV-KO-Myh6 mice had an exquisitely specific transduction pattern. Luciferase activities in the heart and lungs exceeded those of control mice by almost tenfold, while luciferase activity levels in the liver and muscle were reduced to near background levels. These reductions were comparable to those observed in AAVR-KO mice, which have endogenous *Aavr* deleted but do not re-express AAVR (Fig. 4a,b).

To confirm these findings using a different transgene and AAV vector, we injected SELECTIV-KO-Myh6 and control mice with AAV-PHP.eB encoding GFP[43]. At day 28, transduction was evaluated by fluorescence imaging of heart and liver sections (Fig. 4c and Extended Data Fig. 7b). The liver of the control mouse was transduced well with limited transduction in the heart. In SELECTIV-KO-Myh6 mice, most cells in the heart showed high GFP fluorescence while no fluorescence was observed in liver cells.

Having demonstrated a strong effect of AAVR expression on selective transduction, we wanted to determine its effect on biodistribution of the AAV vector. Because AAVR binds directly to AAV particles and is rapidly endocytosed from the cell surface in tissue culture[22,44–46], we hypothesized that it could affect biodistribution of AAV particles in vivo. To test this, we used positron-emission tomography (PET) to track $^{64}$Cu-labeled AAV9 vector distribution shortly after systemic delivery by intravenous injection (Fig. 4d)[47], which allows for tracking of AAV circulation and deposition over time without requiring terminal endpoints. Compared with control mice, SELECTIV-KO-Myh6 mice had increased AAV particle accumulation in the heart while liver accumulation was decreased (Fig. 4e). To confirm the net accumulation of $^{64}$Cu-AAV9 in heart, radioactivity was quantified ex vivo (Fig. 4f). There was greater uptake of $^{64}$Cu-AAV9 in the heart of SELECTIV-KO-Myh6 mice (12 percent injected dose per gram (%ID g$^{-1}$)) compared with

**Fig. 4 | The SELECTIV-KO platform allows for efficient transduction of a tissue of interest with near complete reduction of transduction of nontarget tissue after systemic delivery of AAV. a**, SELECTIV-KO mice were bred with Myh6-Cre mice on an *Aavr*$^{KO}$ background to generate mice with AAVR overexpression in the heart and lung while lacking endogenous *Aavr* expression (SELECTIV-KO-Myh6). Littermate controls heterozygous (control) or homozygous for the endogenous *Aavr*-KO allele (AAVR-KO) were also produced during breeding. **b**, Control (*n* = 6 mice), SELECTIV-KO-Myh6 (*n* = 8 mice) and AAVR-KO (*n* = 6 mice) mice (*n* = 4, 6, 2 mice, respectively, for muscle) were injected with 3 × 10$^{10}$ vg AAV9-luciferase by systemic (i.v.) injection (both male and female, 6–8 weeks of age). At 21 d after injection, mice were euthanized and luciferase activity was determined in indicated organ lysates to determine AAV transduction efficiency. Mean value and s.e.m. are shown. The horizontal dashed line indicates background luminescence in nontransduced controls. Fold-changes are indicated, and the *P* value was calculated using an ordinary one-way ANOVA with a Holm–Šídák's multiple comparisons post-test, with \*\*\*\**P* < 0.0001, \*\*\**P* < 0.001, \*\**P* < 0.01 and \**P* < 0.05 (control versus SELECTIV-KO-Myh6: liver *P* = 9.0 × 10$^{-6}$, heart

*P* = 7.9 × 10$^{-5}$, lung *P* = 0.00051, muscle *P* = 1.0 × 10$^{-5}$; SELECTIV-KO-Myh6 versus AAVR-KO: liver *P* = 0.16, heart *P* = 4.3 × 10$^{-11}$, lung *P* = 8.9 × 10$^{-7}$, muscle *P* = 0.11). **c**, Control and SELECTIV⁻KO-Myh6 (heart-specific) mice were injected with 2 × 10$^{11}$ vg PHP.eB-GFP by systemic (i.v.) injection. Mice were euthanized at 28 d post-transduction and transduction was determined in liver and heart by fluorescence microscopy. Scale bars, 200 μm. Similar results were seen in three biological replicates. **d**, Mice with varying levels of AAVR (control, AAVR-KO or SELECTIV-KO-Myh6, *n* = 3) were injected with AAV9 particles directly labeled with a positron emitter (Cu-64). **e**, PET was used to track particle circulation over time and projected PET/CT images are shown over 21 h, comparing control, AAVR-KO and SELECTIV-KO-Myh6 mice. **f**, Ex vivo quantification of AAV vector particles present in dissected heart, liver, lung and blood as determined at 22 h post-injection by PET. Mean value and s.e.m. are shown (*n* = 3 mice). The *P* value was calculated using an ordinary one-way ANOVA with a Holm–Šídák's multiple comparisons post-test, with \*\**P* < 0.01, \**P* < 0.05 (heart *P* = 0.012, liver *P* = 0.0053, lung *P* = 0.16, blood *P* = 0.17). Bio-D, biodistribution; % ID/cc, percent injected dose per cubic centimeter.

AAVR-KO mice (2.7 %ID g$^{-1}$) and control mice (2.4 %ID g$^{-1}$). There was also a modest but significant decrease in the amount of capsid in the liver of SELECTIV-KO-Myh6 mice. There were no statistically significant

differences between control and SELECTIV-KO-Myh6 mice in lung and blood. These results suggest that AAVR expression affects both AAV particle biodistribution and AAV transduction.

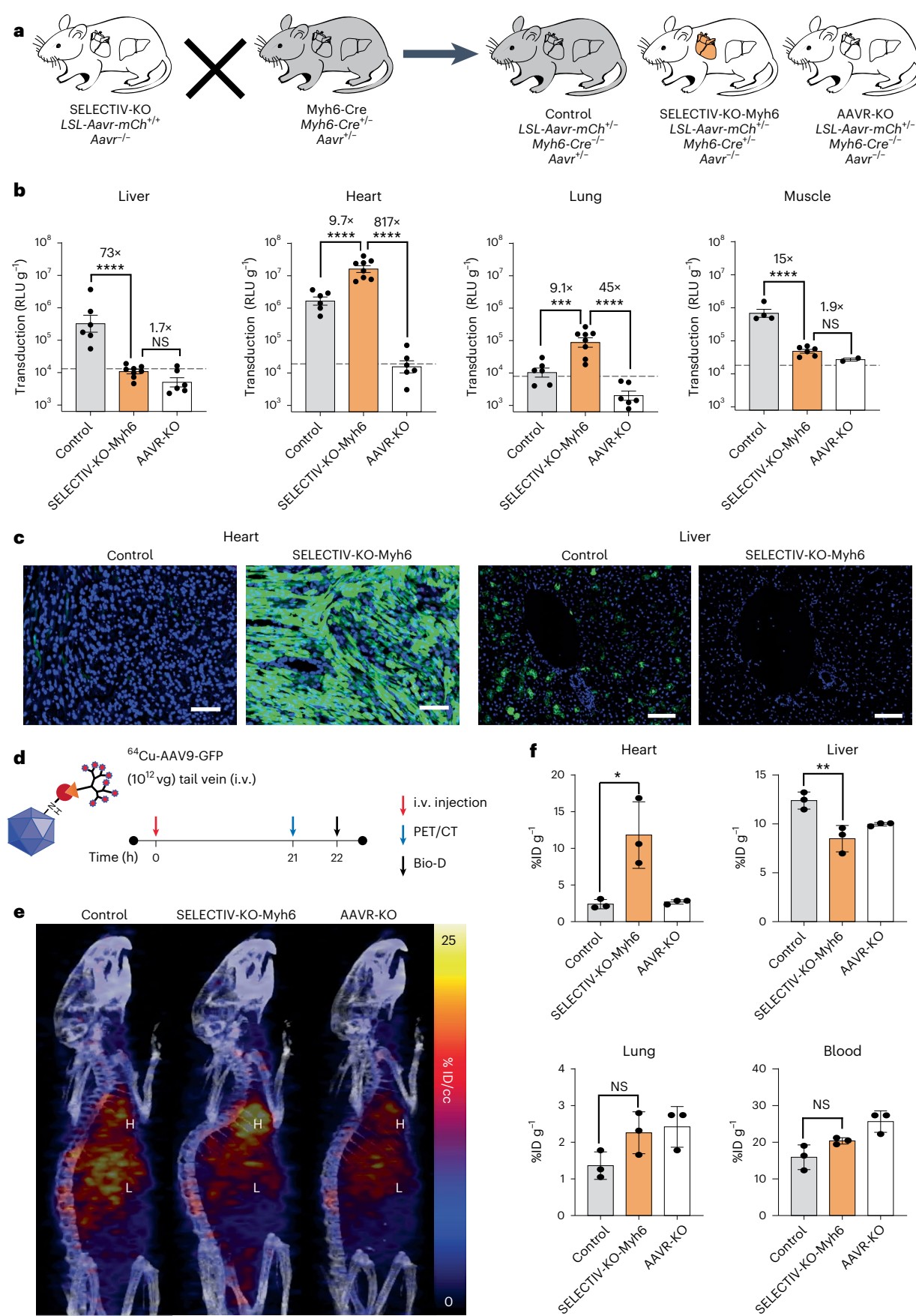

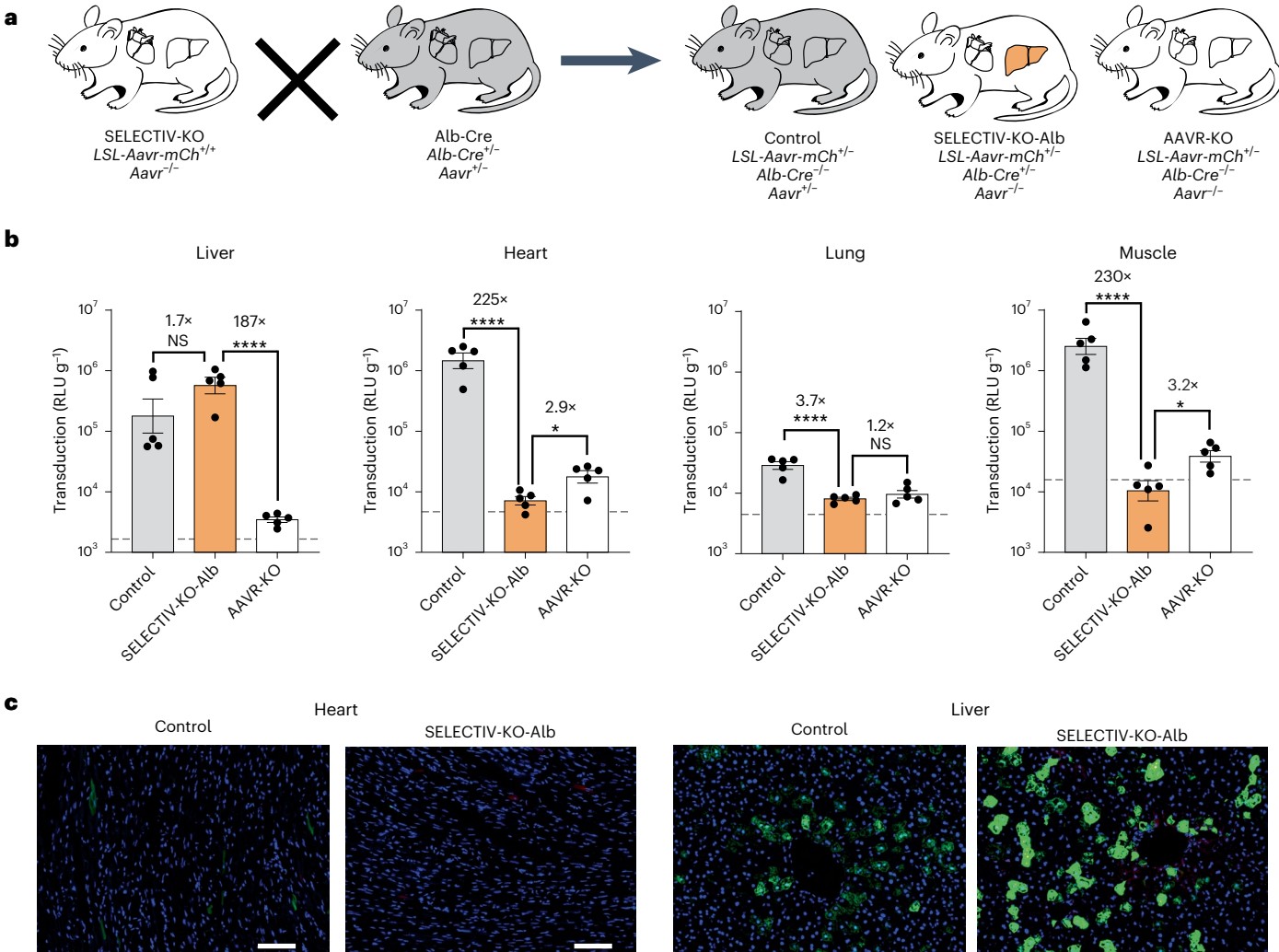

**Fig. 5 | The transduction specificity of the SELECTIV-KO system is tightly controlled by the choice of Cre mouse line. a**, SELECTIV-KO mice were bred with Alb-Cre mice on an *Aavr*^KO background to generate mice with selective AAVR overexpression in liver hepatocytes and KO of *Aavr* in nontarget tissue (SELECTIV-KO-Alb). Littermate controls heterozygous (control) or homozygous for the endogenous *Aavr*-KO allele (AAVR-KO) were also produced during breeding. **b**, Control (*n* = 5 mice), SELECTIV-KO-Alb (liver-specific) (*n* = 5 mice) and AAVR-KO (*n* = 5 mice) mice were injected with 3 × 10^10 vg AAV9-luciferase by systemic (i.v.) injection (both male and female, 6–8 weeks of age). Transduction was tracked over time by in vivo imaging and assessed ex vivo. Transduction was measured ex vivo for the liver, heart, lung and muscle, which demonstrated specific transduction of the liver in the SELECTIV-KO-Alb mice, with detargeting

of the other organs and tissue. Mean value and s.e.m. are shown. The horizontal dashed line indicates background luminescence in lysates of a mouse not transduced by AAV9-luciferase. Fold-changes are indicated, and the *P* value was calculated using an ordinary one-way ANOVA with Holm–Šídák's multiple comparisons post-test, with ****$P < 0.0001$, *$P < 0.05$ (control versus SELECTIV-KO-Alb: liver $P = 0.075$, heart $P = 6.5 × 10^{-9}$, lung $P = 3.2 × 10^{-5}$, muscle $P = 9.4 × 10^{-8}$; SELECTIV-KO-Alb versus AAVR-KO: liver $P = 5.5 × 10^{-6}$, heart $P = 0.020$, lung $P = 0.34$, muscle $P = 0.012$). **c**, Control and SELECTIV-KO-Alb (liver-specific) mice were injected with 2 × 10^11 vg PHP.eB-GFP by systemic (i.v.) injection. Mice were euthanized at 28 d post-transduction and liver and heart were removed to determine transduction by fluorescence microscopy. Scale bars, 200 µm. Similar results were seen in three biological replicates.

To demonstrate the versatility of the SELECTIV-KO platform, we expanded the analysis to liver-specific expression by using the Alb-Cre mouse line, which expresses Cre in liver hepatocytes[48]. We generated SELECTIV-KO-Alb mice which, along with control and AAVR-KO mice, were transduced with AAV9-luc (Fig. 5a). The luciferase signal in the abdomen of the SELECTIV-KO-Alb mice was more concentrated compared with the broader pattern in control mice (Extended Data Fig. 8a). At day 21 post-transduction, organs were evaluated for transduction efficiency by measuring luciferase activity in lysates (Fig. 5b). As in control mice, liver lysates displayed robust luciferase activity in SELECTIV-KO-Alb mice. In other tissues of SELECTIV-KO-Alb mice, transduction was reduced to levels similar to those seen in AAVR-KO mice, with significant decreases in the heart (225-fold), lung (3.7-fold) and muscle (230-fold). Mice were also transduced with AAV-PHP.

eB-GFP to evaluate transduction efficiency at a cellular level. At 28 d post-injection, transduction was determined by GFP levels in the heart and liver (Fig. 5c and Extended Data Fig. 8b). The liver and heart of the control mice were transduced, while only the liver of the SELECTIV-KO-Alb mice was transduced. Together, these results provide strong evidence that the SELECTIV-KO platform allows for robust transduction in target tissues while preventing transduction in nontarget tissues, with the specificity dictated by the chosen Cre-driver mouse line.

### Targeted neuronal transduction in SELECTIV-KO mice
Mice are a powerful model to study neural function and neurological disorders, which often requires transgene expression in defined neuronal subsets[1–3,49–51]. Recently, AAV variants have been developed for widespread gene transfer to the adult rodent brain upon noninvasive

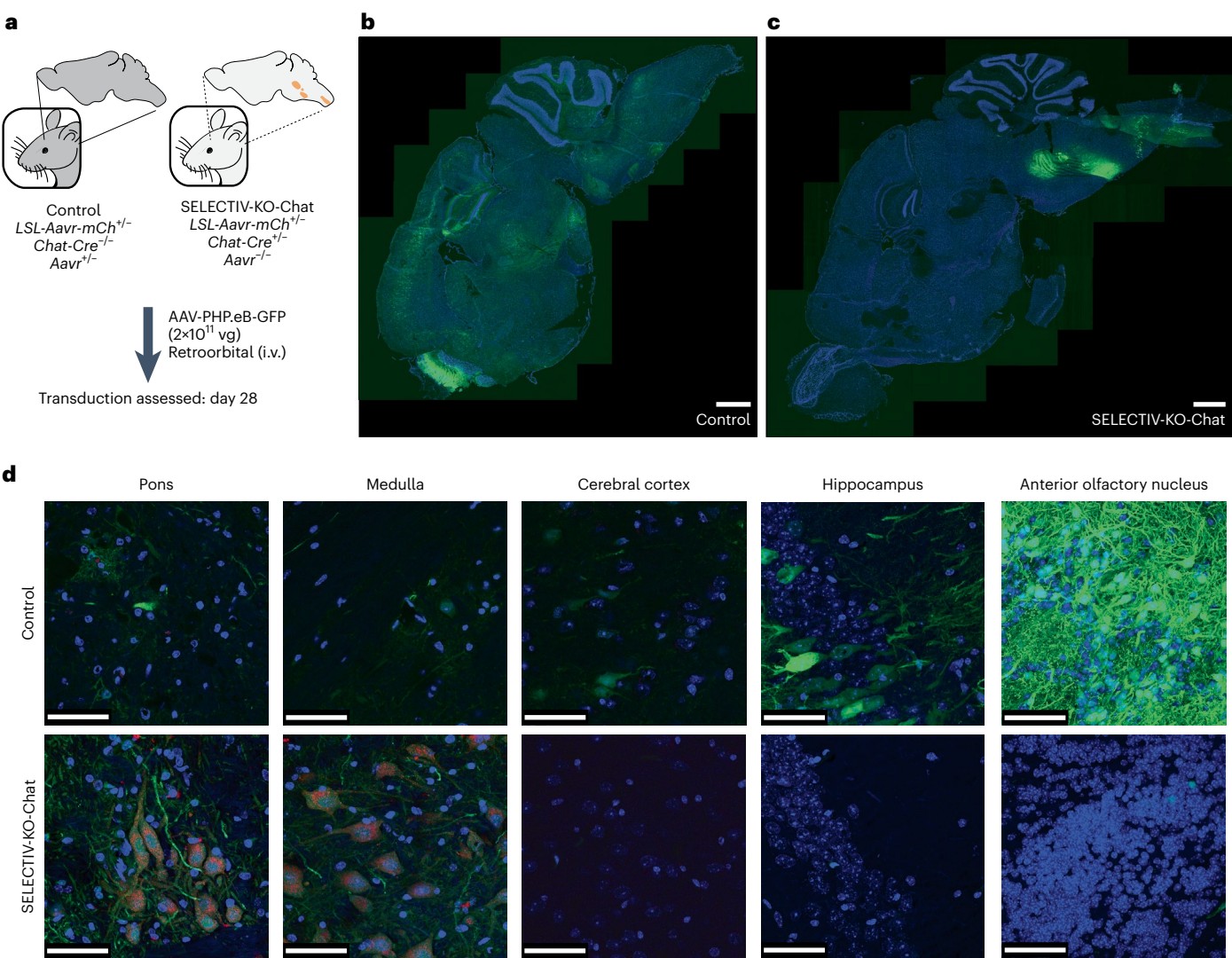

**Fig. 6 | Selective AAVR expression in a neuronal subpopulation targets AAV transduction to specific regions in the brain upon systemic delivery.**
**a**, Schematic of the experiment. Mice were injected with AAV-PHP.eB-GFP, a capsid variant that can enter the brain after systemic injection. Brains were extracted after 28 d, and sections were cut on the sagittal plain and nuclear staining was performed. Similar results were seen in two biological replicates; data for male mice are shown, at 6–8 weeks of age. **b**, Fluorescence microscopy in brain slices of a control mouse. Widespread transduction to different areas in the brain slice was detected (GFP; green). **c**, Fluorescence microscopy in brain slices of a SELECTIV-KO-Chat mice, which was bred to express AAVR in cholinergic neurons with concomitant KO in all other cell types. Transduction was observed in localized areas, which correspond to areas enriched in cholinergic neurons (GFP; green). **d**, Higher-magnification images of the brain slices. The AAVR-mCh expression (red) can be seen in the areas with a high density of cholinergic neurons such as the pons and medulla in the SELECTIV-KO-Chat mice. This correlates with high levels of transduction in these areas, while only moderate transduction was seen in control mice. In areas with low levels of Chat-positive cholinergic neurons, such as cerebral cortex, hippocampus and anterior olfactory nucleus, control mice have clear transduction, while SELECTIV-KO-Chat mice show little to no transduction. Scale bars, 1 mm (**b**,**c**) and 50 μm (**d**).

systemic administration, including AAV-PHP.eB, which more efficiently crosses the blood–brain barrier. To evaluate how SELECTIV-KO mice would perform in directing AAV transduction to specific cell types within the central nervous system, we used Chat-Cre mice which express Cre recombinase exclusively in cholinergic neurons[52]. SELECTIV-KO-Chat and control mice were administered AAV-PHP.eB encoding GFP by intravenous injection, and brains were analyzed 28 d post-transduction (Fig. 6a). As reported previously[43], AAV-PHP.eB efficiently transduces many regions of the adult mouse brain, driving broad transgene expression (Fig. 6b). However, in SELECTIV-KO-Chat mice, GFP expression was much more restricted, with GFP-positive cells concentrated in specific areas of the medulla and pons in line with the known expression pattern of Chat-Cre (Fig. 6c and Extended Data Fig. 9a)[53,54].

Selected areas of interest were imaged at a higher magnification (Fig. 6d and Extended Data Fig. 9b). In the pons and medulla, control mice have some transduction, while SELECTIV-KO-Chat mice have high levels of transduction. At this magnification, AAVR overexpression (as determined by mCherry fluorescence) was detected in areas of high AAV transduction. Transduction in SELECTIV-KO-Chat mice was highly specific with no GFP expression in nontarget cells, which were efficiently transduced in control mice, including in the brain (cerebral cortex, hippocampus and anterior olfactory nucleus) (Fig. 6d and Extended Data Fig. 9b) and liver (Extended Data Fig. 9c). These results demonstrate the utility of the SELECTIV-KO platform to target AAV transduction to specific neurons in the brain after systemic injection. These experiments can be performed using readily available AAV vectors without specialized equipment or techniques, and without

invasive procedures. This highly tractable system will allow for faster production of model systems using currently available mice and AAV vectors, increasing the speed of research into important areas of brain research and testing the effect of transgene expression in the brain.

## Discussion

We have developed the SELECTIV mouse system as a versatile platform for efficient and targeted transgene expression in mice without the need for production of de novo transgenic lines. The specificity of transduction is achieved by regulated AAVR overexpression and inactivation of endogenous *Aavr*. This strategy leads to a near complete loss of transduction of nontarget tissues, while allowing for efficient transgene expression in target cells. We demonstrate the use of this system with five Cre mouse lines, and the SELECTIV platform can be readily tailored for specific AAV targeting to tissues of interest due to the availability of numerous Cre mouse lines driving specific expression in a wide variety of cell types[43].

Once the SELECTIV-KO mice are established for a certain tissue-specific promoter, transgene expression is mediated using mouse extrinsic AAV vectors. This offers advantages over traditional transgenic models, including increased flexibility to modify genes of interest, use of AAV to specifically express genes late in life and the ability to deliver multiple genes in the same mouse. Many of these are hard or impossible using traditional transgenic approaches. The addition of spCas9 in the SELECTIV expression cassette allows for single-gene KOs or library-based in vivo CRISPR screens. The SELECTIV platform will be useful for basic research and in biomedical research due to the increased flexibility of the system compared with traditional methods.

This research gave insight into the in vivo role of AAVR. Whether AAVR predominantly acts to facilitate endocytosis or has a role post-entry to facilitate intracellular trafficking is currently unclear. Antibodies against AAVR in cell culture medium block AAV transduction, suggesting a direct role of AAVR in uptake of AAV particles[54], although glycans at the cell surface such as heparan sulphate and sialic acid play a role for some serotypes[55]. We show a critical role of AAVR in efficient in vitro transduction in both MEFs and MuSCs, with greater increases seen for AAV vectors without clearly defined attachment factors (AAV8 and AAV9). The role of AAVR in vivo was subsequently assessed. We found that selective expression of AAVR in the heart resulted in increased uptake in heart tissue, while liver uptake was decreased. These results suggest that AAVR can act as a cell surface receptor for AAV9, controlling uptake in organs.

Despite advances in developing clinically desirable AAV capsids, the cellular factors that determine the efficiency of in vivo AAV transduction in different cell types are ill-defined[56,57]. We found that AAVR expression is rate-limiting for AAV transduction in vivo for multiple cell types. MuSCs, which are refractory to AAV transduction, express low levels of endogenous *Aavr*. Targeting muscle with AAV gene therapy is relevant for the treatment of inherited disorders of progressive muscular weakness, including Duchenne muscular dystrophy and spinal muscular atrophy. AAV9-based vectors have been used to produce a product for the treatment of spinal muscular atrophy[19] and preclinical data in mouse and dog models of Duchenne muscular dystrophy show promise. However, concerns remain about the longevity of the transgene expression in these degenerating muscle disorders due to the increased muscle turnover[26,57]. AAV capsid variants that better target muscle upon systemic delivery have been developed[26,58]. These variants retained their dependence on AAVR[58]. We have shown that AAVR overexpression in mice increased transduction in MuSCs, suggesting that strategies to modify the AAV–AAVR interaction could be beneficial to further increase efficiency. Importantly, transgene expression in AAVR-overexpressing muscle is maintained after injury, suggesting that transduced MuSCs could act as a dormant pool to sustain transgene expression and counteract loss with tissue turnover.

We similarly found that AAVR overexpression enhanced photoreceptor layer transduction in the eye after intravitreal injection. This suggests that AAVR expression could be rate-limiting for transduction, in addition to the proposed anatomical barriers[59]. Although not directly translatable for gene therapy, this finding creates new opportunities in biomedical research in preclinical models. Our system demonstrated photoreceptor targeting using less invasive intravitreal injection, compared with subretinal injections previously required. Another group recently identified AAV capsid variants that can target photoreceptor cells after intravitreal injection in WT mice[60]. It is also possible that these systems would be complementary if used together, by imposing selectivity using the SELECTIV-KO system in conjunction with a Cre line specific to retinal cell types of interest[61,62]. This could allow for specific photoreceptor or retinal pigment epithelium transduction after intravitreal delivery with the newly discovered vectors or systemic delivery using AAV-PHP.eB[63]. We believe that the SELECTIV system has great potential to work in complementary fashion with novel capsids to develop unique transduction strategies to enhance biomedical research in mice.

The SELECTIV system showed enhanced transduction in multiple tissues with local and systemic administration of AAV. This was observed in cell types refractory to transduction such as MuSCs[1–3,49–51,63], as well as in readily transducible tissues such as the heart. Using common AAV vectors, we showed a strong increase (9.7-fold) in heart transduction in the SELECTIV-KO-MyH6 compared with control mice, while transduction in the liver and muscle was absent. The SELECTIV model system will expand the cell types targetable by AAV, while increasing efficiency of transduction. This allows for lower AAV dosages, reducing cost and potential AAV-vector-mediated toxicity[64].

The increased specificity and efficiency of the SELECTIV system may be useful for testing of preclinical AAV-delivered transgenes. While enforcing tissue-specificity by overexpressing the receptor does not provide information regarding tropism of an AAV capsid variant, it does allow for testing a 'best case scenario' for cell-type-specific transgene delivery in a mouse model. This includes identifying functional differences in types of cells targeted or testing of multiple transgenes to identify the best construct during early development. This can be done in parallel while researching capsid variants and promoters, speeding up specific aspects of preclinical research.

The SELECTIV-KO system allows for precise cell-type specificity, which is especially useful in neuroscience research. Experimental approaches in rodents that study brain circuitry, such as optogenetics or neurodegenerative diseases, often require expression in specific neuronal populations[1–3,49–51,65]. Previous strategies to achieve cell-type specificity include the development of AAV vectors that contain promoters and enhancers of endogenous gene products[66]. However, this strategy is confounded by the limited packaging capacity of AAV, requiring the identification of 'mini-promotors' that still retain cell-type specificity, which is often problematic[67]. Another powerful system is the DIO/FLEx system, which has been developed to mediate Cre-specific transgene expression[68,69]. This system uses AAV vectors containing DNA elements that drive Cre-mediated inversion of the transgene coding sequence. While this system has been used extensively, there can be nonspecific transgene expression due to recombination during vector production, which requires optimization[70]. The SELECTIV-KO system enhances transduction efficiency and specificity by regulating AAVR and therefore does not require specialized AAV vectors.

SELECTIV provides an alternative approach. Some limitations are the extra breeding requirements and the modulation of a host gene, AAVR, which may affect host physiology. The physiological role of AAVR is unknown, but KO mice are viable, are born with expected Mendelian ratios, have no obvious abnormalities in neuronal migration or cortical anatomy, and have a subtle defect in auditory processing[71]. Male *Aavr*[−/−] males are infertile and present a globozoospermia-like phenotype[72]. Despite these limitations, it has several key advantages.

First, the selectivity is obtained through the use of readily available tissue-specific Cre lines. This unlocks tissue specificities not accessible through the use of mini-promoters. Second, overexpression of the receptor enhances the transduction of target cells, including cells that are refractory to AAV transduction, such as MuSCs, which would not be targetable using other systems. Third, uptake of AAV vector particles is reduced in nontarget tissue, reducing off-target transduction and toxicities. Fourth, multiple genes can be delivered at the same time through co-administration of AAVs encoding different genes. Fifth, because AAVR facilitates AAV entry of most natural and engineered AAVs, this strategy is compatible with a wide variety of pre-existing and novel AAV vectors. Moreover, because it does not rely on specific DNA elements in the AAV vector to gain its specificity, it allows further flexibility in choice of promoter. While many cell types can be defined by a single marker, there are cell subtypes that are best defined by multiple markers[73]. We envision that SELECTIV can be paired with AAV capsid variants with novel specificity or AAV vectors containing tissue-specific mini-promoters to target cell subsets characterized by expression of two or more marker genes. The SELECTIV platform has broad utility in biomedical research requiring transgene expression or CRISPR KO in mouse models, and its flexibility allows use with existing and new AAV-based expression systems.

## Online content

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

## Methods

### SELECTIV construct cloning

The plasmid backbone used to generate the recombination template to produce the SELECTIV mice was generated from PhiC31 integrase-mediated cassette exchange (pBT378, Addgene, cat. no. 52554), which was previously modified to express *Streptococcus pyogenes* Cas9 (*spCas9*) to generate paatB-LSL-spCas9. Mouse AAVR (*mAAVR*) and *mCherry* were amplified by PCR to add overlapping primers for Gibson assembly.

The paatB-LSL-spCas9 vector was linearized with NheI and PmeI (NEB). Mouse *Aavr* was amplified with F primer (TCGCGCTCACTGGC CGTCGTTTTACAAGTTTCACCATGGAGAAGAGACTGG) and R primer (CTCCTCGCCCTTGCTCACCATCAAGATCTCCTCCCGTGCGC). A gene encoding mCherry was amplified with F primer (AAGTCCAGGAGCG CACGGGAGGAGATCTTGATGGTGAGCAAGGGCGA) and R primer (AGGTCCAGGGTTGGACTCCACGTCTCCCGCCAACTTGAGAAGGT CAAAATTCAAAGTCTGTTTCACTCCG CTTCCCTTGTACAGCTCGTC CATGC). The mCherry construct was amplified to create overlap with Cas9-Flag from the vector using F primer (GCATGGACGAGCTGTACAA GGGAAGCGGAGTGAAACAGAC) and R primer (CGTGGTCCTTATAGTC CATGGTGGCACCGGTCGTTTGGGGAGGTCCAGGGTTGGACTC). All pieces were assembled using Gibson Assembly Master Mix (NEB, E2611) and clones were confirmed by sequencing. The plasmid is available through Addgene (cat. no. 82743).

### Mouse lines

SELECTIV mice were generated at the Stanford Transgenic, Knockout, and Tumor Model Center using Integrase Mediated Transgenesis[74] and using the construct described above and PhiC31 integrase. The construct was inserted into the *H11* locus using C57BL/6 mice with three attP sites previously knocked-in to the *H11* locus.

Mouse lines purchased from The Jackson Laboratory include C57BL/6J (664, Jax), E2A-Cre (B6.FVB-Tg(EIIa-cre)C5379Lmgd/J, 3724, Jax), Chat-Cre (ChAT-IRES-Cre, 6410, Jax), Alb-Cre (B6.Cg-Speer6-ps1Tg(Alb-cre)21Mgn/J, 3574, Jax), Myh6-Cre (B6.FVB-Tg(Myh6-cre)2182Mds/J HEMI, 11038, Jax) and Pax7-Cre (B6.Cg-Pax7tm1(cre/ERT2)Gaka/J, 17763, Jax). FVB AAVR-KO mice were previously generated using transcription activator-like effector nuclease-mediated gene targeting to generate a 1-base pair deletion in *Au040320*, the gene encoding AAVR[71]. The AAVR-KO allele was introduced into C57BL/6J by backcrossing for ten generations. As reported elsewhere[72], homozygous *Aavr*-KO male mice were sterile on the C57BL/6J background, which requires all crosses to use heterozygous males. Genotyping was performed by Transnetyx using real-time PCR. Mice have been deposited in The Jackson Laboratory Repository with the SELECTIV mice (JAX Stock No. 037553, C57BL/6J-*Igs2*<sup>tm1(CAG-AU040320/mCherry,-cas9*)Janc</sup>/J) and *Aavr*-KO mice backcrossed to C57BL/6 (JAX Stock No. 037596, B6.FVB-*AU040320*<sup>em1Janc</sup>/J). A mix of female and male mice were used for all experiments as available after genotyping from each litter.

Mice were housed in the Association for Assessment and Accreditation of Laboratory Animal Care-accredited Stanford mouse barrier facility (Protocol no. 28856). Husbandry was performed in accordance with the Guide for the Care and Use of Laboratory Animals, 8th edition[75], and the Public Health Service Policy on Humane Care and Use of Laboratory Animals (2015). Room conditions included a temperature of 23 °C, relative humidity of 30–40% and a 12/12-h light/dark cycle (lights on at 7:00). Mice were maintained under specific-pathogen-free conditions in irradiated, disposable, individually ventilated cages (Innocage<sup>R</sup>, Innovive), with irradiated corncob bedding or AlphaDri, irradiated food (Teklad 2918 Global 18% Protein Rodent Diet, Envigo) and ultraviolet-irradiated, acidified (pH 2.5–3.0), reverse-osmosis-purified bottled water (Aquavive<sup>R</sup>, Innovive). The mouse colonies were monitored for adventitious viral, bacterial and parasitic pathogens by dirty-bedding sentinels. Sentinels are tested every 4 months and were found to be free of mouse parvovirus, minute virus of mice, mouse hepatitis virus, mouse rotavirus, Theiler's murine encephalomyelitis virus, murine norovirus, Sendai virus, mouse adenovirus 1 and 2, ectromelia virus, lymphocytic choriomeningitis virus, pneumonia virus of mice, respiratory enterovirus III, *Mycoplasma pulmonis*, *Helicobacter* spp., *Rodentibacter pneumotropica*, fur mites, lice and pinworms.

### MEFs

Control (*Aavr-mCh−/−*) and SELECTIV-WB (*Aavr-mCh+/−*) mice were mated, and between days 9 and 11 post-breeding, the dam was euthanized and embryos were removed to generate MEFs using a method adapted from a standard protocol. In short, embryos were isolated into individual 6-cm dishes (ThermoFisher, 150326) in PBS (Fisher Scientific, MT21030CV) and head, heart and liver were removed. PBS was removed and ~0.5 ml of trypsin-EDTA (ThermoFisher, 25300120) was added to each dish. Embryos were minced using sterile razor blades and incubated at 37 °C for 5–10 min. Cells were resuspended in 5 ml of DMEM + 10% FBS + penicillin/streptomycin (D10) (ThermoFisher, 11995073; Sigma, 4333) with a pipette and left at 37 °C overnight. The next day, media and unattached cells were removed, and cells were split into 10-cm dishes. Cells were passed in D10 medium every 3–4 d. MEFs were kept as individual lines per embryo. Genotyping of lines was performed by Transnetyx to determine if each line was WT or AAVR-overexpressing (SELECTIV-WB).

MEFs used for transduction were plated in 96-well plates (Thomas Scientific, EK-25180) in D10 medium at $10^4$ cells per well. The next day, cells were transduced with AAV2, AAV4, AAV8 and AAV9 encoding luciferase. At 72 h post-transduction, cells were lysed and luciferase activity was assessed using the Luciferase Assay System (Promega, E1500).

### Western blot of AAVR

Organs were collected from WT, AAVR-KO and SELECTIV-WB mice after euthanasia. Organs were placed into gentleMACS M Tubes (Miltenyi Biotec, 130-093-236) with 3–5 ml of DMEM and lysates were made using gentleMACS Octo Dissociator (Miltenyi Biotec, 130-095-937). Laemmli sample buffer (Bio-Rad, 1610737) supplemented with 2-Mercaptoethanol (Bio-Rad, 1610710) was added to the lysates. Samples were heated at 95 °C for 10 min and cooled on ice before analysis by SDS–PAGE. Samples were run on 4–15% Mini-PROTEAN TGX Precast Gels (Bio-Rad, 4561086) and transferred using the Trans-Blot Turbo PVDF transfer kit (Bio-Rad, 1704272). Western blotting was performed using the primary antibody rabbit anti-AAVR at a 1:1,000 dilution (Proteintech, 21016-1-AP) and secondary antibody anti-rabbit-HRP at a 1:4,000 dilution (Genetex, GTX213110-01). GAPDH was detected on the same blots using mouse anti-GAPDH-HRP at a 1:4,000 dilution (Genetex, GTX627408-01). Blots were developed using SuperSignal West Femto Maximum Sensitivity Substrate (Thermo Scientific, 34096) or Supersignal West Dura Extended Duration Substrate (Thermo Scientific, 34076). Imaging was performed on a ChemiDoc Imaging System (Bio-Rad) and density was analyzed using Image Lab Software (Bio-Rad).

### AAVs encoding reporter genes

Purified, titered stocks of recombinant AAV2, AAV4, AAV8 and AAV9 vectors encoding reporter genes under the CMV promoter were purchased from the University of North Carolina Chapel Hill Gene Therapy Center Vector Core. AAV vectors encoding EGFP were self-complementary and those encoding firefly luciferase single-stranded. Purified titered stocks of AAV-PHP.eB were purchased through Addgene (37825-PHPeB), and this is a single-stranded AAV genome with CAG-driven GFP expression.

### Genome editing in MEFs

AAV9 was used to deliver PCSK9 sgRNA (gMH, CAGGTTCCATGGGAT GCTCT) based on previous literature[76]. The guide was cloned into

pAAV-U6-sgRNA-CMV-GFP and purified, titered stocks of AAV9 were generated at the University of North Carolina Chapel Hill Gene Therapy Center Vector Core. SELECTIV-WB MEFs were transduced with this vector at multiplicity of infection (MOI) = 10,000 viral genomes (vg) per cell or MOI = 50,000 vg per cell or were mock transduced. At 3 d post-transduction, FACS was used to sort the GFP-positive/transduced cell population. The gating strategy for FACS is outlined in Supplementary Fig. 1. DNA was collected from these cells and the target region in PCSK9 was amplified by PCR (CCTTACCAGG GGAGCGGTC/CTATTAGCTGAAGGGCTTTTGAAGC). Amplified DNA was sent to the Massachusetts General Hospital Center for Computational and Integrative Biology for CRISPR amplicon sequencing and indel frequency was calculated.

### AAV intramuscular injection in SELECTIV-WB

Control and SELECTIV-WB mice (6–10 weeks old, male and female, $n = 4$ per group) were injected in the tibialis anterior muscle with AAV2-luc ($10^{11}$ vg) while under general anesthesia. Luciferase activity was tracked over time using the Lago in vivo imaging system (Spectral Instruments Imaging) at the Stanford Center for Innovation in *In vivo* Imaging (SCI[3]). Mice were injected with ~20 mg kg$^{-1}$ luciferin in PBS by retroorbital injection and imaged within 15 min of injection. Repeated imaging was performed at days 3, 7, 14, 21 and 28 post-transduction. At day 30 post-transduction, muscle injury and repair by MuSCs was induced by injecting 50 µl of BaCl$_2$ (1.2% w/v in saline) intramuscularly into the tibialis anterior muscle. Luciferase activity was assessed at days 33, 35, 39, 46, 53, 60 and 120 post-transduction.

### MuSC isolation and myoblast culture

Flow cytometry and sorting strategies for MuSCs were performed as previously described[34,77,78]. MuSCs were enriched by FACS using a modified dissociation protocol. Mouse hindlimb muscles were carefully dissected to remove adipose, tendon and nerves, and then minced using scissors until homogenous. Minced tissue was suspended in 10 ml of collagenase solution (700 U ml$^{-1}$ collagenase II (Worthington), 0.2% BSA (Sigma) in Ham's F10 media (Gibco)) and digested at 37 °C for 1 h using the gentleMACS Octo Dissociator. Digested tissues were diluted with 20 ml of 0.2% BSA in Ham's F10 media (Gibco) and pelleted at 500$g$ for 10 min. Supernatant was removed, leaving 8 ml of solution and the cell pellet. Cell pellets were resuspended and 1,000 U of collagenase II and 11 U of dispase I (Life Technologies) in 2 ml of PBS were added, and then further digested for 30 min on the gentleMACS Dissociator using a custom program. Cells were pelleted, washed in FACS buffer (0.5% BSA, 1 mM EDTA in PBS) and filtered through a 40 µm nylon cell filter. Red blood cells were lysed using RBC Lysis Buffer (eBioscience). Single-cell suspensions were incubated with direct APC-Cy7-conjugated antibodies against CD11b (M1/70), CD45 (30-F11), Sca1 (D7) and CD31 (390) (Biolegend, 3 µl of each per mouse); PE-a7-integrin (2 µl per mouse, AbLab); and APC-CD34 (RAM34) (Biolegend, 3 µl per mouse). Cell suspensions were then subjected to FACS for MuSC isolation (CD45$^-$CD11b$^-$CD31$^-$Sca1$^-$α7-integrin$^+$CD34$^+$ cells) or flow cytometry quantification for GFP$^+$ MuSCs (CD45$^-$CD11b$^-$CD31$^-$Sca1$^-$α7-integrin$^+$). Based on previous data, this strategy yields MuSCs at >95% purity based on Pax7$^+$ staining. Antibody gates and compensation were established using unstained, single-stained and Fluorescence Minus One controls. GFP gates were established on untransduced cells or tissues. Sorting was carried out on Sony SH800S or BD LSR II. UV analyzers and analysis was performed with FlowJo and Sony Cell Sorter software. The gating strategy for FACS is outlined in Supplementary Fig. 2.

Primary murine MuSC-derived myoblasts were cultured on collagen-coated plates (0.01% w/v Collagen type I solution from rat tail, Sigma) in growth medium (20% FBS in Ham's F10 medium supplemented with bFGF; Peprotech). Fluorescence imaging of AAV2 and AAV8 transduced cells was carried out using the Revolve microscope (Echo).

### Radiolabeling of AAV9

All radiolabeling experiments were conducted under a Controlled Radiation Authorization approved by Stanford University. scAAV9-CMV-eGFP was purchased from University of North Carolina Chapel Hill Gene Therapy Center Vector Core. Radiolabeling of AAV9 followed a previously reported method[47]. In brief, AAV9 ($7.5 \times 10^{12}$ vg, 13 pmol) in 1 × PBS (0.2 ml, 0.001% PF-68), adjusted to pH 8 with 0.1 M Na$_2$CO$_3$ solution (pH 9.2), was mixed with 2 mM tetrazine-PEG5-NHS (2 nmol, 1 µl) in anhydrous dimethylsulfoxide. The reaction mixtures were incubated for 30 min at 25 °C, quenched by adding 1 × PBS (0.1 ml, 0.001% PF-68) and transferred to a mini-dialysis device (20 kDa molecular weight cut-off) for overnight dialysis in 1 × PBS (two times, each with 0.5 l, 0.001% PF-68). After the recovery of tetrazine-conjugated AAVs from dialysis, these AAVs were reacted for 30 min with (NOTA)$_8$-TCO fully incorporating $^{64}$Cu, which was freshly prepared from the reaction of $^{64}$CuCl$_2$ (137 MBq, 3.7 mCi) and 10 µM (NOTA)$_8$-TCO (50 pmol, 5 µl) in ammonium citrate buffer (20 µl, pH 6.5). The incorporation of $^{64}$Cu to (NOTA)$_8$-TCO was monitored by instant thin-layer chromatography and completed in 30 min. Radiolabeled $^{64}$Cu-AAV9 was washed by a centrifugal filter unit (100 kDa molecular weight cut-off) under 2,500$g$ for 10 min) with three cycles of PBS (15 ml). Concentrated $^{64}$Cu-AAV9 in ~200-µl volume was recovered and reconstituted in saline for the animal study.

### PET/computerized tomography (CT) scans and biodistribution

All animal experiments were conducted under the approved Administrative Panel on Laboratory Animal Care protocol at Stanford University. Control, AAVR-KO and SELECTIV-KO-Alb mice (male and female, 8–10 weeks old, $n = 3$) were anesthetized with 3.0% isoflurane in oxygen and maintained under 1.5–2.0% isoflurane while scanning. $^{64}$Cu-AAV9 ($n = 9$, 201 ± 10 kBq) was administered via tail vein injection to mice on a small animal PET/CT scanner (Siemens). Mice were scanned for 30 min at 21 h post-injection. After the final PET/CT scan at 21 h, mice were euthanized by Euthasol under deep isoflurane and perfused with DMEM and PBS solution. Blood, heart, lungs and liver were collected and radioactivity in each organ was measured with a gamma counter. The biodistribution of $^{64}$Cu-AAV9 was quantified as %ID g$^{-1}$ after decay correction.

### Ex vivo luciferase

Animals were euthanized and liver, heart, lung and tibialis anterior muscle were collected into gentleMACS M Tubes (Miltenyi Biotec, 130-093-236) in 3–6 ml of DMEM, and lysates were made using a gentleMACS Octo Dissociator (Miltenyi Biotec, 130-095-937). Tissue weights were calculated to normalize luciferase activity by tissue weight. Luciferase activity was assessed using the Luciferase Assay System (Promega, E1500).

### In vivo luciferase

Luciferase activity was tracked over time using the Lago in vivo imaging system (Spectral Instruments Imaging) at SCI[3]. Mice were injected with ~20 mg kg$^{-1}$ D-luciferin Firefly (Biosynth, L-8220) in PBS by retroorbital injection and imaged within 15 min of injection.

### Organ collection for GFP

Mice were euthanized and perfused with PBS and freshly prepared 4% PFA (Sigma-Aldrich, P6148) before organ collection. Liver and heart were collected and placed into 4% PFA in PBS for 72 h before embedding. Brain was equilibrated with 15% sucrose + 4% PFA + PBS followed by 30% sucrose + 4% PFA + PBS solution before embedding. Samples were sent to Stanford Animal Histology Services for embedding in OCT media, sectioning at 5–10 µm and mounting on slides. Tissue was stained with Hoechst 33342, Trihydrochloride, Trihydrate (Fisher Scientific, H3570) at 1 µg ml$^{-1}$ for 30 min. Coverslips

were mounted using Prolong Glass Antifade Mounting Media (Invitrogen, P36980).

Slides were imaged on a Leica DM4 Upright Microscope (Canary Center at Stanford). AAV transduction was measured by EGFP fluorescence and AAVR-mCh was measured by mCherry expression, and Hoechst staining for the nucleus.

### Intravitreal injection and retinal transduction

Mice were anesthetized with ketamine and xylazine based on their weight (0.08 mg of ketamine per gram + 0.01 mg of xylazine per gram). For each AAV intravitreal injection, an aperture was made by micropipette around the retinal periphery of 5-month-old mice and advanced into the vitreous chamber to avoid lens damage. Approximately $2 \times 10^9$ vg of AAV2 (scAAV-CMV-EGFP) in 1–1.5 µl was injected intravitreally into each eye of mice.

AAV transduction was tracked over time using in vivo imaging of the mouse retina at days 7, 14 and 21. Pupils were dilated with 1% tropicamide (Akorn). Mouse corneas were covered by a customized contact lens (3.0-mm diameter, 1.6 mm base curve, poly(methyl methacrylate) clear, Advanced Vision Technologies) before imaging. An 870-nm infrared light source and a 30° lens were used on the Heidelberg Spectralis SLO/OCT system (Heidelberg Engineering) for autofluorescent/infrared fundus imaging. We obtained autofluorescent images at the wavelengths of 488-nm absorption and 495-nm emission using a 55° lens. Images of the central retina were taken, with the optic nerve positioned centrally. Mice were euthanized at day 21 and retinas were collected for imaging. Mouse eyes were enucleated and dissected from tendons and extraocular muscles after being perfused by 4% PFA in PBS. Afterwards, the cornea, lens and vitreous were removed. For wholemounts, retinas were dissected carefully from eye cups and radial cuts were made to flatmount on coverslips.

The retinas were incubated in PBS with 0.3% Triton X-100, 5% BSA and 5% normal goat serum (blocking buffer) for 2 h at room temperature. Retinal cups were incubated in primary antibody (Anti-RBPMS; Guinea pig; PhosphoSolutions no. 1832) diluted 1:2,000 in the blocking buffer overnight at 4 °C, rinsed three times in PBS for 10 min each time and stained with secondary antibody solution containing goat anti-Guinea pig Alexa Fluor 647 (Invitrogen) at 1:200 for 2 h at room temperature. In addition, the eye cups were rinsed in PBS three times, each for 20 min. Images were taken with a Zeiss LSM 880 confocal microscope after mounting in Fluoromount-G (SouthernBiotech).

For GFP-positive cell counting, three fields sampled from each retina flatmount stained with RBPMS were picked from peripheral regions (500 µm from edge of retinas, three different quadrants, z-stack) using a ×63 lens with a Zeiss confocal microscope in a masked manner. For the RGC layer, only both RBPMS- and GFP-positive cells were counted. For amacrine cells in the INL, only GFP-positive cells from first layer of the INL (close to the RGC layer) were counted. For the photoreceptor layer, a quantifying image was selected which showed GFP-positive cells the most.

### Immune cell analysis in spleen and blood

To assess the number of immune cells in the spleen, spleens were collected and single-cell suspensions were acquired by mechanical disruption. Suspensions were void of red blood cells via lysis with ACK lysis buffer (Lonza). Subsequently, cell suspensions were incubated with anti-CD16/32 monoclonal antibody (clone 2.4G2; produced in-house) for 20 min at 4 °C to block Fc receptors. Cells were washed with PBS and stained with primary antibodies and LIVE/DEAD Fixable Blue (ThermoFisher) in PBS for 25 min at 4 °C. Cells were then fixed with BD Cytofix/Cytoperm Fixation and Permeabilization solution (BD Biosciences) for 12 min at 4 °C and then washed and resuspended with PBS. Samples were acquired on a 5-laser LSRFortessa X-20 (BD Biosciences), and data were analyzed using FlowJo software (Tree Star). Unstained and single-fluorochrome-stained cells were used for compensation.

The gating strategy is shown in Supplementary Fig. 3. Antibodies used were anti-CD3-FITC (ThermoFisher, 11-0033-82), anti-F4/80-PerCP-Cy5.5 (ThermoFisher, 45-4801-82), anti-SiglecH-APC (Biolegend, 129611), anti-Ly6G-Alexa Fluor 700 (Biolegend, 127622), anti-CD11c-APC-eFluor 780 (ThermoFisher, 47-0114-82), anti-Ly6C-eFluor 450, eBioscience (ThermoFisher, 48-5932-82), anti-MHC-II-Brilliant Violet 510 (Biolegend, 748845), anti-CD11b-Brilliant Violet 650 (Biolegend, 101259), anti-CD19-Brilliant Violet 785 (Biolegend, 115543), LIVE/DEAD Fixable Blue Dead Cell Stain (ThermoFisher, L23105), anti-CD8-BUV737 (BD Biosciences, 612759), anti-CD4-BUV395 (BD Biosciences, 563790) and anti-NK1.1-PE-Cy7 (ThermoFisher, 25-5941-82).

For blood samples, all analyses were performed by the Animal Diagnostic Laboratory in the Department of Comparative Medicine at Stanford University School of Medicine. Automated hematology is currently performed on the Sysmex XN-1000V hematology analyzer system. Blood smears were made for all complete blood count samples and reviewed by a clinical laboratory scientist. Manual differentials were performed as indicated by species and with automated analysis.

### Mouse *Aavr* expression by cell type

Data from Tabula Muris for expression of AU040320, which encodes for AAVR in mice, were used. Data for FACS-sorted cells were obtained (https://www.czbiohub.org/tabula-muris/, retrieved on 12 December 2021) and graphed for all cell types with >100 cells in the original analysis.

### Statistics and reproducibility

Sample sizes were not determined before each study. The numbers of replicates were chosen for in vivo studies based on the number of mice of each genotype available and reasonable to use for each experiment. In vitro experiments had an $n \geq 3$ and as listed in the manuscript. For most studies, the number of mice that were age-matched with the required genotypes were used whenever possible. This resulted in varying group sizes for some studies, with $n$ between 3 and 5 for most experiments and with group size listed in the manuscript. No relevant data were excluded. Some timepoints were not collected with the in vivo imaging due to death of mice. We do not believe deaths were associated with any experimental procedures. The experiments were not randomized, as the treatment was standardized with the variable being the genotype of the animal. While no intentional blinding was used during AAV injections and data collection, mice and samples were numbered by toe tattoo and were all treated similarly. Mice of different genotypes were co-housed and injections, sample processing and data acquisition were performed in parallel without regard for genotype. When bias was possible in cell-type identification in retinal transduction, data were collected and analyzed in a blinded manner as outlined in Methods.

In vitro transduction experiments were repeated at least two times with similar results. In vivo transduction of muscle in SELECTIV-WB mice was performed once. In vivo transduction of MuSCs was performed two times with similar results. Retinal transduction experiments were repeated two times with similar results. Heart- and liver-specific transduction experiments were each performed one time with similar results, but the luciferase and GFP experiments support the outcome of each experiment, with transduction phenotypes being consistent between the two experiments. Three independent mice were shown to have similar transduction for the heart- and liver-specific transduction based on GFP expression. PET/CT experiments were performed once. Brain transduction experiments were performed twice, showing similar results.

### Reporting summary

Further information on research design is available in the Nature Portfolio Reporting Summary linked to this article.

## Data availability

Statistical source data are provided for Figs. 1–5 and Extended Data Fig. 2. Unprocessed western blots are provided for Extended Data Fig. 2. All other datasets generated during and/or analyzed during the current study are available from the corresponding author on request. Chat expression in mouse brains from Extended Data Fig. 9a is available through the Allen Brain Atlas (https://mouse.brain-map.org/experiment/show?id=253). AAVR expression data were obtained from the Tabula Muris (https://www.czbiohub.org/sf/tabula-muris/). Mice have been deposited with The Jackson Laboratory Repository with the SELECTIV mice (JAX Stock No. 037553 C57BL/6J-Igs2tm1(CAG-AU040320/mCherry,-cas9*)Janc/J) and *Aavr*-KO mice backcrossed to C57BL/6 (JAX Stock No. 037596 B6.FVB-AU040320em1Janc/J).

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

## Acknowledgements

We thank J. Yang at Stanford University for helping with tissue processing and data quantification. We thank D. Wu and the rest of Stanford Animal Histology Services for help with preparation of histologic specimens. We also thank the Animal Diagnostics Lab. We thank Stanford Center for Innovation in *In Vivo* Imaging (SCi³) and Stanford Preclinical Imaging Facility at Porter Drive for in vivo imaging help. This work was partially funded by the National Institutes of Health (NIH) grant no. R01 AI130123 (J.E.C. and J.Z.), NIH grant no. R01AI169467 (J.E.C.), NIH grant no. R01AI140186 (J.E.C.), NIH grant no. R01 AI141970 (J.E.C.), NIH grant no. R01 AI153169 (J.E.C.), NIH grant no. T32 AI007328 (J.Z.), NIH grant no. AI158808 (D.R.C. and J.I.), grant no. R01 CA219994 (D.R.C. and J.I.), NIH grant no. R01 EB028646 (K.W.F., B.W., M.R. and J.W.S.), NIH grant no. AG069858 (H.M.B.), NIH grant no. AG020961 (H.M.B.), NIH grant no. R01 EY032159 (Y.S. and V.B.M.), NIH grant no. R01 EY025295 (Y.S.), NIH grant no. R01 EY032159 (Y.S.), grant no. VA I01 CX001481 (Y.S.), NIH grant no. NEI P30 (Stanford, Ophthalmology), International Retinal Research Foundation grant no. PR810542 (K.N.), NIH grant no. R01 EY030151 (V.B.M.), NIH grant no. R01 EY024665 (V.B.M.), NIH grant no. R01 EY025225 (V.B.M.), NIH grant no. P30 EY026877 (V.B.M.), NIH grant no. K99 NS120278 (Y.X.W.), the California Institute for Regenerative Medicine (CIRM) grant no. DISC2-10604 (H.M.B.), CIRM grant no. DISC1-10036 (H.M.B.), the Baxter Foundation (H.M.B.), the Li Ka Shing Foundation (H.M.B.), the Burroughs Wellcome Fund Investigators in the Pathogenesis of Infectious Disease (J.E.C.), the Milky Way Research Foundation (H.M.B.), the Canadian Institutes of Health Research (Y.X.W. and D.R.C.), Stanford ChEM-H IMA (V.B.M.), the Stanford Center for Optic Disc Drusen (V.B.M.), Research to Prevent Blindness New York, New York (V.B.M.), the BrightFocus Foundation (K.N.), the International Retinal Research Foundation (K.N.) and Stanford Maternal & Child Health Research Institute (Y.S.).

## Author contributions

J.Z. designed and performed in vitro and in vivo experiments related to AAV transduction in SELECTIV mice and MEFs, and was responsible for Cre mouse breeding, data analysis and paper preparation. Y.X.W. designed, performed and analyzed in vivo MuSC experiments and MuSC FACS, and assisted with paper preparation. J.W.S. designed, performed and analyzed PET/CT experiments; performed confocal microscopy; and assisted with paper preparation. K.N. designed, performed and analyzed retinal transduction experiments, and assisted with paper preparation. D.R.C. designed, performed and analyzed immune profiling experiments, and assisted with paper preparation. J.N.H., C.H. and S.S. assisted with MuSC experiments. B.W. and M.R. assisted with PET/CT experiments. S.P. and A.S.P. initiated the project and cloned the SELECTIV construct. M.M.W. advised on SELECTIV construct design and assisted with paper preparation. J.I. supervised the immune phenotyping work. C.M.N. supervised mouse work, performed breeding and in vivo experiments, and assisted with paper preparation. Y.S. and V.B.M. supervised retinal transduction experiments. K.W.F. supervised PET/CT experiments. H.M.B. supervised MuSC experiments. J.E.C. supervised production of the SELECTIV construct, mouse generation, in vivo and in vivo SELECTIV experiments, and preparation of the paper. All authors gave input on the paper before submission.

## Competing interests

J.E.C., S.P. and A.S.P. are inventors on a patent filed by Stanford University regarding the use of AAVR to enhance and regulate AAV transduction. The other authors declare no competing interests.

## Additional information

**Extended data** is available for this paper at https://doi.org/10.1038/s41592-023-01896-x.

**Correspondence and requests for materials** should be addressed to Jan E. Carette.

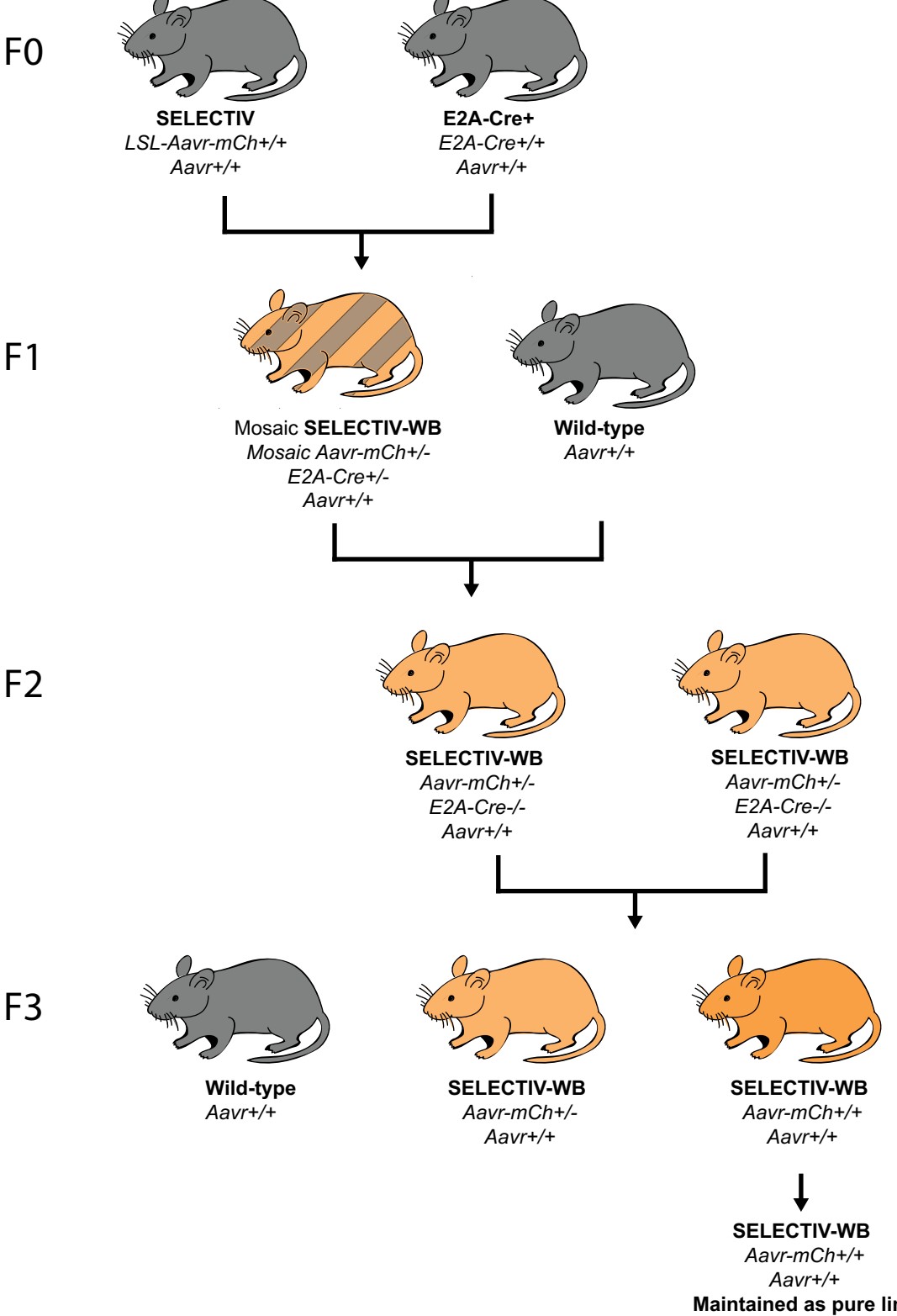

**Extended Data Fig. 1 | Breeding for SELECTIV-WB full body AAVR-overexpressing mice.** SELECTIV (*LSL-Aavr-mCh+/+, Aavr+/+*) mice were bred with E2A-Cre mice from the Jackson Laboratory (*E2A-Cre+/+ Aavr+/+*), which express Cre in early embryogenesis. This results in the F1 generation, which had partial editing of the LSL allele and results in mosaic overexpression of the *Aavr-mCh* transgene, including in some germline cells. These mice are crossed with wild-type mice to generate the F2 generation, where some mice will be *Aavr-mCh+/−* and *E2A-Cre−/−* (SELECTIV-WB). These mice are then crossed to generate the F3 generation, which includes the SELECTIV-WB mice (*Aavr-mCh+/+, Aavr+/+*), which were maintained as a pure line of AAVR-mCh overexpressing mice.

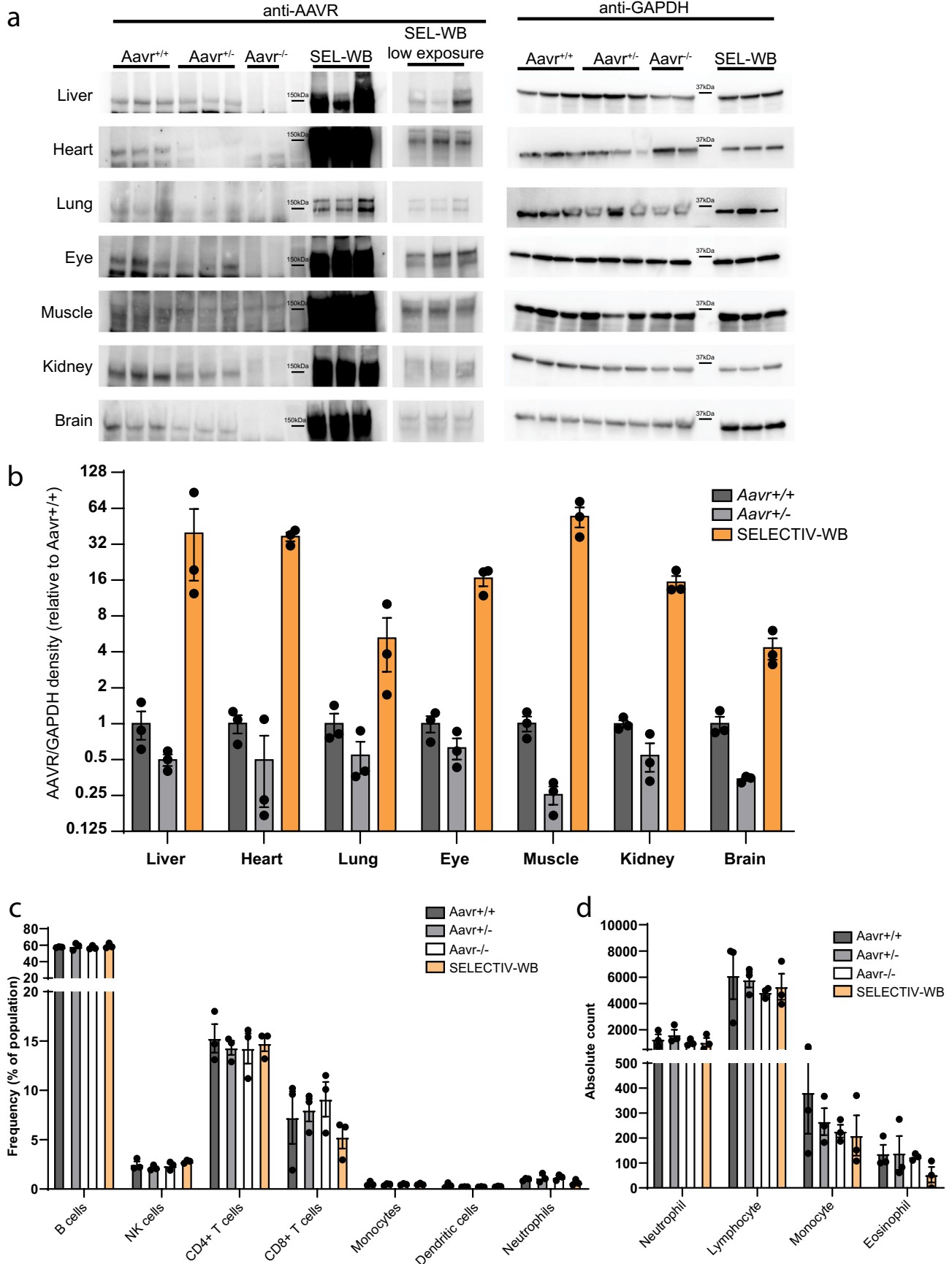

**Extended Data Fig. 2 | See next page for caption.**

**Extended Data Fig. 2 | AAVR protein expression and immune cell profiling in mouse lines.** Tissue from $Aavr^{+/+}$, $Aavr^{+/-}$, $Aavr^{-/-}$ and, SELECTIV-WB mice were homogenized and AAVR and GAPDH protein was detected by western blotting. Samples were normalized for tissue weight and equal amounts were loaded except for the SELECTIV-WB samples for heart and lung, which were diluted 1:10 prior to loading. AAVR was not detected in AAVR-KO mice, while SELECTIV-WB mice have highly increased AAVR protein levels. **b**, Semi-quantitative analysis was performed by quantifying the relative expression of AAVR compared to GAPDH for each sample (all samples normalized to $Aavr^{+/+}$ for each tissue). Mean value and SEM are shown (n = 3 mice) **c-d**, Immune cells populations in the spleen (**c**) and circulation (**d**) were quantified. There were no apparent differences in cell quantities compared to $Aavr^{+/+}$ mice.

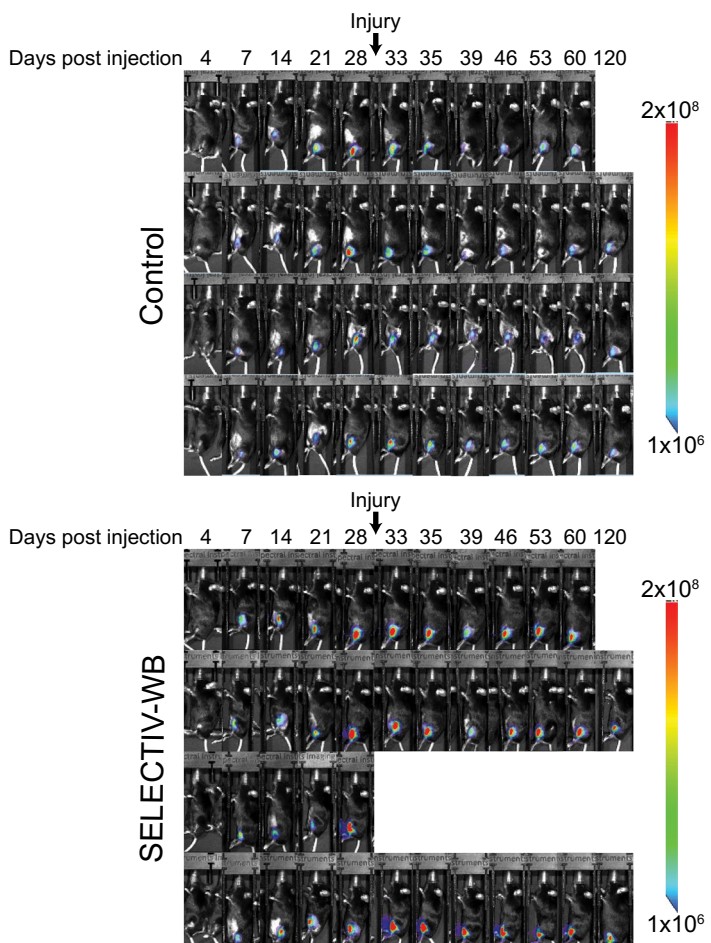

**Extended Data Fig. 3 | *In vivo* imaging of mice transduced with AAV2-luc by intramuscular injection.** Control and SELECTIV-WB mice were injected with AAV2-luciferase by intramuscular injection and luciferase activity was tracked over time through *in vivo* imaging. Mice were injured by BaCl$_2$ injection at day 30, and transduction continued to be tracked until day 120 post transduction.

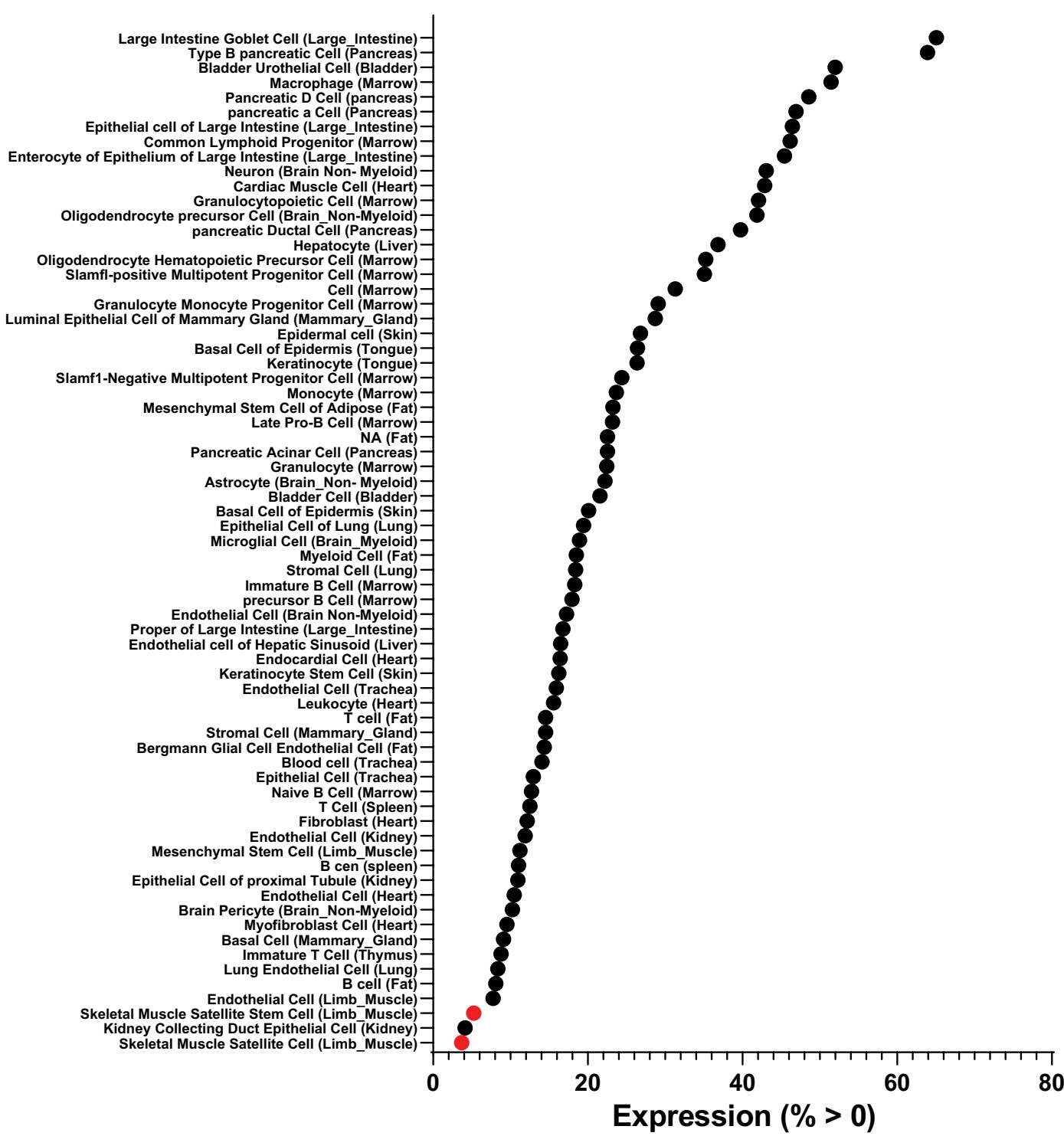

**Extended Data Fig. 4 | Comparison of expression of AU040320 *in vivo* in mice in various muscle cell types.** Data from Tabula Muris for expression of AU040320 (*Aavr*, mouse gene encoding AAVR), which encodes for AAVR in mice, based on single cell sequencing of FACS sorted cells. Data for the percentage of cells with detectable levels of AU040320 were graphed for all cell types with data for >100 individual cells in the original analysis. Expression of AU040320, was lowest in skeletal muscle satellite cells, with 96.3% of cells having no detectable expression. The similarly annotated skeletal muscle satellite stem cells also had minimal expression of AU040320.

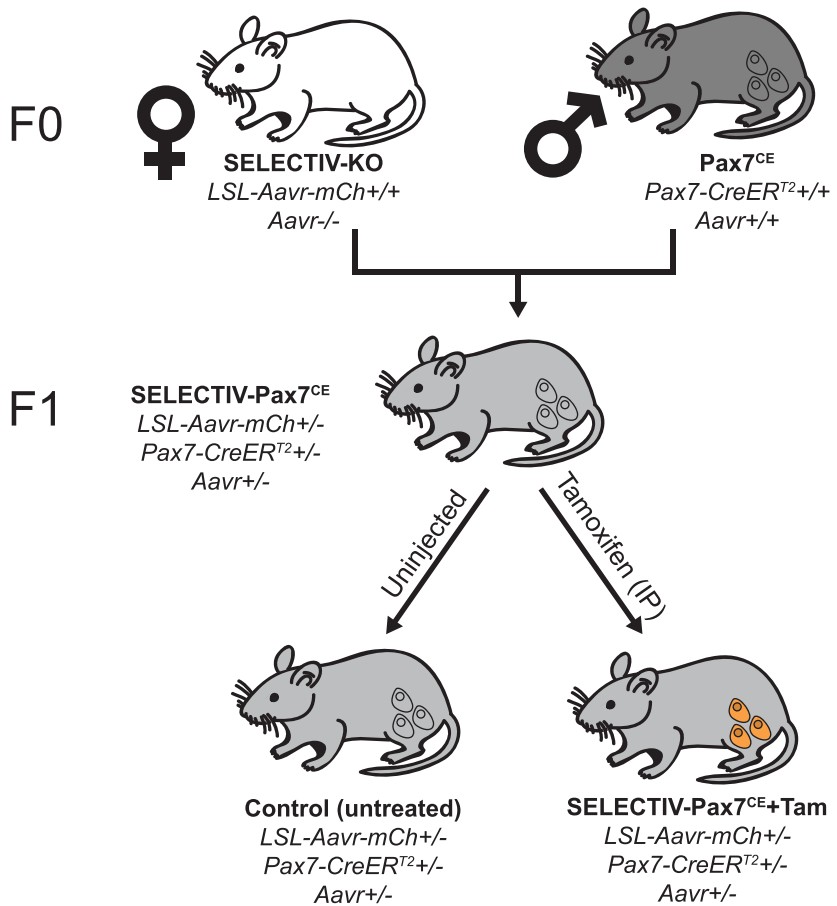

**Extended Data Fig. 5 | Breeding for SELECTIV-Pax7^CE mice with inducible muscle stem cell specific AAVR overexpression.** Female SELECTIV-KO (*LSL-Aavr-mCh+/−, Aavr−/−*) mice were bred with male Pax7^CE (*Pax7-CreER^T2*) mice that were purchased from Jax (Stock No: 017763) to generate SELECTIV-Pax7^CE (*LSL-Aavr-mCh+/−, Pax7-CreER^T2+/−, Aavr+/−*). Treatment with tamoxifen by IP injection results in Pax7-dependent expression of the SELECTIV construct.

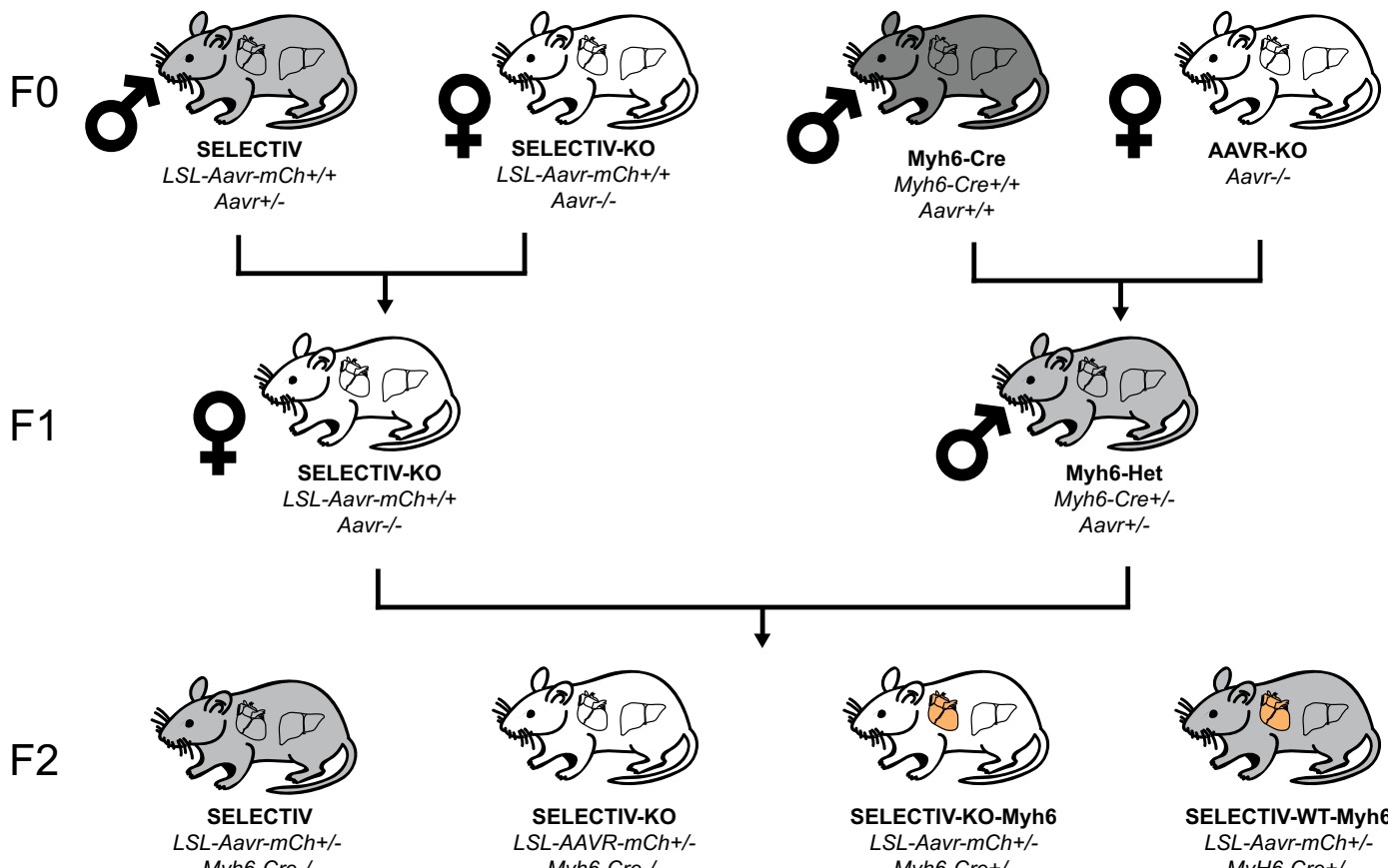

**Extended Data Fig. 6 | Breeding for SELECTIV-KO-Myh6 mice with heart cardiomyocyte specific AAVR overexpression.** SELECTIV (LSL-*Aavr-mCh*+/+, *Aavr*+/−) male and SELECTIV-KO (*LSL-Aavr-mCh*+/+, *Aavr*−/−) female mice were crossed to generate and maintain SELECTIV-KO female mice for breeding. Male mice on the BL/6 background that are *Aavr*−/− are sterile, so they must be maintained as *Aavr*+/−. Myh6-Cre (*Myh6-Cre*+/+, *Aavr*+/+) male mice were purchased from Jax (Stock No: 011038) and bred with AAVR-KO (*Aavr*−/−) female mice to generate Myh6-Het (*Myh6-Cre*+/−, *Aavr*+/−) male mice (optional breeding can be carried out to generate *Myh6-Cre*+/+, Aavr−/− males). The F1 generation SELECTIV-KO and Myh6-Het mice were bred to generate the F2 mice, which results in SELECTIV (*LSL-AAVR*+/−, *Myh6-Cre*−/−, *AAVR*+/−), SELECTIV-KO (LSL-*Aavr-mCh*+/−, *Myh6-Cre*−/−, *Aavr*−/−), SELECTIV-KO-Myh6 (*LSL-AAVR-mCh*+/−, *Myh6-Cre*+/−, *Aavr*−/−), and SELECTIV-WT-Myh6 (*LSL-Aavr-mCh*+/−, *Myh6-Cre*+/−, *Aavr*+/−). The SELECTIV-KO-Myh6 mice are used to show specific targeting of the heart with detargeting in the rest of the body, while other mice can be used as controls.

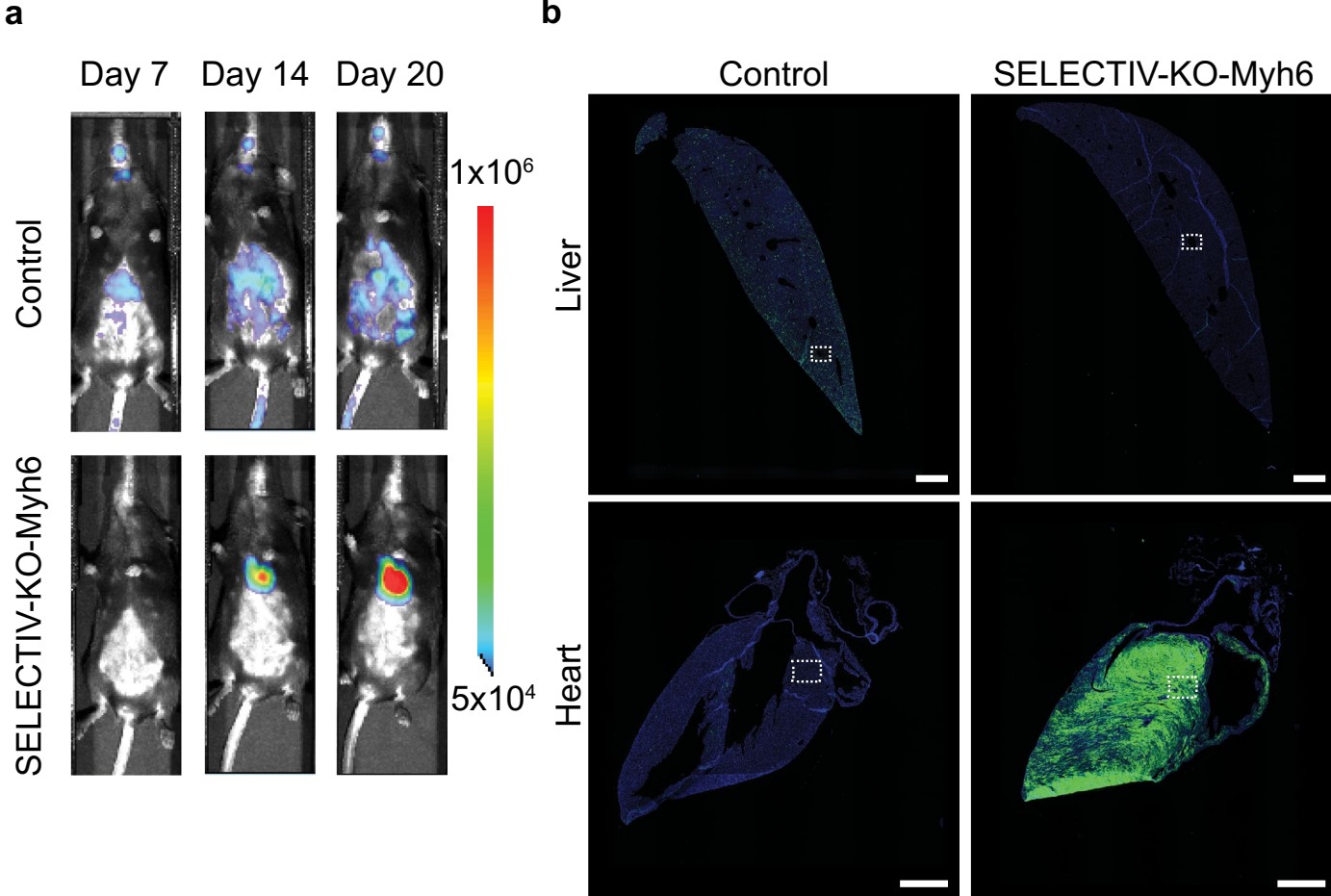

**Extended Data Fig. 7 | Transduction of SELECTIV-KO-Myh6 mice. a**, *In vivo* imaging of mice with heart cardiomyocyte specific expression of AAVR transduced with AAV9-luc. Control mice (*Aavr*+/−) or SELECTIV-KO-Myh6 mice were injected intravenously with AAV9 encoding luciferase. Representative mice are shown for mice on days 7, 14, and 20 mice after injection with luciferin and imaging using the Lago *in vivo* imaging system. Luciferase activity is more dispersed in Control mice, while activity in SELECTIV-KO-Myh6 mice is concentrated in a distinct area of the chest. **b**, Specific transduction of SELECTIV-KO-Myh6 mouse hearts by AAV-GFP. SELECTIV-KO-Myh6 and Control mice were transduced with AAV-PHP.eB encoding EGFP. 28-days post AAV transduction, hearts and livers were removed, and transduction was determined by microscopy (GFP; green). Hoechst nuclear staining in blue. Stitched pictures are shown for the inlays (dotted boxes) shown at higher magnification in Fig. 4c. Scale bar: 1 mm.

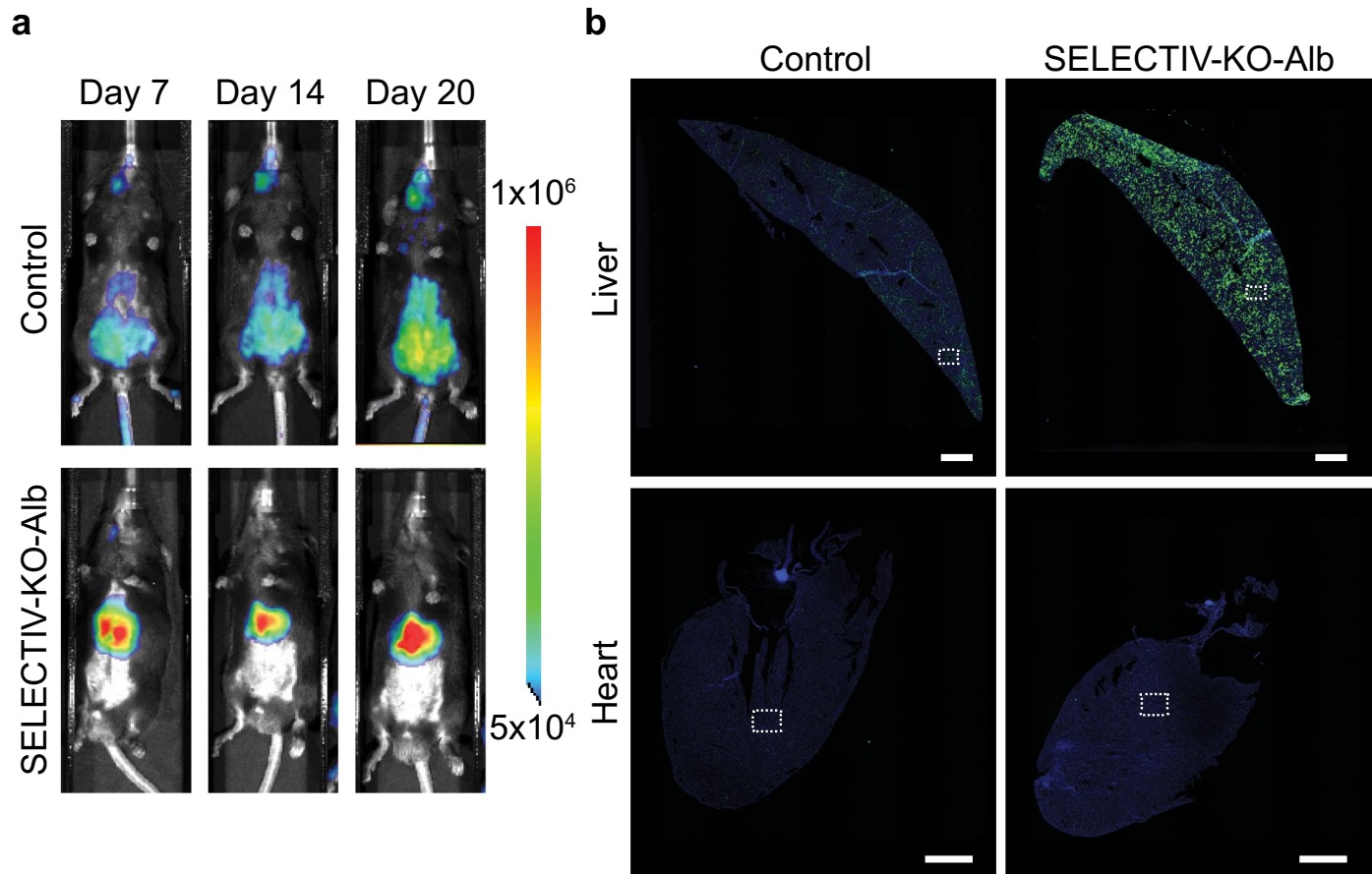

**Extended Data Fig. 8 | Transduction of SELELCTIV-KO-Alb mice. a**, *In vivo* imaging of mice with liver specific expression of AAVR transduced with AAV9-luc. Control mice (*Aavr*+/−) or SELECTIV-KO-Alb mice were injected intravenously with AAV9 encoding luciferase. Representative mice are shown for mice on days 7, 14, and 20 mice after injection with luciferin and imaging using the Lago *in vivo* imaging system. Luciferase activity can be seen throughout the body of Control mice, while high levels of activity can only be seen in the abdomen of SELECTIV-KO-Alb mice. **b**, Specific transduction of SELECTIV-KO-Alb mouse livers by AAV-GFP. Control and SELECTIV-KO-Alb mice were transduced with AAV-PHP.eB encoding EGFP. 28-days post AAV transduction, hearts and livers were removed, and transduction was determined by microscopy (GFP; green). Hoechst nuclear staining in blue. Stitched pictures are shown for the inlays (dotted boxes) shown at higher magnification in Fig. 5c. Scale bar: 1 mm.

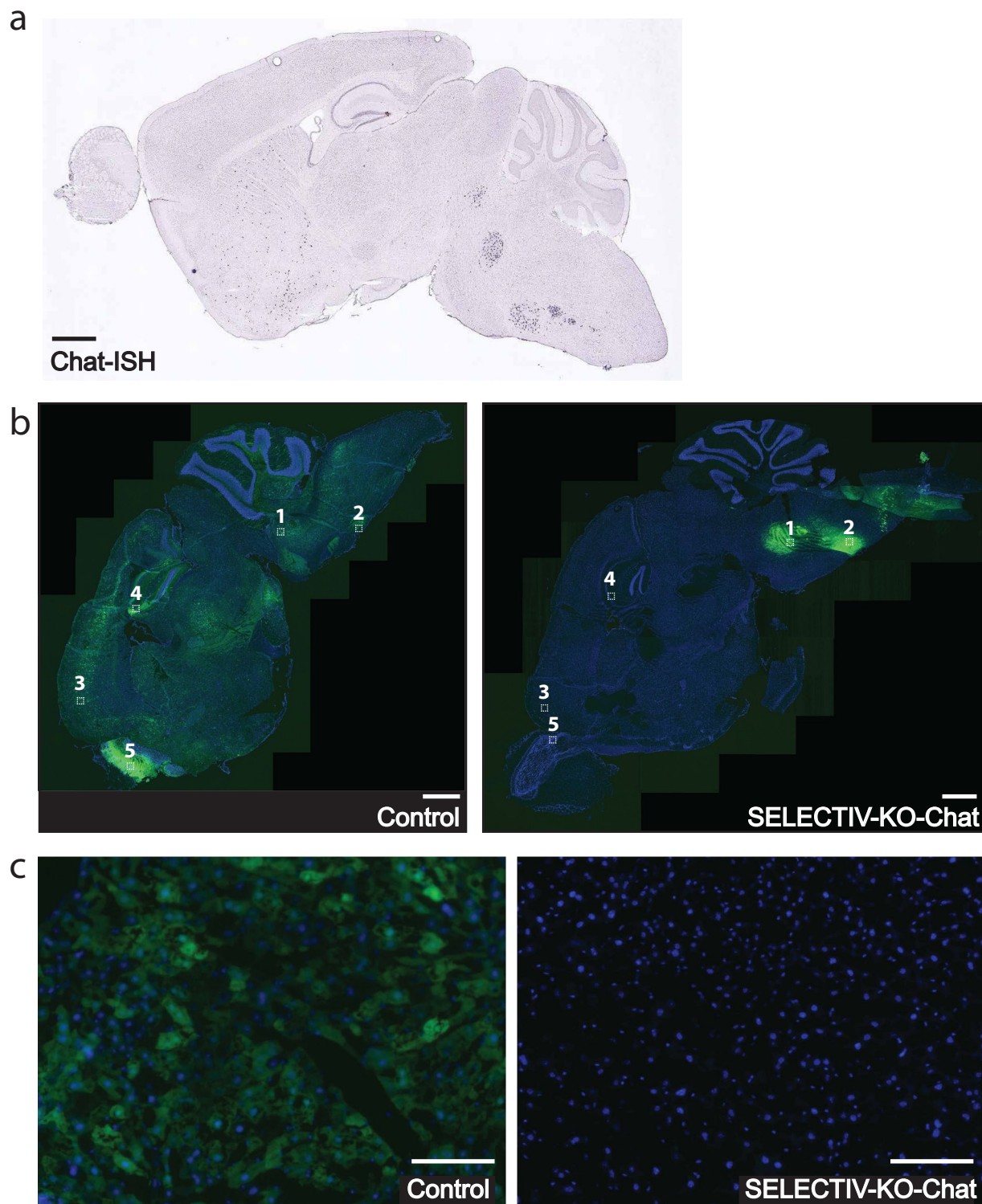

**Extended Data Fig. 9 | *In vivo* transduction of SELECTIV-KO-Chat mice. a**, Chat (choline acetyltransferase) RNA in the brain of a 56 day old male mouse (Allen Mouse Brain Atlas, https://mouse.brain-map.org/experiment/show?id=253)[54]. Scale bar: 1 mm. **b**, Outline of areas used to demonstrate AAVR overexpression and AAV transduction in Fig. 6d. Control and SELECTIV-KO-Chat mice were transduced with PHP.eB-GFP and low magnification confocal is shown of the entire brain. Outlines are shown for where higher-magnification imaging was performed in various areas of the brain: 1) Pons, 2) Medulla, 3) Cerebral Cortex, 4) Hippocampus, 5) Anterior Olfactory nucleus. Scale bar: 1 mm. **c**, Transduction in the liver of mice transduced with AAV-PHP.EB-GFP corresponding the mice in Fig. 6. Livers from the Control and SELECTIV-KO-Chat mice imaged in Fig. 6 were collected and sectioned for imaging by microscopy. Transduction was seen throughout the liver of Control mice, while no transduction was seen in the SELECTIV-KO-Chat mouse livers. Scale bar: 130 μm.

# nature research

# Reporting Summary

Nature Research wishes to improve the reproducibility of the work that we publish. This form provides structure for consistency and transparency in reporting. For further information on Nature Research policies, see our Editorial Policies and the Editorial Policy Checklist.

## Statistics

For all statistical analyses, confirm that the following items are present in the figure legend, table legend, main text, or Methods section.

| n/a | Confirmed | |
|---|---|---|
| ☐ | ☒ | The exact sample size (*n*) for each experimental group/condition, given as a discrete number and unit of measurement |
| ☐ | ☒ | A statement on whether measurements were taken from distinct samples or whether the same sample was measured repeatedly |
| ☐ | ☒ | The statistical test(s) used AND whether they are one- or two-sided *Only common tests should be described solely by name; describe more complex techniques in the Methods section.* |
| ☐ | ☒ | A description of all covariates tested |
| ☐ | ☒ | A description of any assumptions or corrections, such as tests of normality and adjustment for multiple comparisons |
| ☐ | ☒ | A full description of the statistical parameters including central tendency (e.g. means) or other basic estimates (e.g. regression coefficient) AND variation (e.g. standard deviation) or associated estimates of uncertainty (e.g. confidence intervals) |
| ☐ | ☒ | For null hypothesis testing, the test statistic (e.g. *F*, *t*, *r*) with confidence intervals, effect sizes, degrees of freedom and *P* value noted *Give P values as exact values whenever suitable.* |
| ☒ | ☐ | For Bayesian analysis, information on the choice of priors and Markov chain Monte Carlo settings |
| ☒ | ☐ | For hierarchical and complex designs, identification of the appropriate level for tests and full reporting of outcomes |
| ☒ | ☐ | Estimates of effect sizes (e.g. Cohen's *d*, Pearson's *r*), indicating how they were calculated |

*Our web collection on statistics for biologists contains articles on many of the points above.*

## Software and code

Policy information about availability of computer code

| Data collection | In vivo luciferase data was collected using Aura version 4.0.0 (Spectral Instruments Imaging). Luciferase data was collected using GloMax (Promega). Retinal imaging was collected using Heidelberg Spectralis SLO/OCT system software. FACS data was collected using SH800S (Sony) and FACSDiva version 8.0.1 (BD). LAS X, Echo Revolve D270 version 3.0.3, Keyence BZ-X800 software, and Zeiss Zen software were used for image collection. |
|---|---|
| Data analysis | In vivo luciferase data was analyzed using Aura version 4.0.0 (Spectral Instruments Imaging). GraphPad Prism 8.0.2 was use for data analysis and statistical testing. LAS X 3.7.4.23463, ImageJ version 1.52n, Zeiss Zen black 2.3 SP1 for imaging z-stack and max projections and Zeiss Zen Blue 2.3 for quantification, and Keyence BZ-X800 software version 1.1.2.4 were used for fluorescent image processing. Sequencher version 5.1 was used for sequence analysis. FlowJo version 10.8.2 and Sony Cell Sorter software version 2.1.5 was used for FACS analysis. Western Blot analysis was performed using Bio-Rad Image Lab Software version 6.1.0 build 7. |

For manuscripts utilizing custom algorithms or software that are central to the research but not yet described in published literature, software must be made available to editors and reviewers. We strongly encourage code deposition in a community repository (e.g. GitHub). See the Nature Research guidelines for submitting code & software for further information.

## Data

Policy information about availability of data

All manuscripts must include a data availability statement. This statement should provide the following information, where applicable:
- Accession codes, unique identifiers, or web links for publicly available datasets
- A list of figures that have associated raw data
- A description of any restrictions on data availability

Statistical source data is included for Fig. 1, 2, 3, 4, and 5 and Extended Data Fig. 2. Unprocessed Western Blots are provided for Extended Data. Fig. 2. All other

Chat expression in mouse brains from Extended Data Fig 9a is available through Allen Brain Atlas (https://mouse.brain-map.org/experiment/show?id=253). AAVR expression data was obtain from the Tabula Muris (https://www.czbiohub.org/sf/tabula-muris/).
Mice have been deposited with The Jackson Laboratory Repository with the SELECTIV mice (JAX Stock No. 037553 C57BL/6J-Igs2tm1(CAG-AU040320/mCherry,-cas9*)Janc/J) and Aavr-KO mice backcrossed to C57BL/6 (JAX Stock No. 037596  B6.FVB-AU040320em1Janc/J).

# Field-specific reporting

Please select the one below that is the best fit for your research. If you are not sure, read the appropriate sections before making your selection.

☒ Life sciences   ☐ Behavioural & social sciences   ☐ Ecological, evolutionary & environmental sciences

For a reference copy of the document with all sections, see nature.com/documents/nr-reporting-summary-flat.pdf

# Life sciences study design

All studies must disclose on these points even when the disclosure is negative.

| | |
|---|---|
| Sample size | Sample sizes were not determined before each study. The number of replicates was chosen for in vivo studies based on the number of mice of each genotype available and reasonable to use for each experiment.  In vitro experiments had an n>=3 and listed in the manuscript. For most studies the number of mice that were age matched with the required genotypes were used whenever possible.  This resulted in varying group sizes for some studies with n between 3 and 5 for most experiments with group size listed in the mansucript. |
| Data exclusions | No relevant data were excluded. Some timepoints were not collected with the in vivo imaging due to death of mice. We do not believe deaths were associated with any experimental procedures. |
| Replication | As no data were excluded, all values from each experiment are presented and the variation can be seen in the figures. Statistical analysis as described throughout the manuscript provided the ability to confidently assess differences between different experimental conditions. |
| Randomization | No randomization was used. Mice were used from litters as needed for experiments. Mice of different genotypes were not treated differently and were not |
| Blinding | While no intentional binding was used during AAV injections and data collection, mice and samples were number by toe tattoo and were all treated similarly. Mice of different genotypes were co-housed injections, samples processing, and data acquisition were performed in parallel without regard for genotype. When bias was possible in cell type identification in retinal transduction, data was collected and analyzed in a blinded manner as outlined in the methods. |

# Reporting for specific materials, systems and methods

We require information from authors about some types of materials, experimental systems and methods used in many studies. Here, indicate whether each material, system or method listed is relevant to your study. If you are not sure if a list item applies to your research, read the appropriate section before selecting a response.

## Materials & experimental systems

| n/a | Involved in the study |
|---|---|
| ☐ | ☒ Antibodies |
| ☐ | ☒ Eukaryotic cell lines |
| ☒ | ☐ Palaeontology and archaeology |
| ☐ | ☒ Animals and other organisms |
| ☒ | ☐ Human research participants |
| ☒ | ☐ Clinical data |
| ☒ | ☐ Dual use research of concern |

## Methods

| n/a | Involved in the study |
|---|---|
| ☒ | ☐ ChIP-seq |
| ☐ | ☒ Flow cytometry |
| ☒ | ☐ MRI-based neuroimaging |

# Antibodies

| | |
|---|---|
| Antibodies used | rabbit anti-KIAA0319L, Proteintech, 21016-1-AP, 1:1000<br>anti-rabbit-HRP, Genetex, GTX213110-01, 1:4000<br>PE-Cy7-CD11b, Biolegend, M1/70, cat# 101216, 3μl/mouse<br>PE-Cy7-CD45, Biolegend, 23-F11, cat# 103114, 3μl/mouse<br>PE-Cy7-Sca1, Biolegend, D7, cat# 108114, 3μl/mouse<br>PE-Cy7-CD31, Biolegend, 390 cat# 102418, 3μl/mouse<br>PE-a7-integrin, AbLab, R2F2, SKU: 53-0010-05, 2μl/mouse<br>APC-CD34, Biolegend, RAM34, cat# 128612, 3μl/mouse<br>RBPMS, PhosphoSolutions, cat# 1832-RBPMS, 1:2000<br>anti-Guinea pig Alexa Fluor 647, Invitrogen, A-21450, 1:200<br>anti-CD3-FITC (ThermoFisher, 11-0033-82), 1:100 |

anti-F4/80-PerCP-Cy5.5 (ThermoFisher, 45-4801-82), 1:100
anti-SiglecH-APC (Biolegend, 129611), 1:200
anti-Ly6G-Alexa Fluor® 700 (Biolegend, 127622), 1:100
anti-CD11c-APC-eFluor™ 780 (ThermoFisher, 47-0114-82), 1:200
anti-Ly6C-eFluor 450, eBioscience™ (ThermoFisher, 48-5932-82), 1:100
anti-MHC-II-Brilliant Violet 510™ (Biolegend, 748845), 1:200
anti-CD11b-Brilliant Violet 650™ (Biolegend, 101259), 1:400
anti-CD19-BV78Brilliant Violet 785™5 (Biolegend, 115543), 1:100
anti-FIXBlue-LIVE/DEAD™ Fixable Blue Dead Cell Stain  (ThermoFisher, L23105), 1:200
anti-CD8-BUV737 (BD Bioscience, 612759), 1:400
anti-CD4-BUV395 (BD Bioscience, 563790), 1:200
anti-NK1.1-PE-Cy7 (ThermoFisher, 25-5941-82), 1:100
anti-CD16/32 mAb (clone 2.4G2; produced in house)

**Validation**

Rabbit anti-KIAA0319L antibody was validated on AAVR KO cells and tissue (as shown in supplemental).
anti-CD16/32 mAb was produced by hybridoma in house and tested functionally for Fc receptor blockade for mouse cells prior to use.
All other antibodies were validated by supplier prior to use as described below.
PE-Cy7-CD11b, Biolegend, M1/70, cat# 101216, Application: FC (Flow Cytometry).The validation has been performed by the manufacturer: https://www.biolegend.com/en-ie/products/pe-cyanine7-anti-mouse-human-cd11b-antibody-1921
PE-Cy7-CD45, Biolegend, 23-F11, cat# 103114, Application: FC (Flow Cytometry).The validation has been performed by the manufacturer: https://www.biolegend.com/en-us/products/pe-cyanine7-anti-mouse-cd45-antibody-1903
PE-Cy7-Sca1, Biolegend, D7, cat# 108114, Application: FC (Flow Cytometry).The validation has been performed by the manufacturer: https://www.biolegend.com/en-us/search-results/pe-cyanine7-anti-mouse-ly-6a-e-sca-1-antibody-3137
PE-Cy7-CD31, Biolegend, 390 cat# 102418, Application: FC (Flow Cytometry).The validation has been performed by the manufacturer: https://www.biolegend.com/en-us/products/pe-cyanine7-anti-mouse-cd31-antibody-3942
PE-a7-integrin, AbLab, R2F2, SKU: 53-0010-05, Application: FC (Flow Cytometry).The validation in a published protocol: https://www.ncbi.nlm.nih.gov/pmc/articles/PMC5034768/
APC-CD34, Biolegend, RAM34, cat# 128612, Application: FC (Flow Cytometry).The validation has been performed by the manufacturer: https://www.biolegend.com/en-us/products/apc-anti-mouse-cd34-antibody-6520
RBPMS, PhosphoSolutions, cat# 1832-RBPMS, Application: IFA (Flow Cytometry).The validation has been performed by the manufacturer: https://www.phosphosolutions.com/products/anti-rbpms-antibody-1832-rbpms
anti-Guinea pig Alexa Fluor 647, Invitrogen, A-21450, Application: IFA (Flow Cytometry).The validation has been performed by the manufacturer: https://www.thermofisher.com/antibody/product/Goat-anti-Guinea-Pig-IgG-H-L-Highly-Cross-Adsorbed-Secondary-Antibody-Polyclonal/A-21450
anti-CD3-FITC (ThermoFisher, 11-0033-82), Application: FC (Flow Cytometry).The validation has been performed by the manufacturer: https://www.thermofisher.com/antibody/product/CD3e-Antibody-clone-eBio500A2-500A2-Monoclonal/11-0033-82
anti-F4/80-PerCP-Cy5.5 (ThermoFisher, 45-4801-82), Application: FC (Flow Cytometry).The validation has been performed by the manufacturer: https://www.thermofisher.com/antibody/product/F4-80-Antibody-clone-BM8-Monoclonal/45-4801-82
anti-SiglecH-APC (Biolegend, 129611), Application: FC (Flow Cytometry).The validation has been performed by the manufacturer: https://www.biolegend.com/en-us/products/apc-anti-mouse-siglec-h-antibody-6906
anti-Ly6G-Alexa Fluor® 700 (Biolegend, 127622), Application: FC (Flow Cytometry).The validation has been performed by the manufacturer: https://www.biolegend.com/en-us/products/alexa-fluor-700-anti-mouse-ly-6g-antibody-6754
anti-CD11c-APC-eFluor™ 780 (ThermoFisher, 47-0114-82), Application: FC (Flow Cytometry).The validation has been performed by the manufacturer: https://www.thermofisher.com/antibody/product/CD11c-Antibody-clone-N418-Monoclonal/47-0114-82
anti-Ly6C-eFluor 450, eBioscience™ (ThermoFisher, 48-5932-82), Application: FC (Flow Cytometry).The validation has been performed by the manufacturer: https://www.thermofisher.com/antibody/product/Ly-6C-Antibody-clone-HK1-4-Monoclonal/48-5932-82
anti-MHC-II-Brilliant Violet 510™ (Biolegend, 748845), Application: FC (Flow Cytometry).The validation has been performed by the manufacturer: https://www.biolegend.com/de-de/products/brilliant-violet-510-anti-mouse-i-a-i-e-antibody-7997
anti-CD11b-Brilliant Violet 650™ (Biolegend, 101259), Application: FC (Flow Cytometry).The validation has been performed by the manufacturer: https://www.biolegend.com/en-us/products/brilliant-violet-650-anti-mouse-human-cd11b-antibody-7638
anti-CD19-BV78Brilliant Violet 785™ (Biolegend, 115543), Application: FC (Flow Cytometry).The validation has been performed by the manufacturer: https://www.biolegend.com/en-ie/products/brilliant-violet-785-anti-mouse-cd19-antibody-7962
anti-FIXBlue-LIVE/DEAD™ Fixable Blue Dead Cell Stain  (ThermoFisher, L23105), Application: FC (Flow Cytometry).The validation has been performed by the manufacturer: https://www.thermofisher.com/order/catalog/product/L23105
anti-CD8-BUV737 (BD Bioscience, 612759), Application: FC (Flow Cytometry).The validation has been performed by the manufacturer: https://www.bdbiosciences.com/en-us/products/reagents/flow-cytometry-reagents/research-reagents/single-color-antibodies-ruo/buv737-rat-anti-mouse-cd8a.612759
anti-CD4-BUV395 (BD Bioscience, 563790), Application: FC (Flow Cytometry).The validation has been performed by the manufacturer: https://www.bdbiosciences.com/en-us/products/reagents/flow-cytometry-reagents/research-reagents/single-color-antibodies-ruo/buv395-rat-anti-mouse-cd4.565974
anti-NK1.1-PE-Cy7 (ThermoFisher, 25-5941-82), Application: FC (Flow Cytometry).The validation has been performed by the manufacturer: https://www.thermofisher.com/antibody/product/NK1-1-Antibody-clone-PK136-Monoclonal/25-5941-82

# Eukaryotic cell lines

Policy information about cell lines

| | |
|---|---|
| Cell line source(s) | Mouse embryonic fibroblasts (MEFS) and myoblasts were generated as described in the methods from control and SELECTIV-WB mice. |
| Authentication | We used primary MEFs and myoblasts using standard isolation protocols. |
| Mycoplasma contamination | Cell lines are negative for mycoplasma as tested by mycoalert plus mycoplasma detection kit (Lonza). |

| Commonly misidentified lines (See ICLAC register) | No commonly misidentified lines were used in these studies. |
|---|---|

# Animals and other organisms

Policy information about studies involving animals; ARRIVE guidelines recommended for reporting animal research

| Laboratory animals | Mouse lines purchased from Jackson Laboratory include C57BL/6J (664, Jax), E2A-Cre (B6.FVB-Tg(EIIa-cre)C5379Lmgd/J, 3724, Jax), Chat-Cre (ChAT-IRES-Cre, 6410, Jax), Alb-Cre (B6.Cg-Speer6-ps1Tg(Alb-cre)21Mgn/J, 3574, Jax), Myh6-Cre (B6.FVB-Tg(Myh6-cre)2182Mds/J HEMI, 11038, Jax),  and Pax7-Cre (B6.Cg-Pax7tm1(cre/ERT2)Gaka/J, 17763, Jax). FVB AAVR KO mice were previously generated. The AAVR KO allele was introduced into C57BL/6J by backcrossing for 10 generations. SELECTIV mice were generated at the Stanford Transgenic, Knockout, and Tumor Model Center  using Integrase Mediated Transgenesis61 using the construct described above and  PhiC31 integrase. The construct was inserted into the H11 locus using C57BL/6 mice with three attP sites previously knocked into the H11 locus. Male and female mice were used for all studies as available using littermate or age matched controls. All mice were 6-12 weeks old at time of AAV injection, except for retinal transduction studies where mice were 5 months old. |
|---|---|
| Wild animals | This study did not involve wild animals. |
| Field-collected samples | This study did not involve samples collected from the field. |
| Ethics oversight | At Stanford University, the Institutional Animal Care and Use Committee (IACUC) is appointed by the University Vice Provost and Dean of Research, and is known as the Administrative Panel on Laboratory Animal Care (APLAC). Stanford's APLAC membership is comprised of faculty, veterinarians, public members, students, and senior staff. The APLAC reports to the Office of the Vice Provost and Dean of Research. The laboratory animal care program at Stanford is accredited by AAALAC International. All procedures were carried out under the approved protocol APLAC #28856. Mice were housed in the AAALAC-accredited Stanford mouse barrier facility. Husbandry was performed in accordance with the Guide for the Care and Use of Laboratory Animals, 8th edition (PMID 21595115), and the Public Health Service Policy on Humane Care and Use of Laboratory Animals (2015). |

Note that full information on the approval of the study protocol must also be provided in the manuscript.

# Flow Cytometry

## Plots

Confirm that:

☒ The axis labels state the marker and fluorochrome used (e.g. CD4-FITC).

☒ The axis scales are clearly visible. Include numbers along axes only for bottom left plot of group (a 'group' is an analysis of identical markers).

☒ All plots are contour plots with outliers or pseudocolor plots.

☒ A numerical value for number of cells or percentage (with statistics) is provided.

## Methodology

| Sample preparation | Mouse hindlimb muscles were carefully dissected to remove adipose, tendon and nerves and minced using scissors until homogenous. Minced tissue was suspended in 10mL of collagenase solution (700U/mL collagenase II (Worthington), 0.2% BSA (Sigma) in Ham's F-10 media (Gibco) and digested at 37C for 1h using the gentleMACS Octo Dissociator. Digested tissues were diluted with 20mL of 0.2% BSA in Ham's F-10 media) and pelleted at 500g for 10mins. Supernatant removed leaving 8mL of solution and cell pellet. Cell pellets were resuspended and 1000U of collagenase II and 11U of dispase I (Life Technologies) in 2mL of PBS were added and further digested for 30mins on the gentleMACS Dissociator using a custom program. Cells were pelleted, washed in FACS buffer (0.5% BSA, 1mM EDTA in PBS) and filtered through a 40um nylon cell filter. Red blood cells were lysed using RBC Lysis Buffer (eBioscience). Single-cell suspensions were incubated with direct APC-Cy7 conjugated antibodies against CD11b (M1/70), CD45 (30-F11), Sca1 (D7), and CD31 (390) (Biolegend, 3uL of each/mouse), PE-a7-integrin (2uL/mouse, AbLab), and APC-CD34 (RAM34, Biolegend). Cell suspensions were then subjected to FACS sorting for MuSC isolation (CD45– CD11b– CD31– Sca1– α7-integrin+ CD34+ cells) or flow cytometry quantification for GFP+ MuSCs (CD45– CD11b– CD31– Sca1– α7-integrin+). |
|---|---|
| | To assess the number of immune cells in the spleen, spleens were harvested and single cell suspensions were acquired by mechanical disruption. Suspensions were void of red blood cells via lysis with Ammonium-Chloride-Potassium (ACK) Lysis buffer (Lonza). Subsequently, cell suspensions were incubated with anti-CD16/32 mAb (clone 2.4G2; produced in house) for 20 minutes at 4°C to block Fc receptors. Cells were washed with PBS and stained with primary antibodies and LIVE/DEAD Fixable Blue (ThermoFisher) in PBS for 25 minutes at 4°C. Cells were then fixed with BD Cytofix/Cytoperm Fixation and Permeabilization solution (BD Bioscience) for 12 mins at 4°C and then washed and resuspended with PBS. Samples were acquired on a 5-laser LSRFortessa X-20 (BD Biosciences), and data was analyzed using FlowJo software (Tree Star, Inc). Unstained and single-fluorochrome-stained cells were used for compensation. Antibodies used were anti-CD3-FITC (ThermoFisher, 11-0033-82), anti-F4/80-PerCP-Cy5.5 (ThermoFisher, 45-4801-82), anti-SiglecH-APC (Biolegend, 129611), anti-Ly6G-Alexa Fluor® 700 (Biolegend, 127622), anti-CD11c-APC-eFluor™ 780 (ThermoFisher, 47-0114-82), anti-Ly6C-eFluor 450, eBioscience™ (ThermoFisher, 48-5932-82), anti-MHC-II-Brilliant Violet 510™ (Biolegend, 748845), anti-CD11b-Brilliant Violet 650™ (Biolegend, 101259), anti-CD19-BV78Brilliant Violet 785™5 (Biolegend, 115543), anti-FIXBlue-LIVE/DEAD™ Fixable Blue Dead Cell Stain  (ThermoFisher, L23105), anti-CD8-BUV737 (BD Bioscience, 612759), anti-CD4-BUV395 (BD Bioscience, 563790) and,anti-NK1.1-PE-Cy7 (ThermoFisher, 25-5941-82). |

| | |
|---|---|
| Instrument | MEF and Muscle samples: Sony SH800S and BD LSR II UV. analyzer. Spleen samples: 5-laser LSRFortessa X-20 (BD Biosciences) |
| Software | Analysis was performed with FlowJo and Sony Cell Sorter Software |
| Cell population abundance | After processing MuSCs made up ~1-2% of cells in the diaphragm samples and 2-5% of cells in the TA samples. MEF transduction was seen by GFP positivity in ~13-14% of cells. |
| Gating strategy | Flow cytometry and sorting strategies for MuSCs were performed as previously described in Sacco et al., 2008 Nature, Cosgrove, et al. 2014 Nat Med, Ho et al., 2017 PNAS. Briefly, live mononuclear cells dissociated from AAV injected muscles were gated by FSC-A vs. SSC-A to remove low and high SSC debris. Then singlets were selected for by gating on FSC-w and FSC-h. Live cells were selected based on low DAPI signal. MuSCs were enriched by gating for a7-PE positivity and lineage marker (CD45-PE-Cy7, CD11b-PE-Cy7, CD31-PE-Cy7, Sca1-PE-Cy7) negativity and further selected by gating on CD34-APC positive and SSC-A low populations. Based on previous data, this strategy yields MuSCs at >95% purity based on Pax7+ staining. Antibody gates and compensation were established using unstained, single stained, and FMO controls. GFP gates were established on untransduced cells or tissues. <br><br> Gating strategy for immune cells is shown in Supplementary Fig. 3. Cells were gating on SSC-A/FSC-A, followed by isolation of live cells that did not stain positive for Fix Blue. Singlets were then isolated on a FSC-W/FSC-A plot. CD19+/CD3- cells were called B-cells. The rest of the cells were further gated, with Ly6G+ CD11b+ cells being neutrophils and the rest of the cells further gated to identify NK1.1 cells as NK cells. The rest of the cells were further gated to identify CD3+ CD8+ to be CD8+ T-cells and CD3+ CD4+ to be CD4+ T cells. CD11c+ cells were identified as dendritic cells. And Ly6c+ cells were identified as Monocytes. |

☒ Tick this box to confirm that a figure exemplifying the gating strategy is provided in the Supplementary Information.

