## [Peer Review File · Nature Methods]

Peer Review Information

Manuscript Title: Hardwiring tissue-specific AAV transduction in mice through engineered receptor expression

Corresponding author name(s): Jan E. Carette

Editorial Notes: n/a

Reviewer Comments & Decisions:

Decision Letter, initial version:

Dear Jan,

Your Article, "Hardwiring tissue-specific AAV transduction in mice through engineered receptor expression", has now been seen by 3 reviewers. As you will see from their comments below, although the reviewers find your work of considerable potential interest, they have raised a number of concerns. We are interested in the possibility of publishing your paper in Nature Methods, but would like to consider your response to these concerns before we reach a final decision on publication.

We therefore invite you to revise your manuscript to address these concerns. Importantly, I strongly recommend you to focus on the basic biological applications of SELECTIV rather than the clinical aspects to keep with the editorial scope of Nature Methods.

* include a point-by-point response to the reviewers and to any editorial suggestions

* please underline/highlight any additions to the text or areas with other significant changes to facilitate review of the revised manuscript

- * address the points listed described below to conform to our open science requirements
- * ensure it complies with our general format requirements as set out in our guide to authors at www.nature.com/naturemethods
- * resubmit all the necessary files electronically by using the link below to access your home page

[Redacted] This URL links to your confidential home page and associated information about manuscripts you may have submitted, or that you are reviewing for us. If you wish to forward this email to co-authors, please delete the link to your homepage.

We hope to receive your revised paper within six weeks. If you cannot send it within this time, please let us know. In this event, we will still be happy to reconsider your paper at a later date so long as nothing similar has been accepted for publication at Nature Methods or published elsewhere.

OPEN SCIENCE REQUIREMENTS

REPORTING SUMMARY AND EDITORIAL POLICY CHECKLISTS

Please note that these forms are dynamic 'smart pdfs' and must therefore be downloaded and completed in Adobe Reader. We will then flatten them for ease of use by the reviewers. If you would

like to reference the guidance text as you complete the template, please access these flattened versions at <http://www.nature.com/authors/policies/availability.html>.

IMAGE INTEGRITY

DATA AVAILABILITY

All novel DNA and RNA sequencing data, protein sequences, genetic polymorphisms, linked genotype and phenotype data, gene expression data, macromolecular structures, and proteomics data must be deposited in a publicly accessible database, and accession codes and associated hyperlinks must be provided in the “Data Availability” section.

To further increase transparency, we encourage you to provide, in tabular form, the data underlying the graphical representations used in your figures. This is in addition to our data-deposition policy for specific types of experiments and large datasets. For readers, the source data will be made accessible directly from the figure legend. Spreadsheets can be submitted in .xls, .xlsx or .csv formats. Only one (1) file per figure is permitted: thus if there is a multi-paneled figure the source data for each panel should be clearly labeled in the csv/Excel file; alternately the data for a figure can be included in multiple, clearly labeled sheets in an Excel file. File sizes of up to 30 MB are permitted. When submitting source

data files with your manuscript please select the Source Data file type and use the Title field in the File Description tab to indicate which figure the source data pertains to.

Please include a “Data availability” subsection in the Online Methods. This section should inform readers about the availability of the data used to support the conclusions of your study, including accession codes to public repositories, references to source data that may be published alongside the paper, unique identifiers such as URLs to data repository entries, or data set DOIs, and any other statement about data availability. At a minimum, you should include the following statement: “The data that support the findings of this study are available from the corresponding author upon request”, describing which data is available upon request and mentioning any restrictions on availability. If DOIs are provided, please include these in the Reference list (authors, title, publisher (repository name), identifier, year). For more guidance on how to write this section please see: <http://www.nature.com/authors/policies/data/data-availability-statements-data-citations.pdf>

MATERIALS AVAILABILITY

SUPPLEMENTARY PROTOCOL

To help facilitate reproducibility and uptake of your method, we ask you to prepare a step-by-step Supplementary Protocol for the method described in this paper. We [encourage authors to share their step-by-step experimental protocols](https://www.nature.com/nature-research/editorial-policies/reporting-standards#protocols) on a protocol sharing platform of their choice and report the protocol DOI in the reference list. Nature Portfolio's Protocol Exchange is a free-to-use and open resource for protocols; protocols deposited in Protocol Exchange are citable and can be linked from the published article. More details can be found at www.nature.com/protocolexchange/about.

ORCID

Nature Methods is committed to improving transparency in authorship. As part of our efforts in this direction, we are now requesting that all authors identified as ‘corresponding author’ on published

papers create and link their Open Researcher and Contributor Identifier (ORCID) with their account on the Manuscript Tracking System (MTS), prior to acceptance. This applies to primary research papers only. ORCID helps the scientific community achieve unambiguous attribution of all scholarly contributions. You can create and link your ORCID from the home page of the MTS by clicking on 'Modify my Springer Nature account'. For more information please visit www.springernature.com/orcid.

Sincerely,
Madhura

Madhura Mukhopadhyay, PhD
Senior Editor
Nature Methods

Reviewers' Comments:

Reviewer #1:

Remarks to the Author:

The manuscript by Zengel et al, describes the development of transgenic mouse models that can direct AAV mediated transgene expression in specific cell types. As an alternative to traditional methods in generating such mouse lines with germline approaches which can be time- and resource-intensive they describe a new way of model generation they refer to as

SELECTIV. Their methodology enables efficient and specific transgene expression by coupling adeno-associated virus (AAV) vectors with Cre-inducible overexpression of the multi-serotype AAV receptor, AAVR. Their experimental demonstrations suggest that transgenic AAVR overexpression increases the efficiency of transduction of diverse cell types using recombinant AAVs.

Moreover superior specificity can be achieved by combining Cre-mediated AAVR overexpression with whole-body knockout of endogenous AAVR. The enhanced efficacy and specificity of the SELECTIV method will have utility in studies where cell specific targeting is of primary importance.

I am convinced that this highly tractable system will allow for faster production of model systems using currently available mice and AAVs, increasing both the speed and accuracy of basic and experimental

biomedical research. I however have several conceptual and technical criticisms which need to be addressed:

Conceptual: Although the enhancement of the specificity and strength of transgene expression via overexpression and k/o our AAVR is an attractive concept, and SELECTIV will facilitate cell type specific expression of desired transgenes across a range of tissues in mice, given the species-specific attributes of AAVR tissue distribution it is unclear how this approach provides an advantage as a testing platform for clinical AAV-based vectors in murine models. The authors should remove this claim or substantiate their claim by showing similar biodistribution of AAVR across several species in at least one tissue of biomedical interest.

Results presented in figure 1 show that the SELECTIV system functionally expressed spCas9 for use in CRISPR based experiments and this will be of interest for various gene k/o experiments that can be conducted with great selectivity using this tool. However, it is important to know if such long-term expression of both the guide RNAs and Cas9 at high levels leads to undesired off target modifications. The quantification of potential off target edits at least on the predicted sites with most likelihood of off targets would be necessary to validate this tool especially in timepoints beyond 30-60 days.

Based on data presented in figure 2, transduction with AAV2 or AAV8 vectors in AAVR overexpressing cells revealed that increases in transduction occurs in a serotype specific manner with AAV2 mediated expression increasing 8.5-fold whereas AAV8 rises 403-fold (Fig. 2c). Can the authors comment why there is such staggering differences between these serotypes?

Technical: In Extended Data Fig. 2- the Western blot analysis of tissue lysates from different mice shows two lanes for the SEL-WB and Control conditions and a single lane for AAVR k/o: can the authors explain if the two lanes correspond to technical replicates or samples from different mice? What is the lower mw band seen in the AAVR-KO in the liver? It seems the same band is present for the spleen samples and at least one of the control lanes for retina.

Reviewer #2:

Remarks to the Author:

Zengel et al. demonstrates a clever approach for enhancing AAV potency based on the authors' prior seminal discovery of AAVR as a receptor for many AAV serotypes. The method is based on AAVR overexpression and uses new transgenic mouse lines that either (1) constitutively overexpress AAVR, called SELECTIV or (2) knock out endogenous AAVR and introduce Cre-dependent AAVR overexpression (SELECTIV-KO). By breeding SELECTIV-KO mice with established Cre-transgenic mouse lines, offspring

compatible with enhanced AAV potency matched with Cre-selective tropism may be created. In Figure 2c the 403-fold increase for AAV8 is impressive.

The key benefit of this method is the potential for increased potency in mouse cell types and tissues that are resistant to existing AAV tools and the authors provide one nice example (muscle stem cells). The other in vivo examples (cardiomyocytes, liver, ChAT neurons), while showing that the method is robust and easy to use, do not show a unique capability that is not already possible with existing Cre lines (by including Lox sites in the AAV genome and using higher AAV doses than the low to moderate ones used here). Given the longer timelines and potential unknown effects on cell health and membrane properties from long term AAVR overexpression and knockout, the greatest benefit from these methods would be in cell types for which there are presently no viable tools. If the authors could show SELECTIV or SELECTIV-KO unlocking more cell populations that are resistant to existing AAV, notably microglia in the brain, the method would provide a key and much needed reagent to the community.

Major points:

1. Both the speed and translation claims in the manuscript regarding SELECTIV-KO should be diminished as the method still requires pre-existing Cre-transgenic mouse lines to cross with the SELECTIV-KO mice and waiting for offspring to mature. (The authors briefly note this in the discussion, “although it requires extra breeding”, but should be in the introduction as well.) While SELECTIV-KO can increase the efficiency of existing approaches, it does not add new specificity that cannot be achieved more quickly by simply pairing Cre-transgenic mice with AAV containing Lox sites.
2. It would be helpful to include examples of experimental paradigms that are now unlocked that were not possible before. Either based on the fold increases in reporter transgene potency or, ideally, supported by a demonstration experiment.
3. Clarification is needed on how overexpression of AAVR boosts AAV potency and transgene expression, especially in cell types with low endogenous AAVR expression, but AAVR haploinsufficiency in the control mice used throughout the manuscript does not introduce a detrimental effect. Can the authors include details on AAVR expression level dynamic ranges and ceiling/floor effects? Including heterozygous AAVR control animals in the AAVR protein expression blot in Extended Data Figure 2 would also be helpful.
4. It is promising that SELECTIV and SELECTIV-KO mice were “viable, fertile” but it would be helpful to include a more detailed characterization of cell health (beyond overall animal health, which might miss important subtleties for experimentally relevant cell populations). This is important both because of

possible effects of constitutive AAVR overexpression or knockout and because mCherry is known to aggregate.

5. In the discussion of the retinal layer experiments in Figure 3, the current state of the art should be more fully conveyed because the proposed 2-component system is both more difficult and not translatable.

Minor points:

1. Quantification of potency in Figure 6d would be helpful.
2. Extended Data Figure 4 presents the AAVR expression frequencies for many cell types. Notably, multiple cell types regarded as difficult for AAV to target (e.g. large intestine cells, macrophage) have broad AAVR expression. It would be helpful for readers if the manuscript more explicitly presented the tissues and cell types for which SELECTIV would be most beneficial.
3. It may be helpful for the discussion to note the AAVR independent serotypes that will not benefit from this method, such as AAV4 (<https://doi.org/10.1128/JVI.02213-17>).

Overall: For the many claims regarding the high AAV titers required in the clinic, AAVR overexpression is not a clear path for translation so the manuscript would be improved by focusing on what the method delivers: a platform for rodent experiments whose value will be based on the types of refractory cell types and tissues that are shown. So far the paper has only one (muscle stem cells) and an additional example (e.g. microglia or others) would make the work much more impactful.

Reviewer #3:

Remarks to the Author:

The manuscript by Zengel et al. is focused on developing inducible mouse models that allow selective overexpression of AAVR (the cognate receptor for most AAV serotypes) in specific cells/tissues, while simultaneously reducing expression in other organs. The authors showcase the utility of the SELECTIV mouse model approach in muscle stem cells, cardiomyocytes, hepatocytes and cholinergic neurons by evaluating the improved transduction efficiencies of different AAVs in vitro and in vivo. The observed differences in transduction efficiency raise a significant number of questions pertaining to the altered biology of AAV in these mouse models and the impact on utilizing this tool/resource for studying AAV biology is not obvious in this manuscript. Nevertheless, despite the varying degrees of improved

transduction observed, it is evident that this model system when combined with AAV vectors can be exploited for rapid analysis of tissue-specific overexpression or possibly KO of different genes by bypassing the need for new mouse model development for every target. Overall, the manuscript is well written, showcases the utility of the SELECTIV platform and is technically well executed. Some concerns and recommendations to consider are noted below:

1. The engineered locus appears to express AAVR-mCherry fusion protein based on the cassette schematic. No supporting data validating the expression of this fusion protein (by MW) is shown. Moreover, it is unclear whether such fusion impacts the co-localization of AAVR in tissues. Fluorescent images of mouse tissue assessing mCherry expression and possibly intracellular localization of the fusion would be important to validate if proposing to utilize this model to study AAV biology.
2. Relatedly, data pertaining to the altered biodistribution of AAV vectors (quantification of vector genomes/cell in target and off-target tissues) would be important to include. This would be critical information to include if proposing to utilize these models for studying AAV biology in vivo.
3. Does overexpression of AAVR impact other genes? (at least other key genes identified by this group such as GPR108 etc.). Have the authors carried out transcript analysis of assessed other genes that might be impacted through western blot/proteomics. Relatedly, western blot analysis of AAVR overexpression is presented without appropriate loading and housekeeping controls.
4. The AAVR KO/overexpression profile in circulating or splenic immune cell populations would be important to include. This may influence subsequent assessment of target genes/different organ systems since this may impact the AAV transduction profile and the immune response to AAV/transgene expression.

Author Rebuttal to Initial comments

Responses to reviewers

[Nature Methods manuscript submission [Redacted]]

We thank the reviewers for their positive comments and insightful suggestions concerning our manuscript. We have made modifications to the manuscript to refocus on the utility of the SELECTIV system in mouse model development instead of translational/clinical research. In the response to the reviewers comments we have obtained additional data that helps better characterize the expression of AAVR in the mice used in the manuscript and we address concerns related to the effect of *Aavr* expression on cellular transcription, mouse health, and mouse immune phenotypes. We have also obtained additional data concerning AAVR protein in various organs and AAVR-mCh detection by western blot and immunofluorescence.

Based on reviewer comments we felt that the original manuscript could benefit from clarifying the advantages of detargeting AAV in the SELECTIV-KO mice, so we have expanded the discussion on this. At the same time, we have more explicitly compared the drawbacks and benefits of the SELECTIV system to other existing systems. We have also outlined potential future experiments and how the SELETIV system can be used synergistically with advances in the AAV field.

We think that the manuscript presents compelling data that will entice readers of Nature Methods to want to use the SELETIV mice for their own research, and that this mouse line will become a widely used tool. In order to ensure that the SELECTIV system can be used by other researchers, we have confirmed with the Jackson Laboratory Repository that they are interested in providing these mice and we are depositing the mice with them. We believe that this will aid in the adoption of the SELECTIV model in transgenic model development and AAV research.

Reviewer #1:

Remarks to the Author:

The manuscript by Zengel et al, describes the development of transgenic mouse models that can direct AAV mediated transgene expression in specific cell types. As an alternative to traditional methods in generating such mouse lines with germline approaches which can be time- and resource-intensive they describe a new way of model generation they refer to as SELECTIV. Their methodology enables efficient and specific transgene expression by coupling adeno-associated virus (AAV) vectors with Cre-inducible overexpression of the multi-serotype AAV receptor, AAVR. Their experimental demonstrations suggest that transgenic AAVR overexpression increases the efficiency of transduction of diverse cell types using recombinant AAVs.

Moreover superior specificity can be achieved by combining Cre-mediated AAVR overexpression with whole-body knockout of endogenous AAVR. The enhanced efficacy and specificity of the SELECTIV method will have utility in studies where cell specific targeting is of primary importance.

I am convinced that this highly tractable system will allow for faster production of model systems using currently available mice and AAVs, increasing both the speed and accuracy of basic and experimental biomedical research. I however have several conceptual and technical criticisms which need to be addressed:

Conceptual: Although the enhancement of the specificity and strength of transgene expression via overexpression and k/o our AAVR is an attractive concept, and SELECTIV will facilitate cell type specific expression of desired transgenes across a range of tissues in mice, given the species-specific attributes of AAVR tissue distribution it is unclear how this approach provides an advantage as a testing platform for clinical AAV-based vectors in murine models. The authors

should remove this claim or substantiate their claim by showing similar biodistribution of AAVR across several species in at least one tissue of biomedical interest.

We thank the reviewer for highlighting the utility of the approach as a highly tractable system allowing faster production of model systems using currently available mice and AAVs. Although the system allows for precise control of expression of AAV-based vectors in preclinical mouse models, we agree with the reviewer that the potential usefulness in testing clinical AAV-based vectors is complicated by species-specific differences. We have therefore removed claims regarding direct translational utility in testing AAV vectors throughout the manuscript but highlighted its use in basic and experimental biomedical research. We did include a more nuanced and specific discussion on how SELECTIV might be useful in preclinical research in early development in the manuscript:

“Because of the increased specificity and efficiency of the SELECTIV system, we believe that this may be useful for testing of preclinical AAV-delivered transgenes. While enforcing the tissue specificity by overexpressing the receptor does not allow one to test the tropism of AAV capsid variants, it does allow one to test a “best case scenario” for cell-type specific transgene delivery in a mouse model. This includes identifying functional differences in types of cells targeted or testing of multiple transgenes to identify the best construct during early development. This can be done in parallel while researching capsid variants and promoters in parallel, speeding up specific aspects of preclinical research.”

Results presented in figure 1 show that the SELECTIV system functionally expressed spCas9 for use in CRISPR based experiments and this will be of interest for various gene k/o experiments that can be conducted with great selectivity using this tool. However, it is important to know if such long-term expression of both the guide RNAs and Cas9 at high levels leads to undesired off target modifications. The quantification of potential off target edits at least on the predicted sites with most likelihood of off targets would be necessary to validate this tool especially in timepoints beyond 30-60 days.

We appreciate the concern for off-target effects in *in vivo* experiments. The most crucial element in avoiding off-target effects is the choice of the guide RNA. The guide RNA we used for our studies was based on a publication that carefully assessed *in vivo* off-targets effects of different guide RNAs using a sophisticated and elaborate method to assess *in vivo* off-targets¹. This study showed the impact of guide RNA choice ranging from readily detected off-target effects (with a guide RNA deliberately chosen to have many sites with a small number of mismatches) to no detectable off target effects with the optimal guide (named gMH). Because we used gMH for our studies, we expect to not be able to demonstrate any off-target effects. Interestingly, off-targets observed with the suboptimal guide in the same study were already observed early (4 days post injection of the adenovirus vector) and did not increase after prolonged Cas9/gRNA expression

(30 days), suggesting that long-term expression is less important than guide RNA choice. Because our system is comparable to other approaches that rely on stable expression of spCas9 in mice² it is unlikely that it is inherently more prone to off target effects. Moreover, because gRNA choice is so dominant, we do not think determining the off-target effects for a particular guide RNA will be informative.

Based on data presented in figure 2, transduction with AAV2 or AAV8 vectors in AAVR overexpressing cells revealed that increases in transduction occurs in a serotype specific manner with AAV2 mediated expression increasing 8.5-fold whereas AAV8 rises 403-fold (Fig. 2c). Can the authors comment why there is such staggering differences between these serotypes?

Certain serotypes including AAV2 can use cell surface heparan sulphate proteoglycans (HSPGs) or other glycans as low affinity attachment factors, which increase *in vitro* transduction efficiency³. Although we have shown that AAVR is critical for transduction of these serotypes (AAVR-KO cells are resistant to transduction), the gain of AAVR overexpression is less dramatic because they are more efficient at using alternative attachment factors. AAV8 on the other hand does not have a well characterized attachment factor⁴. AAV8 likely uses AAVR as a primary attachment factor *in vitro*, similar to what we see with AAV9 *in vivo*. This effect was also observed in MEFs, where AAV8 and AAV9 show almost no transduction without overexpression of AAVR. We have now incorporated this in the manuscript: “*The fold increase was especially apparent for AAV8 and AAV9, which are known to have low transduction efficiency in vitro⁵ and therefore likely depend more on AAVR expression levels. However, even for AAV2, which efficiently transduces in vitro by use heparan sulfate proteoglycans (HSPGs) as a primary attachment factor⁶, we found that AAVR overexpression substantially enhanced transduction.*”

Technical: In Extended Data Fig. 2- the Western blot analysis of tissue lysates from different mice shows two lanes for the SEL-WB and Control conditions and a single lane for AAVR k/o: can the authors explain if the two lanes correspond to technical replicates or samples from different mice? What is the lower mw band seen in the AAVR-KO in the liver? It seems the same band is present for the spleen samples and at least one of the control lanes for retina.

These individual lanes are from different mice, which we have now explicitly stated in the text. Based on this question and similar questions from the other reviewers (Reviewer #2, Point 3 and Reviewer #3, Point 1), we have repeated the western blot analysis and increased the number of mice to n=3 for *Aavr*^{+/+}, *Aavr*^{+/-}, *Aavr*^{-/-} and SELECTIV-WB and performed semi-quantitative analysis (Reviewer Fig. 1, Manuscript Extended Data Fig. 2a,b.).

Figure 1. a, Western blot analysis of tissue from mice. Tissue from *Aavr*^{+/+}, *Aavr*^{+/-}, *Aavr*^{-/-} and, SELECTIV-WB mice were homogenized and AAVR and GAPDH protein was detected by western blot. SELECTIV-WB tissues were diluted 1:10 for heart and lung samples and other samples had the same weight of tissue loaded in each well. AAVR was not detected in AAVR-KO mice, while SELECTIV-WB mice have highly increased AAVR protein levels. **b**, Semi-quantitative analysis was performed by quantifying the relative expression of AAVR compared to GAPDH for each sample (all samples normalized to *Aavr*^{+/+} for each tissue).

The availability for AAVR antibodies that can detect wild-type levels of AAVR is very limited, so we are using a rabbit polyclonal that has multiple background bands that are present in different amounts in different tissues (**Reviewer Fig. 2**). The inclusion of *Aavr*^{-/-} mice allowed us to assign and quantify the band corresponding to endogenous AAVR. As expected, the level of AAVR protein expression was reduced by approximately half in *Aavr*^{+/-} mice compared to *Aavr*^{+/+} mice. Overexpression in SELECTIV-WB was apparent and showed 2 bands at higher molecular weight than endogenous AAVR. The ratio of these bands differed in different tissues and we speculate that the lower band may be due to proteolytic cleavage either *in vivo* or during tissue collection and processing or ectodomain shedding similar to what is seen with the related protein KIAA0319⁷. By updating Manuscript Extended Data Fig. 2, we believe we have improved the characterization of the extent of overexpression (even with the limitations of the available antibodies).

Figure 2. Western blot analysis of AAVR expression in mice from *Aavr*^{+/+}, *Aavr*^{+/-}, *Aavr*^{-/-} and, SELECTIV-WB mice had tissue harvested and homogenized for analysis of AAVR expression by western blot. Blots of the lysate from liver and brain show AAVR expression at ~150kDa and AAVR-mCherry expression at ~180kDa.

Reviewer #2:

Remarks to the Author:

Zengel et al. demonstrates a clever approach for enhancing AAV potency based on the authors' prior seminal discovery of AAVR as a receptor for many AAV serotypes. The method is based on AAVR overexpression and uses new transgenic mouse lines that either (1) constitutively overexpress AAVR, called SELECTIV or (2) knock out endogenous AAVR and introduce Cre-dependent AAVR overexpression (SELECTIV-KO). By breeding SELECTIV-KO mice with established Cre-transgenic mouse lines, offspring compatible with enhanced AAV potency matched with Cre-selective tropism may be created. In Figure 2c the 403-fold increase for AAV8 is impressive.

The key benefit of this method is the potential for increased potency in mouse cell types and tissues that are resistant to existing AAV tools and the authors provide one nice example (muscle stem cells). The other *in vivo* examples (cardiomyocytes, liver, ChAT neurons), while showing that the method is robust and easy to use, do not show a unique capability that is not already possible with existing Cre lines (by including Lox sites in the AAV genome and using higher AAV doses than the low to moderate ones used here). Given the longer timelines and potential unknown effects on cell health and membrane properties from long term AAVR overexpression and knockout, the greatest benefit from these methods would be in cell types for which there are presently no viable tools. If the authors could show SELECTIV or SELECTIV-KO unlocking more cell populations that are resistant to existing AAV, notably microglia in the brain, the method would provide a key and much needed reagent to the community.

Major points:

1. Both the speed and translation claims in the manuscript regarding SELECTIV-KO should be diminished as the method still requires pre-existing Cre-transgenic mouse lines to cross with the SELECTIV-KO mice and waiting for offspring to mature. (The authors briefly note this in the discussion, "although it requires extra breeding", but should be in the introduction as well.) While SELECTIV-KO can increase the efficiency of existing approaches, it does not add new specificity that cannot be achieved more quickly by simply pairing Cre-transgenic mice with AAV containing Lox sites.

We appreciate the valid point that this system should be compared with alternative methods. We have edited the text to introduce the requirement for breeding earlier in the text and now more explicitly mentioned this as *limitation* of the approach in the discussion.

We have already compared our approach to generating a “minipromoter” approach in the Discussion and have now also included the approach mentioned by the reviewer (by including Lox sites in the AAV genome and using higher AAV doses than the low to moderate ones used in our manuscript) in the discussion. We have included “*Another powerful system is the DIO/FLEx system, which has been developed to mediate Cre-specific transgene expression*^{8,9}. *This system uses specialized AAV vectors that contain paired recombination sites to allow for Cre-mediated inversion of the transgene coding sequence. While this system has been used extensively, there can be non-specific transgene expression due to recombination during vector production, which requires optimization*¹⁰. *Because the SELECTIV-KO system enhances transduction efficiency and specificity by regulating AAVR, it does not require specialized AAV vectors.*”

Although previously developed systems have great utility, we believe that our system increases the breadth of cell types that can be efficiently targeted, while allowing researchers to use previously characterized AAV vectors. Minipromoter or Lox-based systems require specialized vectors and do not have the benefits of increasing transduction efficiency in the target tissue.

We also believe that the SELECTIV-KO system has additional utility in given the ability to detarget transduction and AAV capsid deposition in other organs. This may be critical for preventing toxicity that would be associated with high levels of AAV required in other systems, which may result in cellular toxicity due to capsid deposition in other tissues such as the liver and nervous system, even when transduction is not seen in systems such as the AAV-lox system. Toxicity has been demonstrated in a dose dependent manner in mice and the toxicity has been shown to be independent of transgene expression^{11,12}. Toxicity has also been demonstrated in the dorsal root ganglia of mice upon high doses AAV administration¹³. The SELECTIV-KO system allows for robust transduction of cell types of interest without requiring high doses of AAV. We have included in the text of the manuscript: “*This was coupled with complete absence of transduction in the liver and muscle, while using an “off the shelf” AAV vector. The SELECTIV model system will increase the range of cell types targetable by AAV vectors, while also allowing for increased efficiency of targeting cell types of interest. The latter facilitates usage of lower AAV dosages, which reduces costs and decreases the likelihood of AAV vector mediated toxicity*¹⁴.”

2. It would be helpful to include examples of experimental paradigms that are now unlocked that were not possible before. Either based on the fold increases in reporter transgene potency or, ideally, supported by a demonstration experiment.

As we have mentioned in the point above, we believe that the utility of the approach extends beyond rare cell types. In our manuscript, we have robustly showed the utility of the approach using different AAV serotypes, upon local and systemic administration of AAV vectors, using 5 different Cre-driver lines (E2A-Cre, Pax7-CE, Alb-Cre, Myh6-Cre, and Chat-Cre), and by reporter gene transduction and biodistribution in diverse organs (eye, muscle, heart, liver, brain). We believe that experimentally showing more examples extends beyond the scope of this manuscript. The flexibility of the approach will allow its application in many more areas, and we have expanded the discussion on this.

3. Clarification is needed on how overexpression of AAVR boosts AAV potency and transgene expression, especially in cell types with low endogenous AAVR expression, but AAVR haploinsufficiency in the control mice used throughout the manuscript does not introduce a detrimental effect. Can the authors include details on AAVR expression level dynamic ranges and ceiling/floor effects? Including heterozygous AAVR control animals in the AAVR protein expression blot in Extended Data Figure 2 would also be helpful.

In different parts of the manuscript, we have used *Aavr*^{+/-} *Cre*^{-/-} control mice. We have done this because using these controls allowed us to use genetically matched *Cre*^{-/-} mice that were littermates. The need for using littermate controls is well documented¹⁵. We agree with the reviewer that haploinsufficiency could influence transduction efficiency. Previous experiments showed a small effect of a heterozygous knockout allele of *Aavr*^{+/-} in FVB mice on transduction by AAV9 although this was not statistically significant with the number of mice tested¹⁶. Balancing the need for littermate controls with the potential of this small effect, we chose to use the *Aavr*^{+/-} *Cre*^{-/-} mice as control. We have now included in the text of the manuscript “*As littermate controls, we use mice that do not express Cre and are heterozygous for Aavr (named “Control”). We weighted the distinct advantage of using littermate controls¹⁵ against the potential disadvantage of a heterozygous Aavr knockout allele which may show haploinsufficiency. Although we cannot exclude haploinsufficiency in all tissues, we have previously found that heterozygous deletion of endogenous Aavr does not significantly reduce AAV9-luciferase transduction upon systemic administration¹⁶.*”

We have now updated the manuscript to include tissue from *Aavr*^{+/+}, *Aavr*^{+/-}, *Aavr*^{-/-} and SELECTIV-WB mice with 3 mice per group (**Reviewer Fig. 1**, Manuscript Extended Data Fig. 2a).

Figure 3. *In vivo* transduction experiments previously performed¹⁶. **a**, FVB mice (*Aavr*^{+/+}, *Aavr*^{+/-}, or *Aavr*^{-/-}) were transduced with AAV9 encoding luciferase and transduction was assessed by *in vivo* imaging. **b**, Transduction was tracked over time for 14 days and average radiance was reported. **c**, Data from day 7 post transduction comparing the various mice show no significant difference between transduction in *Aavr*^{+/+} and *Aavr*^{+/-} mice.

4. It is promising that SELECTIV and SELECTIV-KO mice were “viable, fertile” but it would be helpful to include a more detailed characterization of cell health (beyond overall animal health, which might miss important subtleties for experimentally relevant cell populations). This is important both because of possible effects of constitutive AAVR overexpression or knockout and because mCherry is known to aggregate.

We thank the reviewer for suggesting to further characterize cell health in the SELECTIV and SELECTIV-KO mice. The physiological role of AAVR (also named KIAA0319L and AU040320) is unknown. It has homology with KIAA0319, which is associated with dyslexia. Because of the association with dyslexia, *Aavr*^{-/-} (*AU040320*^{-/-}) mice have been thoroughly characterized focusing on detailed examination of the developmental trajectory of the neocortex in mice¹⁷. The authors noted that *AU040320*^{-/-} mice were born with the expected Mendelian ratios. Absence of *AU040320*

does not (i) alter cortical neurogenesis, (ii) lead to abnormalities in neuronal lamination, or (iii) alter radial migration in the neocortex in the mouse brain. Furthermore AU040320^{-/-} mice did not display any differences compared to wild type controls in startle reflex, prepulse inhibition, or gap detection. These tests indicate that AU040320^{-/-} mice are not impaired in the ability to discriminate subtle differences in auditory stimuli in the presence of noise. The only test where there was a significant difference between wild-type and AU040320^{-/-} mice was for suprathreshold deficits in auditory processing using auditory brainstem responses. As we have also noted AU040320^{-/-} mice are sterile caused by a disruption of acrosome biogenesis¹⁸. Thus, previous studies only implicate AAVR in dyslexia and sperm cell generation. We have characterized blood analytes (**Reviewer Fig. 4**) and immune cell populations (**Reviewer Fig. 5**, Manuscript Extended Data Fig. 2c,d) and did not find statistical differences compared to wild-type suggesting no global changes. However, we agree that AAVR over or under expression could influence the specific biological system being studied. We have now included this as a potential limitation of the SELECTIV system in the text of the manuscript: *“The limitations of the approach are that it requires extra breeding and that it is dependent on AAVR knockout and overexpression. The latter may introduce potential effects on host physiology. The physiological role of AAVR is unknown but knockout mice are viable, born with expected Mendelian ratios, have no obvious abnormalities in neuronal migration or cortical anatomy, and have a subtle defect in auditory processing¹⁷. Male Aavr^{-/-} males are infertile and present a globozoospermia-like phenotype¹⁸.”*

Figure 4. Clinical chemistry on mice with varying *Aavr* expression. Terminal bleeds were performed on *Aavr*^{+/+}, *Aavr*^{+/-}, *Aavr*^{-/-} and, SELECTIV-WB mice (n=3 or n=2 for ion levels in some mice where blood volume was limited) and clinical chemistry analysis was performed on the Siemens Dimension EXL200/LOCI analyzer. Values were normalized for all testes to the *Aavr*^{+/+} group. There was no significant difference for any group when compared to *Aavr*^{+/+} for any of the measurements.

Figure 5. Immune cell profiles in the spleen and circulation of mice. **a**, Spleens from *Aavr*^{+/+}, *Aavr*^{+/-}, *Aavr*^{-/-} and SELECTIV-WB mice (n=3) were processed and immune cell profiles were determined by flow cytometry. There was no significant difference in immune cell counts for any of the groups when compared to *Aavr*^{+/+}. **b**, Immune cells in the blood were determined by automated hematology on the Sysmex XN-1000V hematology analyzer system. There was no significant difference in immune cell counts for any of the groups when compared to *Aavr*^{+/+}.

We also appreciate the possibility of mCherry fusion protein aggregation, which has previously been reported and may be associated with lysosomal localization²¹. Although in our case mCherry is fused to AAVR which lowers its expression and limits its localization we cannot exclude effects. Despite this, mCherry is widely used in mouse research so this is not unique to the SELECTIV system. For future iterations, it may be reasonable to develop a model with untagged AAVR or with a next generation tag such as mScarlet or Crimson.

5. In the discussion of the retinal layer experiments in manuscript Figure 3, the current state of the art should be more fully conveyed because the proposed 2-component system is both more difficult and not translatable.

Sorry for the confusion, we should have been more clear that we do not suggest to use the 2-component system for translational applications. Our system was able to demonstrate the ability to better target photoreceptors using less invasive intravitreal injection, compared the subretinal injections previously required for efficient photoreceptor transduction. Another group recently identified AAV capsid variants that can target photoreceptors after intravitreal injection in wild-type mice²⁰. Our system allows for the use of a commonly available AAV serotype, while their system allows for the use of wild-type mice. It is also possible that these systems would be complementary if used together. There has also been an effort to use AAV-PHP.eB to transduce retinal cells²¹, although IVT injection did not result in efficient retinal transduction. It is possible that using the SELECTIV-KO system in conjunction with a retinal specific Cre line²², would allow

for photoreceptor and RPE transduction after systemic or IVT delivery of AAV-PHP.eB. We believe that the SELECTIV system will work in a complementary fashion with ongoing discovery and production of novel AAV capsids. We have updated that discussion with this text and hope that it represents the current state of the art and how the SELECTIV system may contribute to this field. The text in the manuscript now reads *“We similarly found that AAVR overexpression enhanced photoreceptor layer transduction in the eye after intravitreal injection. This suggests that AAVR expression could be rate-limiting for transduction in addition to the proposed anatomical barriers²³ that could limit transduction of this clinically relevant cell type. Although there is no direct relevance for translational use of this system for gene therapy, we believe that this will create new opportunities in biomedical research in preclinical models. Our system demonstrated the ability to better target photoreceptors using less invasive intravitreal injection, compared the subretinal injections previously required for efficient photoreceptor transduction. Another group recently identified AAV capsid variants that can target photoreceptor cells after intravitreal injection in wild-type mice²⁰, while the SELECTIV system allows for the use of a commonly available AAV serotype. It is also possible that these systems would be complementary if used together. One benefit of the SELECTIV system is the ability to impose selectivity using the SELECTIV-KO system in conjunction with a Cre line specific to retinal cell types of interest^{22,24}. This could allow for specific photoreceptor or RPE transduction after intravitreal delivery with the newly discovered vectors or possibly after systemic delivery using AAV-PHP.eB, which has recently been shown to transduce the retina after systemic delivery²¹. We believe that the SELECTIV system has great potential to work in complementary fashion with novel capsids to develop unique transduction strategies to enhance biomedical research in mice.”*

Minor points:

1. Quantification of potency in Figure 6d would be helpful.

While all pictures were taken at the same setting, we do not know if quantification would be completely accurate due to some differences in the exact slice depth and depth of the z-stack used to make the images. We would need to collect additional data with new mice to properly assess a quantitative measurement of transduction, but we believe that the current data shows a clear difference between WT and SELECTIV-KO-Chat transduction.

2. Extended Data Figure 4 presents the AAVR expression frequencies for many cell types. Notably, multiple cell types regarded as difficult for AAV to target (e.g. large intestine cells, macrophage) have broad AAVR expression. It would be helpful for readers if the manuscript more explicitly presented the tissues and cell types for which SELECTIV would be most beneficial.

While we think that our system is uniquely able to target cell types with little to no AAVR expression, we think that this system is more broadly useful for many applications, including in cell types that express AAVR. This can be seen in the highly efficient heart targeting in the SELECTIV-KO-MyH6 mice, even though cardiac muscle cells show a high percentage of cells expressing *Aavr* according to the Tabula Muris data. Even in this cell type that is relatively well transduced we showed a 10-fold increase in transduction compared on control mice (Manuscript Fig. 4b).

We have included the analysis of % of cells with *Aavr* expression in mouse cells based on data from the Tabula Muris consortium, but we do not think that this should be the only resource used to identify research that may benefit from using the SELECTIV mice. Showing efficient transduction may also be related to cell location and fast turnover (large intestine cells). It is also possible that the Tabula Muris data is not very robust for some cell types, including macrophages, which have been shown to be difficult to capture in these large data sets²⁶. Given the robust transduction data that has been shown for the five Cre lines tested, we are confident that it will translate to other systems that people are interested in.

3. It may be helpful for the discussion to note the AAVR independent serotypes that will not benefit from this method, such as AAV4 (<https://doi.org/10.1128/JVI.02213-17>).

We have included data showing that AAV4 was not affected by *Aavr* overexpression in Manuscript Fig. 1b and mentioned that this is the expected result in results section. We have now added in the text: “*The large majority of AAV serotypes belonging to divergent evolutionary lineages utilize AAVR for their entry, while only two related serotypes (AAV4 and AAVrh32.33) have been shown to enter independently from AAVR^{16,26}.*”

Overall: For the many claims regarding the high AAV titers required in the clinic, AAVR overexpression is not a clear path for translation so the manuscript would be improved by focusing on what the method delivers: a platform for rodent experiments whose value will be based on the types of refractory cell types and tissues that are shown. So far the paper has only one (muscle stem cells) and an additional example (e.g. microglia or others) would make the work much more impactful.

We agree that we have not demonstrated clear translational relevance for this research, and we have modified the manuscript to deemphasize the translational aspects of this research and focus on the usefulness of the model development that the SELECTIV system makes possible.

We believe that we have demonstrated that our system allows for a unique ability to detarget both AAV-based transduction and capsid deposition in mice. This allows for robust transduction in

tissues without off-target transduction. While MuSCs were the only truly refractory cell type that we used to demonstrate *in vivo* transduction, we have also demonstrated improved transduction in other cell types including cardiac muscle. We have also shown specific transduction of cholinergic neurons by PHP.eB, demonstrating the ability of this system to specifically target cells in the brain after systemic vector delivery. Given the multiple examples we already show in the manuscript in different tissues, using local and systemic methods of delivery, and using different AAV serotypes, we believe that additional examples is beyond the scope of the manuscript.

Reviewer #3:

Remarks to the Author:

The manuscript by Zengel et al. is focused on developing inducible mouse models that allow selective overexpression of AAVR (the cognate receptor for most AAV serotypes) in specific cells/tissues, while simultaneously reducing expression in other organs. The authors showcase the utility of the SELECTIV mouse model approach in muscle stem cells, cardiomyocytes, hepatocytes and cholinergic neurons by evaluating the improved transduction efficiencies of different AAVs in vitro and in vivo. The observed differences in transduction efficiency raise a significant number of questions pertaining to the altered biology of AAV in these mouse models and the impact on utilizing this tool/resource for studying AAV biology is not obvious in this manuscript. Nevertheless, despite the varying degrees of improved transduction observed, it is evident that this model system when combined with AAV vectors can be exploited for rapid analysis of tissue-specific overexpression or possibly KO of different genes by bypassing the need for new mouse model development for every target. Overall, the manuscript is well written, showcases the utility of the SELECTIV platform and is technically well executed. Some concerns and recommendations to consider are noted below:

We thank the reviewer for their positive assessment of the work.

1. The engineered locus appears to express AAVR-mCherry fusion protein based on the cassette schematic. No supporting data validating the expression of this fusion protein (by MW) is shown. Moreover, it is unclear whether such fusion impacts the co-localization of AAVR in tissues. Fluorescent images of mouse tissue assessing mCherry expression and possibly intracellular localization of the fusion would be important to validate if proposing to utilize this model to study AAV biology.

We appreciate the comment on how we demonstrated AAVR-mCherry overexpression in this paper and that this was not clear. We have repeated the western blot analysis and increased the number of mice to n=3 for *Aavr*^{+/+}, *Aavr*^{+/-}, *Aavr*^{-/-} and SELECTIV-WB and performed semi-quantitative analysis (Reviewer Fig. 1, Manuscript Extended Data Fig. 2a,b). We also included a low exposure for the SELECTIV-WB mice to better show the gel shift due to the fusion. We have also included a dilution experiment that very clearly shows the shift in size for the AAVR-mCh protein (Reviewer Fig. 6). We have shown the endogenous mCherry signal in the SELECTIV-KO-Chat mice, although the AAVR-mCh distribution may have been less clear due to the high levels of GFP transduction. We have included an image of just nuclear staining and mCherry signal

(Reviewer Fig. 7). The mCherry signal appears to be perinuclear as expected but can also be seen in the dendrites.

We have also deemphasized the utility of this model to study AAV biology and focused on the usefulness the SELECTIV model for specific transgene expression when combined with various AAV vectors for delivery.

Figure 6. Size analysis of AAVR and AAVR-mCherry in mouse tissue. In order to better characterize the size difference in AAVR and AAVR-mCh on the same western blot, tissue from wild-type control mice were loaded alongside dilutions of SELECTIV-WB mouse tissue. The blot was performed with an anti-AAVR antibody and developed. The shift in size can be seen with AAVR at approximately 150kDa and AAVR-mCh at approximately 180kDa, as expected.

Figure 7. AAVR-mCherry expression in cholinergic neurons of SELECTIV-KO-Chat mice. Images showing transduction in Manuscript Fig. 6 without the GFP channel are shown with only red signal (AAVR-mCh) and blue (nuclear stain). AAVR-mCherry can be seen in the Pons and Medulla. Scale bar: 50 μm .

2. Relatedly, data pertaining to the altered biodistribution of AAV vectors (quantification of vector genomes/cell in target and off-target tissues) would be important to include. This would be critical information to include if proposing to utilize these models for studying AAV biology in vivo.

We agree that vector biodistribution is critical for demonstrating the utility of the SELECTIV-KO model system. We believe that the PET/CT data is able to clearly show the altered distribution in the SELECTIV-KO-MyH6 mice compared to control mice (Manuscript Fig. 4d-f). This novel methodology was previously developed by authors on this paper and the usefulness of this system was shown in detail previously²⁷. We think that using this method provides a robust demonstration of the role of AAVR on AAV vector distribution. We found a reduced off target deposition of AAV vector particles concomitant with increased vector particles in target tissue.

3. Does overexpression of AAVR impact other genes? (at least other key genes identified by this group such as GPR108 etc.). Have the authors carried out transcript analysis of assessed other genes that might be impacted through western blot/proteomics. Relatedly, western blot analysis of AAVR overexpression is presented without appropriate loading and housekeeping controls.

We appreciate the suggestion to investigate if AAVR overexpression impacts other genes. To test this, we performed mRNA sequencing and transcriptomic comparisons on some of the MEFs generated from Control and SELECTIV-WB embryos, which were used for the transduction experiments in Manuscript Fig. 1b. Using multiple biological and experimental replicates from

each line and comparing the entire transcriptome, only AU040320 (AAVR) was shown to have a significant and large difference in expression (**Reviewer Fig. 8**). All other gene had either a $p(\text{adj}) < 0.01$ or a fold change less than 1.5-fold. Of the few genes that were significant but with a fold change less than 1.5-fold, there was nothing that has a known function in AAV transduction.

Figure 8. Expression profile in Control and SELECTIV-WB mouse embryonic fibroblasts. MEFs were generated from littermate controls were grown and total RNA was isolated for mRNA-seq analysis to assess gene expression profiles by Novogene (Davis) and comparisons were made between Control (*Aavr*^{+/+}) and SELECTIV-WB MEFs. A horizontal line shows the adjusted p value of 0.01 and the vertical lines show a change of ± 1.5 -fold between the two groups.

We have also updated our analysis of AAVR abundance in tissue to include additional mice along with loading controls and quantification based on loading (**Reviewer Fig. 1**, Manuscript Extended

Data Fig. 2a,b). Although we only believe this is semiquantitative due to the large difference in expression between endogenous AAVR and expression in SELETIV mice, we think it shows a clear overexpression in all tissues tested.

4. The AAVR KO/overexpression profile in circulating or splenic immune cell populations would be important to include. This may influence subsequent assessment of target genes/different organ systems since this may impact the AAV transduction profile and the immune response to AAV/transgene expression.

Although AAVR has not been implicated as having a role in immune responses, we cannot exclude that AAVR might impact immune cell populations. To address these concerns, we used the same mice used in the western blot analysis to also determine the composition of splenic and circulating immune cells (**Reviewer Fig. 5**, Extended Data Fig. 2c,d). We found that there were no significant differences in the immune cell populations for *Aavr* KO or overexpressing mice compared to mice with wild-type levels of AAVR. This demonstrates that, at least at baseline, there is no global difference in immune cell populations for these mice.

1. Akcakaya, P. *et al.* In vivo CRISPR editing with no detectable genome-wide off-target mutations. *Nature* **561**, 416–419 (2018).
2. Platt, R. J. *et al.* CRISPR-Cas9 knockin mice for genome editing and cancer modeling. *Cell* **159**, 440–455 (2014).
3. Qiu, J., Handa, A., Kirby, M. & Brown, K. E. The interaction of heparin sulfate and adeno-associated virus 2. *Virology* **269**, 137–147 (2000).
4. Pillay, S. & Carette, J. E. Host determinants of adeno-associated viral vector entry. *Curr. Opin. Virol.* **24**, 124–131 (2017).
5. Ellis, B. L. *et al.* A survey of ex vivo/in vitro transduction efficiency of mammalian primary cells and cell lines with Nine natural adeno-associated virus (AAV1-9) and one engineered adeno-associated virus serotype. *Virol. J.* **10**, 74 (2013).

6. Shen, S., Bryant, K. D., Brown, S. M., Randell, S. H. & Asokan, A. Terminal N-linked galactose is the primary receptor for adeno-associated virus 9. *J. Biol. Chem.* **286**, 13532–13540 (2011).
7. Velayos-Baeza, A., Levecque, C., Kobayashi, K., Holloway, Z. G. & Monaco, A. P. The dyslexia-associated KIAA0319 protein undergoes proteolytic processing with {gamma}-secretase-independent intramembrane cleavage. *J. Biol. Chem.* **285**, 40148–40162 (2010).
8. Sohal, V. S., Zhang, F., Yizhar, O. & Deisseroth, K. Parvalbumin neurons and gamma rhythms enhance cortical circuit performance. *Nature* **459**, 698–702 (2009).
9. Atasoy, D., Aponte, Y., Su, H. H. & Sternson, S. M. A FLEX switch targets Channelrhodopsin-2 to multiple cell types for imaging and long-range circuit mapping. *J. Neurosci.* **28**, 7025–7030 (2008).
10. Fischer, K. B., Collins, H. K. & Callaway, E. M. Sources of off-target expression from recombinase-dependent AAV vectors and mitigation with cross-over insensitive ATG-out vectors. *Proc. Natl. Acad. Sci. U. S. A.* **116**, 27001–27010 (2019).
11. Xiong, W. *et al.* AAV cis-regulatory sequences are correlated with ocular toxicity. *Proc. Natl. Acad. Sci. U. S. A.* **116**, 5785–5794 (2019).
12. Khabou, H., Cordeau, C., Pacot, L., Fisson, S. & Dalkara, D. Dosage thresholds and influence of transgene cassette in adeno-associated virus-related toxicity. *Hum. Gene Ther.* **29**, 1235–1241 (2018).
13. Bolt, M. W., Brady, J. T., Whiteley, L. O. & Khan, K. N. Development challenges associated with rAAV-based gene therapies. *J. Toxicol. Sci.* **46**, 57–68 (2021).

14. Hinderer, C. *et al.* Severe toxicity in nonhuman primates and piglets following high-dose intravenous administration of an adeno-associated virus vector expressing human SMN. *Hum. Gene Ther.* **29**, 285–298 (2018).
15. Holmdahl, R. & Malissen, B. The need for littermate controls. *Eur. J. Immunol.* **42**, 45–47 (2012).
16. Pillay, S. *et al.* An essential receptor for adeno-associated virus infection. *Nature* **530**, 108–112 (2016).
17. Guidi, L. G. *et al.* Knockout mice for dyslexia susceptibility gene homologs KIAA0319 and KIAA0319L have unaffected neuronal migration but display abnormal auditory processing. *Cereb. Cortex* **27**, 5831–5845 (2017).
18. Guidi, L. G. *et al.* AU040320 deficiency leads to disruption of acrosome biogenesis and infertility in homozygous mutant mice. *Sci. Rep.* **8**, 10379 (2018).
19. Costantini, L. M. *et al.* A palette of fluorescent proteins optimized for diverse cellular environments. *Nat. Commun.* **6**, 7670 (2015).
20. Pavlou, M. *et al.* Novel AAV capsids for intravitreal gene therapy of photoreceptor disorders. *EMBO Mol. Med.* **13**, (2021).
21. Palfi, A. *et al.* AAV-PHP.eB transduces both the inner and outer retina with high efficacy in mice. *Mol. Ther. Methods Clin. Dev.* **25**, 236–249 (2022).
22. Ivanova, E., Hwang, G.-S. & Pan, Z.-H. Characterization of transgenic mouse lines expressing Cre recombinase in the retina. *Neuroscience* **165**, 233–243 (2010).

23. Dalkara, D. *et al.* Inner limiting membrane barriers to AAV-mediated retinal transduction from the vitreous. *Mol. Ther.* **17**, 2096–2102 (2009).
24. Schneider, S. *et al.* Generation of an inducible RPE-specific Cre transgenic-mouse line. *PLoS One* **13**, e0207222 (2018).
25. Millard, S. M. *et al.* Fragmentation of tissue-resident macrophages during isolation confounds analysis of single-cell preparations from mouse hematopoietic tissues. *Cell Rep.* **37**, 110058 (2021).
26. Dudek, A. M. *et al.* An alternate route for adeno-associated virus (AAV) entry independent of AAV receptor. *J. Virol.* **92**, (2018).
27. Seo, J. W. *et al.* Positron emission tomography imaging of novel AAV capsids maps rapid brain accumulation. *Nat. Commun.* **11**, 2102 (2020).

Decision Letter, first revision:

Dear Jan,

Thank you for submitting your revised manuscript "Hardwiring tissue-specific AAV transduction in mice through engineered receptor expression" (NMEMH-A49338A). It has now been seen by the original referees and their comments are below. The reviewers find that the paper has improved in revision, and therefore we'll be happy in principle to publish it in Nature Methods, pending minor revisions to satisfy the referees' final requests and to comply with our editorial and formatting guidelines.

TRANSPARENT PEER REVIEW

Nature Methods offers a transparent peer review option for new original research manuscripts submitted from 17th February 2021. We encourage increased transparency in peer review by publishing the reviewer comments, author rebuttal letters and editorial decision letters if the authors agree. Such peer review material is made available as a supplementary peer review file. Please state in the cover letter 'I wish to participate in transparent peer review' if you want to opt in, or 'I do not wish to participate in transparent peer review' if you don't. Failure to state your preference will result in delays in accepting your manuscript for publication.

ORCID

Sincerely,
Madhura

Madhura Mukhopadhyay, PhD
Senior Editor
Nature Methods

Reviewer #1 (Remarks to the Author):

The authors have taken into account all of the criticism of this reviewer either in the text or in figures where needed. The answers to comments are satisfactory. The manuscript has been improved. This new version is appropriate for publication.

Reviewer #2 (Remarks to the Author):

We thank the authors for removing their translational claims for the SELECTIV system, providing additional data and text on cell health concerns, and for contextualizing the retinal layer experiments in manuscript Figure 3 with comparison to the current state of the art.

In the authors response to our first major point, we appreciate the inclusion of discussion of Cre-dependent vectors. However, the authors should minimize language such as “specialized AAV vectors” or “off the shelf” vectors that can create confusion between Cre- or Flp-dependent vectors and constitutive vectors. Cre-dependent vectors are well-established and widely available from all vector cores and companies. Addgene, for example, offers more Cre-/Flp-dependent AAV genomes than constitutively-expressing genomes among their standard AAV vector offerings. This dichotomy is used to justify the increased time and difficulty entailed in the SELECTIV system by mating existing Cre-dependent mouse lines to SELECTIV mice. Balancing the cost savings of using lower AAV doses in the SELECTIV system against the increased animal costs and multi-month delays, it is not obvious that SELECTIV would come out on top most of the time – authors should include a discussion around this to clarify for the reader.

The authors do raise specialized concerns in some applications: pre-flipped genomes (though the reference used to support this includes a simple AAV genome modification that solves the issue), potential off-target toxicity (primarily an issue in primates, not mice), and intersectional targeting by pairing Cre-lines of one specificity and AAV genome promoters a second specificity (which is still in many cases with cell type promoters and lox sites). In experiments confronting these specific challenges, SELECTIV would have a clear advantage but unfortunately, the authors do not include any experimental examples in these settings, a validation necessary to backup the utility and versatility of SELECTIV. The authors also declined to provide a second example of an AAV-resistant cell type, as requested in our second major point. In our opinion this leaves the manuscript light on experiments highlighting the unique benefits of the SELECTIV system that would drive its adoption and somewhat vague on those applications that would most benefit from adopting the SELECTIV system.

In response to our third major point, the authors now include Reviewer Fig #1, which shows that the AAVR +/- control mice do have markedly lower AAVR expression across all tissues. However, the reviewers decline to update their text to discuss how increasing AAVR expression increases AAV potency but decreasing AAVR expression does not decrease AAV potency. That prior work with a coarse-grained

luciferase assay relying on enzymatic signal amplification could not detect differences in AAV potency does not remove this conceptual point. Given that many potency-enhancement claims for SELECTIV are based on comparison to control mice with lower than WT AAVR expression, we believe this topic deserves greater treatment.

Reviewer #3 (Remarks to the Author):

The authors have addressed reviewer concerns and the revised manuscript is much improved.

Author Rebuttal, first revision:

Reviewer #1:

Remarks to the Author:

The authors have taken into account all of the criticism of this reviewer either in the text or in figures where needed. The answers to comments are satisfactory. The manuscript has been improved. This new version is appropriate for publication.

Thank you. We appreciate your time and previous comments. We believe that edits made based on your suggestion improved the paper.

Reviewer #2:

Remarks to the Author:

We thank the authors for removing their translational claims for the SELECTIV system, providing additional data and text on cell health concerns, and for contextualizing the retinal layer experiments in manuscript Figure 3 with comparison to the current state of the art.

Thank you. We believe that focusing the paper more clearly on development of the SELECTIV mouse system for research use has improved the manuscript. Your comments have allowed us to better place the SELECTIV system within the current state of the art to show how it can be used in conjunction with current and future technology.

In the authors response to our first major point, we appreciate the inclusion of discussion of Cre-dependent vectors. However, the authors should minimize language such as “specialized AAV vectors” or “off the shelf” vectors that can create confusion between Cre- or Flp-dependent vectors and constitutive vectors. Cre-dependent vectors are well-established and widely available from all vector cores and companies. Addgene, for example, offers more Cre-/Flp-dependent AAV genomes than constitutively-expressing genomes among their standard AAV vector offerings. This dichotomy is used to

justify the increased time and difficulty entailed in the SELECTIV system by mating existing Cre-dependent mouse lines to SELECTIV mice. Balancing the cost savings of using lower AAV doses in the SELECTIV system against the increased animal costs and multi-month delays, it is not obvious that SELECTIV would come out on top most of the time – authors should include a discussion around this to clarify for the reader.

We have tried to highlight the current state of the art and how we think the SELECTIV system can fit within this. We have included in the discussion possible drawbacks of the system including *“Some limitations are the extra breeding requirements and the modulation of a host gene, AAVR, which may affect host physiology. The physiological role of AAVR is unknown but knockout mice are viable, born with expected Mendelian ratios, have no obvious abnormalities in neuronal migration or cortical anatomy, and have a subtle defect in auditory processing. Male Aavr^{-/-} males are infertile and present a globozoospermia-like phenotype.”* We also have tried to limit language that minimizes the previous technology and instead highlight how these systems may be complementary.

The authors do raise specialized concerns in some applications: pre-flipped genomes (though the reference used to support this includes a simple AAV genome modification that solves the issue), potential off-target toxicity (primarily an issue in primates, not mice), and intersectional targeting by pairing Cre-lines of one specificity and AAV genome promoters a second specificity (which is still in many cases with cell type promoters and lox sites). In experiments confronting these specific challenges, SELECTIV would have a clear advantage but unfortunately, the authors do not include any experimental examples in these settings, a validation necessary to backup the utility and versatility of SELECTIV. The authors also declined to provide a second example of an AAV-resistant cell type, as requested in our second major point. In our opinion this leaves the manuscript light on experiments highlighting the unique benefits of the SELECTIV system that would drive its adoption and somewhat vague on those applications that would most benefit from adopting the SELECTIV system.

We appreciate the concerns related to the potentially usefulness of the SELECTIV system. We believe that the ability to transduce difficult to transduce cell types such as muscle stem cells and MEFs efficiently is an improvement on any current system. We hope to soon have additional data from our group or other highlighting this in other cell types. We believe that one of the strongest uses for this system is the detargeting aspect of the SELECTIV-KO system, which can be seen in the heart, liver, and cholinergic neurons. All of which show high levels of transduction while eliminating transduction in other cells.

In response to our third major point, the authors now include Reviewer Fig #1, which shows that the AAVR +/- control mice do have markedly lower AAVR expression across all tissues. However, the reviewers decline to update their text to discuss how increasing AAVR expression increases AAV potency but decreasing AAVR expression does not decrease AAV potency. That prior work with a coarse-grained luciferase assay relying on enzymatic signal amplification could not detect differences in AAV potency

does not remove this conceptual point. Given that many potency-enhancement claims for SELECTIV are based on comparison to control mice with lower than WT AAVR expression, we believe this topic deserves greater treatment.

In order to address this concern, we have included text explaining why we used AAVR +/- mice in these studies: *"We generated SELECTIV-KO-Myh6 mice that express Cre in heart cardiomyocytes and modestly in pulmonary vascular smooth muscle cells, while endogenous Aavr is not expressed (Fig. 4a, Extended Data Fig. 6). As littermate controls, we use mice that do not express Cre and are heterozygous for Aavr (named "Control"). We weighted the distinct advantage of using littermate controls⁴¹ against the potential disadvantage of a heterozygous Aavr knockout allele which may show haploinsufficiency. Although we cannot exclude haploinsufficiency in all tissues, we have previously found that heterozygous deletion of endogenous Aavr does not significantly reduce AAV9-luciferase transduction upon systemic administration."* We also believe it is important to note that all mice with Cre-dependent overexpression of AAVR in these figures are also AAVR +/- at the original locus, so the overexpression of AAVR is responsible for the phenotypes seen. We also have multiple experiments that compare mice with wildtype expression of AAVR to mice with overexpression in the SELETIV-WB mice, including in Figure 1 and 3.

Reviewer #3:

Remarks to the Author:

The authors have addressed reviewer concerns and the revised manuscript is much improved.

Thank you. We appreciate your time and previous comments. We believe that edits made based on your suggestion improved the paper.

Final Decision Letter:

Dear Jan,

I am pleased to inform you that your Article, "Hardwiring tissue-specific AAV transduction in mice through engineered receptor expression", has now been accepted for publication in Nature Methods. Your paper is tentatively scheduled for publication in our June print issue, and will be published online prior to that. The received and accepted dates will be May 23, 2022 and Apr 25, 2023. This note is intended to let you know what to expect from us over the next month or so, and to let you know where to address any further questions.

Acceptance is conditional on the data in the manuscript not being published elsewhere, or announced in the print or electronic media, until the embargo/publication date. These restrictions are not intended to

deter you from presenting your data at academic meetings and conferences, but any enquiries from the media about papers not yet scheduled for publication should be referred to us.

Once your paper is typeset, you will receive an email with a link to choose the appropriate publishing options for your paper and our Author Services team will be in touch regarding any additional information that may be required.

Please note that *Nature Methods* is a Transformative Journal (TJ). Authors may publish their research with us through the traditional subscription access route or make their paper immediately open access through payment of an article-processing charge (APC). Authors will not be required to make a final decision about access to their article until it has been accepted. [Find out more about Transformative Journals](https://www.springernature.com/gp/open-research/transformative-journals)

Your paper will now be copyedited to ensure that it conforms to Nature Methods style. Once proofs are generated, they will be sent to you electronically and you will be asked to send a corrected version within 24 hours. It is extremely important that you let us know now whether you will be difficult to contact over the next month. If this is the case, we ask that you send us the contact information (email, phone and fax) of someone who will be able to check the proofs and deal with any last-minute problems.

If, when you receive your proof, you cannot meet the deadline, please inform us at rjsproduction@springernature.com immediately.

Once your manuscript is typeset and you have completed the appropriate grant of rights, you will receive a link to your electronic proof via email with a request to make any corrections within 48 hours. If, when you receive your proof, you cannot meet this deadline, please inform us at rjsproduction@springernature.com immediately.

Once your paper has been scheduled for online publication, the Nature press office will be in touch to confirm the details.

Once your paper has been scheduled for online publication, the Nature press office will be in touch to confirm the details.

Content is published online weekly on Mondays and Thursdays, and the embargo is set at 16:00 London time (GMT)/11:00 am US Eastern time (EST) on the day of publication. If you need to know the exact publication date or when the news embargo will be lifted, please contact our press office after you have submitted your proof corrections. Now is the time to inform your Public Relations or Press Office about your paper, as they might be interested in promoting its publication. This will allow them time to prepare an accurate and satisfactory press release. Include your manuscript tracking number [Redacted] and the name of the journal, which they will need when they contact our office.

About one week before your paper is published online, we shall be distributing a press release to news organizations worldwide, which may include details of your work. We are happy for your institution or funding agency to prepare its own press release, but it must mention the embargo date and Nature Methods. Our Press Office will contact you closer to the time of publication, but if you or your Press Office have any inquiries in the meantime, please contact press@nature.com.

Nature Portfolio journals [encourage authors to share their step-by-step experimental protocols](https://www.nature.com/nature-research/editorial-policies/reporting-standards#protocols) on a protocol sharing platform of their choice. Nature Portfolio 's Protocol Exchange is a free-to-use and open resource for protocols; protocols deposited in Protocol Exchange are citable and can be linked from the published article. More details can found at www.nature.com/protocolexchange/about.

Please note that you and any of your coauthors will be able to order reprints and single copies of the issue containing your article through Nature Portfolio 's reprint website, which is located at <http://www.nature.com/reprints/author-reprints.html>. If there are any questions about reprints please send an email to author-reprints@nature.com and someone will assist you.

Best regards,
Madhura

Madhura Mukhopadhyay, PhD
Senior Editor
Nature Methods